# miR-708-5p is elevated in bipolar patients and can induce mood disorder-associated behavior in mice

Carlotta Gilardi [1], Helena C Martins[1], Brunno Rocha Levone [1], Alessandra Lo Bianco[1], Silvia Bicker[1], Pierre-Luc Germain [1,2,3], Fridolin Gross[1,10], Ayse Özge Sungur[4,5,6,7], Theresa M Kisko[4,5,6,7], Frederike Stein[8], Susanne Meinert[9], Rainer K W Schwarting [4], Markus Wöhr [4,5,6,7], Udo Dannlowski[9], Tilo Kircher[8] & Gerhard Schratt [1]✉

## Abstract

**Mood disorders (MDs) are caused by an interplay of genetic and environmental (GxE) risk factors. However, molecular pathways engaged by GxE risk factors are poorly understood. Using small-RNA sequencing in peripheral blood mononuclear cells (PBMCs), we show that the bipolar disorder (BD)-associated microRNA miR-708-5p is upregulated in healthy human subjects with a high genetic or environmental predisposition for MDs. miR-708-5p is further upregulated in the hippocampus of rats which underwent juvenile social isolation, a model of early life stress. Hippocampal overexpression of miR-708-5p in adult male mice is sufficient to elicit MD-associated behavioral endophenotypes. We further show that miR-708-5p directly targets Neuronatin (*Nnat*), an endoplasmic reticulum protein. Restoring *Nnat* expression in the hippocampus of miR-708-5p-overexpressing mice rescues miR-708-5p-dependent behavioral phenotypes. Finally, miR-708-5p is upregulated in PBMCs from patients diagnosed with MD. Peripheral miR-708-5p expression allows to differentiate male BD patients from patients suffering from major depressive disorder (MDD). In summary, we describe a potential functional role for the miR-708-5p/Nnat pathway in MD etiology and identify miR-708-5p as a potential biomarker for the differential diagnosis of MDs.**

**Keywords** Biomarker; Bipolar Disorder; Genetic & Environmental Risk Factors; Mood Disorders; Neuronatin
**Subject Categories** Molecular Biology of Disease; Neuroscience; RNA Biology

## Introduction

Mood-disorders (MDs), which include major depressive disorder (MDD) and bipolar disorder (BD), are a group of chronic psychiatric diseases affecting mood and cognition. While MDD is characterized by long, persistent, depressive episodes, BD has a distinct pattern of mood oscillations, from depressive episodes (like in MDD), and (hypo)manic phases. Furthermore, BD is subclassified in BD type I (mania and depression), and BD type II (hypomania and depression). It is known that MDs are highly heritable and share a common genetic signature (McGuffin et al, 2003). In particular, BD is characterized by up to 70% heritability in monozygotic twins (Craddock and Sklar, 2013). However, complex polygenetic mechanisms cannot fully explain the onset of MD, and environmental factors, such as early life stress (ELS; e.g., physical and sexual abuse, emotional neglect), play an important role in the etiology and outcome of MD (Aas et al, 2020; Nemeroff, 2016; Rodriguez et al, 2021). Moreover, MDD has a strong sex-specific component (e.g., higher prevalence of MDD, but not BD in females) (Noble, 2005), whose underlying biological mechanisms are unknown.

Although recapitulating the full spectrum of MD-associated symptoms is challenging, specific endophenotypes can be reliably assessed by employing dedicated behavioral tests in rodents. These include, for example, behavioral despair (forced swim test/FST and tail suspension test/TST), anhedonia (sucrose preference), anxiety (open field test/OFT and elevated plus maze/EPM), cognitive impairments (Y-maze, novel object recognition/NOR) and compulsive/manic-like behavior (marble burying) (Hoffman, 2013). Various chronic stress models have been established to mimic the impact of stress on MD symptomatology. Among them, post-weaning juvenile social isolation in rats represents a robust model to induce MD-associated endophenotypes, e.g., deficits in social communication and cognitive abilities (Seffer et al, 2015; Valluy et al, 2015). On the other hand, haploinsufficiency for the *Cacna1c* cross-disorder psychiatric risk gene is frequently used to study gene *vs* environment (GxE) interactions relevant for MD (Dedic et al,

[1]Laboratory of Systems Neuroscience, Institute for Neuroscience, Department of Health Science and Technology, ETH Zurich, 8057 Zurich, Switzerland. [2]Laboratory of Molecular and Behavioural Neuroscience, Institute for Neuroscience, Department of Health Science and Technology, ETH Zurich, 8057 Zurich, Switzerland. [3]Lab of Statistical Bioinformatics, IMLS, University of Zürich, 8057 Zurich, Switzerland. [4]Behavioral Neuroscience, Experimental and Biological Psychology, Faculty of Psychology, Philipps-University of Marburg, D-35032 Marburg, Germany. [5]Center for Mind, Brain and Behavior, Philipps-University Marburg, D-35032 Marburg, Germany. [6]Laboratory of Biological Psychology, Social and Affective Neuroscience Research Group, Research Unit Brain and Cognition, Faculty of Psychology and Educational Sciences, KU Leuven, B-3000 Leuven, Belgium. [7]Leuven Brain Institute, KU Leuven, B-3000 Leuven, Belgium. [8]Department of Psychiatry and Psychotherapy, University of Marburg, Marburg, Germany. [9]Institute for Translational Psychiatry, University of Münster, Münster, Germany. [10]Present address: CNRS UMR5164 ImmunoConcEpT, University of Bordeaux, Bordeaux, France. ✉E-mail: Gerhard.schratt@hest.ethz.ch

2018). For example, *Cacna1c*$^{+/-}$ mice show decreased immobility in the TST and FST, higher preference for sucrose as well as decreased anxiety behavior, although the latter was only observed in female mice (Dao et al, 2010). Moreover, in *Cacna1c*$^{+/-}$ rats, a long-term environmental impact on object recognition, spatial memory and reversal learning was observed (Braun et al, 2019).

microRNAs (miRNAs) are a large family of small (~22 nt), noncoding RNAs that act as posttranscriptional regulators by binding to complementary sequences in the 3'-untranslated region (UTR) of target messenger RNAs (mRNAs) (Bartel, 2018). A recent publication from our lab has highlighted the role of miRNAs in the pathogenesis of MDs (Martins and Schratt, 2021). The main pathways affected by miRNA dysregulation in the context of MD animal models include serotonergic neurotransmission (e.g., miR-16, miR-34), glucocorticoid signaling (e.g., miR-17-92 cluster, miR-15), neurotrophins (e.g., miR-182), Wnt signaling (e.g., miR-124), calcium homeostasis (e.g., miR-499-5p) and synaptic plasticity (e.g., miR-134, miR-218). While these candidate studies in rodents provided important insight into biological mechanisms, the translational value for MD therapeutics and diagnostics is mostly limited by the lack of corresponding human data.

In humans, miRNAs have been associated with MD etiology in expression studies of postmortem brain tissue (Moreau et al, 2011) and blood samples from living patients (Dwivedi, 2011). Differential expression of miRNAs in peripheral blood mononuclear cells (PBMCs) has been previously investigated for biomarker discovery in MD. For example, we previously found that miR-499-5p was significantly upregulated in female and male PBMC samples of BD patients (Martins et al, 2022). miR-124-3p was found to be significantly upregulated in PBMCs samples of MDD compared to healthy controls and it was downregulated after 8 weeks of antidepressant treatment (He et al, 2016). However, the overlap between these studies is usually low, and therefore these attempts have so far not yielded reliable miRNA biomarkers in MDs.

We therefore undertook an unbiased, back-translational approach to identify MD-associated miRNAs from a large human cohort of healthy subjects at high genetic and environmental risk to develop MD. One of the identified GxE regulated miRNAs, the BD-associated miR-708-5p (Forstner et al, 2015), was further functionally characterized in rodent models and subsequently tested for its diagnostic potential in MD patient subgroups.

## Results

### miR-708-5p is upregulated in the peripheral blood of human healthy subjects harboring an elevated genetic and environmental risk for MD

We hypothesized that miRNAs whose expression correlates with environmental (ER) and genetic risk (GR) for MD in human subjects might be good candidates for molecules with a functional role in MD pathophysiology. Furthermore, by starting from a human cohort, we hoped to identify miRNA candidates with high translational potential, meaning that they could serve as targets for miRNA therapeutics and/or diagnostics in MD.

To identify candidate miRNAs in an unbiased manner, we performed small-RNA sequencing with total RNA obtained from peripheral blood mononuclear cells (PBMCs) of healthy subjects ($n = 52$) from the FOR2107 cohort (Kircher et al, 2019) characterized by a high genetic

(GR group; at least one first degree relative diagnosed with a mood disorder; $n = 14$) or environmental (ER group; childhood trauma based on childhood trauma questionnaire (CTQ) score; $n = 16$) predisposition to develop MDs (Fig. 1A; Appendix Table S1). We chose PBMCs for this analysis since they represent a very reliable source of miRNAs, and the correlation between brain expression and PBMCs is often higher as between brain and plasma or serum (Kos et al, 2022). Only samples obtained from female subjects were considered, since we were unable to form homogeneous GR and ER groups for males due to an overall lower representation of male samples in the cohort. We then focused on miRNAs that were differentially expressed in both ER and GR compared to healthy control subjects (CTL group; no known genetic or environmental risk factors; $n = 18$). We found that a total of six miRNAs (miR-412-5p, miR-100-5p, miR-501-3p, miR-642a-5p, miR-4999-5p, miR-708-5p) fulfilled this criterium (Fig. 1B–E). Among them, miR-501-3p was previously found to be downregulated in SCZ (Liang et al, 2022), whereas miR-100-5p was associated with specific Huntington disease states (Diez-Planelles et al, 2016). However, miR-708-5p was of specific interest, since its expression was previously shown to be induced by various forms of cellular stress (Behrman et al, 2011; Lin et al, 2015; McIlwraith et al, 2022; Rodriguez-Comas et al, 2017; Yang et al, 2015) and it had been associated with BD (Forstner et al, 2015). We therefore decided to focus on miR-708-5p for our further studies. By assessing our samples by qPCR, we confirmed that miR-708-5p is significantly upregulated in GR and ER groups in comparison to CTL (Fig. 1F), and that other unrelated miRNAs—miR-16-5p, miR-30a-5p, and miR-1248-5p—did not change in the ER group compared to controls (Fig. EV1A–C). Thus, we focused our further studies on the characterization of miR-708-5p functional roles in MDs.

### miR-708-5p is expressed in rat hippocampal neurons and upregulated in the hippocampus of rat models of environmental or genetic risk for MDs

To characterize miR-708-5p roles on MD, we used rodents, as they have been extensively used to model psychiatric conditions. As a first step, we investigated whether miR-708-5p was expressed in different regions of the rat brain (Fig. 2A). We observed robust expression in regions classically implicated in MDs and cognitive function, such as the amygdala, frontal cortex, and hippocampus (Fig. 2A). Next, we assessed miR-708-5p expression in rat hippocampal primary neurons. Single-molecule fluorescence in situ hybridization (sm-FISH) showed that miR-708-5p is expressed in both CamK2a+ excitatory and GAD2+ inhibitory neurons in rat primary hippocampal neurons, illustrating its widespread neuronal distribution (Fig. 2B). miR-708-5p expression in rat hippocampal cultures decreased over time during in vitro development, indicating a role for miR-708-5p at early stages of neuronal development (Fig. 2C). Motivated by the findings of miR-708-5p elevation in peripheral blood of humans at risk to develop MDs (Fig. 1B–F), we hypothesized that miR-708-5p would be similarly dysregulated in rat genetic and environmental models of MDs. As a genetic model, we chose rats heterozygous for *Cacna1c* (*Cacna1c*$^{+/-}$), a repeatedly validated cross-disorder psychiatric risk gene (Bhat et al, 2012; Dao et al, 2010; Dedic et al, 2018; Harrison et al, 2022). Environmental risk was modeled by juvenile social isolation, a widely recognized model for early life trauma, e.g., childhood maltreatment (Braun et al, 2019; Seffer et al, 2015) (Fig. 2D). We focused on male rats since we previously observed that juvenile social isolation had pronounced effects on memory and social behavior in this sex

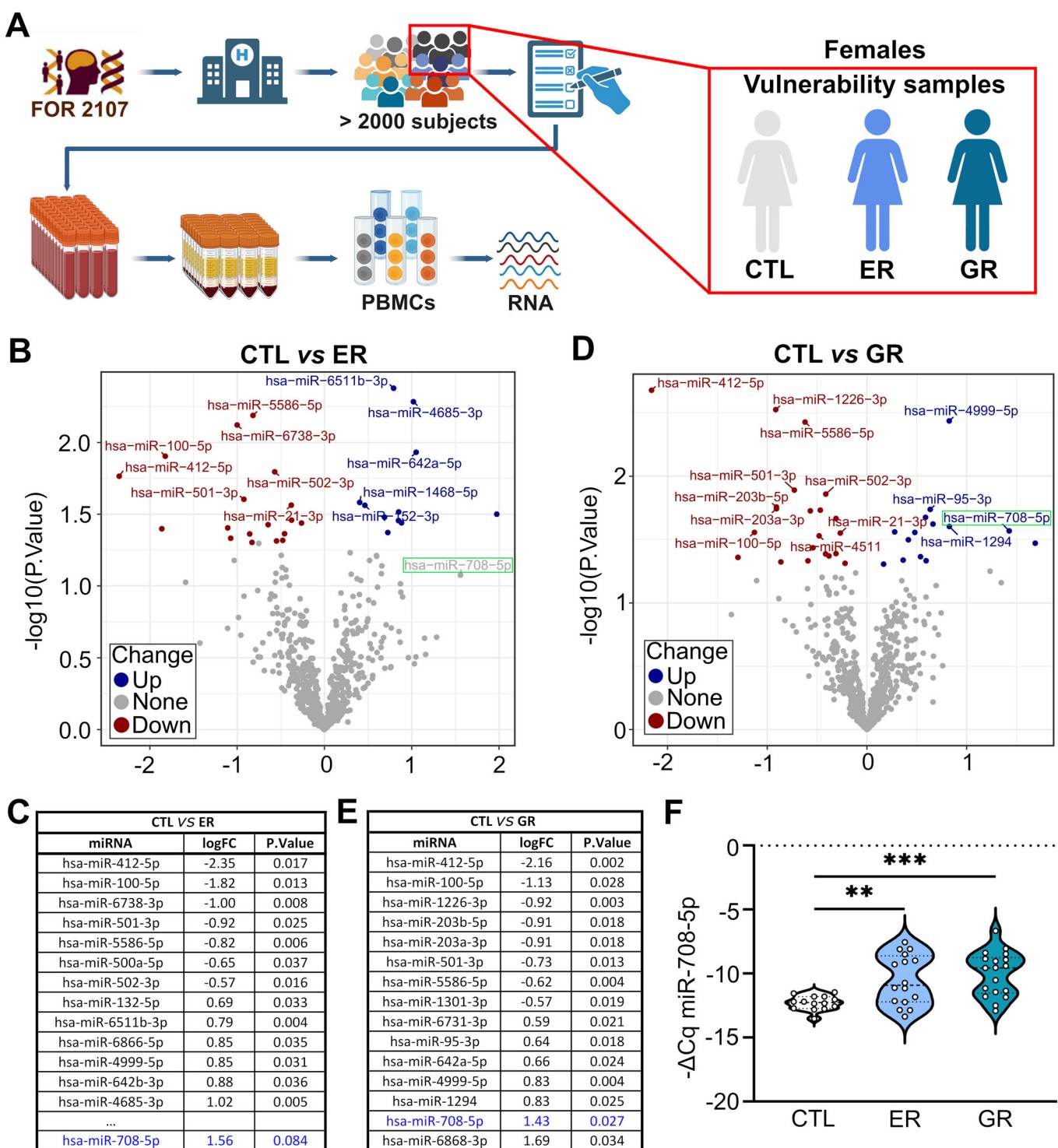

(Braun et al, 2019; Seffer et al, 2015). miR-708-5p was significantly upregulated by social isolation in the hippocampus of juvenile wild-type male rats compared to their group-housed counterparts (Fig. 2E), indicating parallels between the early life stress-related modulation of miR-708-5p expression in human peripheral blood and rat brain. Furthermore, *Cacna1c* heterozygosity led to an upregulation of miR-708-5p in the rat hippocampus, suggesting that, in analogy to our

human PBMC data (Fig. 1D–F), a genetic predisposition for MDs is sufficient to induce miR-708-5p expression in the rat brain (Fig. 2E). Surprisingly, juvenile social isolation did not significantly increase miR-708-5p levels in *Cacna1c*$^{+/-}$ rats compared to group-housed controls ($p = 0.11$), suggesting that Cacna1c heterozygosity attenuates the response to early life stress. Furthermore, the expression of an unrelated miRNA—miR129-5p (Fig. 2F)—remained unchanged

Figure 1. miR-708-5p is upregulated in the peripheral blood of human subjects at risk for MDs.

(A) Schematic representation of the workflow to obtain PBMCs samples from the FOR2107 cohort (see also Methods and Protocols section). Subjects underwent psychiatric and cognitive evaluation. After the data was collected, blood was withdrawn and PBMCs prepared followed by RNA extraction and downstream gene expression analysis. Samples were collected from psychiatrically healthy female subjects (CTL), or psychiatrically healthy female subjects with genetic predisposition for mood disorder (genetic risk – GR) or that suffered from childhood maltreatment (environmental risk – ER) (Vulnerability samples). (B) Volcano plot depicting differentially expressed miRNAs based on small-RNA sequencing from RNA isolated from PBMCs of CTL ($n = 18$) vs ER ($n = 18$) subjects. Up- (blue) and downregulated (red) miRNAs are labeled. Statistics: differential expression analysis (limma/voom with correction for two technical batches). (C) Table showing the top 15 differentially expressed miRNAs from CTL vs ER subjects, ordered from smallest to largest p-value. miRNA-708-5p is highlighted in blue. Statistics: differential expression analysis (limma/voom with correction for two technical batches). (D) Volcano plot depicting differentially expressed miRNAs based on small-RNA sequencing from RNA isolated from PBMCs of CTL ($n = 18$) vs GR ($n = 14$) subjects. Up- (blue) and downregulated (red) miRNAs are labeled. Statistics: differential expression analysis (limma/voom with correction for two technical batches). (E) Table showing the top 15 differentially expressed miRNAs from CTL vs GR subjects, ordered from smallest to largest p-value. miRNA-708-5p is highlighted in blue. Statistics: differential expression analysis (limma/voom with correction for two technical batches). (F) miR-708-5p qPCR analysis of total RNA isolated from PBMCs of CTL ($n = 18$), ER ($n = 16$), and GR ($n = 18$) subjects. One-way ANOVA with Holm-Sidak's multiple comparisons test, CTL vs ER, **$P = 0.0064$; CTL vs GR, ***$P = 0.0006$; ER vs GR, ns. Data are presented as violin plots with median, quartiles and data points. See also: Fig. EV1. Source data are available online for this figure.

across the groups. Our analysis across various experimental paradigms revealed that miR-708-5p expression is linked to both environmental stressors and genetic factors associated with MDs in both humans and rodents. Therefore, manipulation of miR-708-5p in the rodent brain represents a viable strategy to study the involvement of miR-708-5p in the development of mood disorder-associated endophenotypes as well as the underlying molecular mechanisms.

## miR-708-5p overexpression in the mouse hippocampus is sufficient to induce MD-associated endophenotypes

We next investigated whether stress-induced upregulation of miR-708-5p in the rodent hippocampus is causally involved in the development of MD-associated behavioral endophenotypes. Towards this end, miR-708-5p was ectopically overexpressed in the mouse hippocampus using stereotactic injection of a recombinant rAAV expressing a miR-708-5p hairpin (hp) under the control of the human synapsin (hSYN) promoter (Fig. EV2A). The hippocampus was chosen for functional manipulation since it represents a key brain structure for cognitive and emotional processing, and our previous data indicates strong effects of juvenile social isolation on hippocampal miRNA expression (Martins et al, 2022; Valluy et al, 2015) (Fig. 2E). To ensure the functionality of the overexpressing construct, we performed luciferase assays upon transfection of miR-708-5p over-expressing plasmid (hp708) together with a luciferase sensor harboring two perfect binding sites for miR-708-5p. Thereby, we detected a significant decrease in luciferase activity upon miR-708-5p over-expression compared to the control (hpCTL) (Fig. EV2B), demonstrating efficient repression of the reporter gene by the overexpressed miR-708-5p. We went on to perform stereotactic surgeries on seven to eight-week-old mice, which then underwent a four-week recovery period before behavioral assessments were conducted (Fig. 3A). Our approach led to a widespread infection of both the dorsal and ventral hippocampus (Fig. 3B). We first assessed cognitive function since cognitive impairments are frequently observed in MD patients. Both male and female mice overexpressing miR-708-5p in the hippocampus displayed an impairment in short-term memory in the Novel Object Recognition (NOR) test, failing to discriminate between familiar and novel objects after a brief 5-min interval (Fig. 3C–E; Appendix Fig. S1A–D). All animals spent a comparable amount of time exploring the objects (Appendix Fig. S1B,D), ruling out possible artifacts. As the results in male mice were more promising than in females, we also tested long-term memory in a separate cohort of mice

and showed that miR-708-5p overexpression led to a significant impairment in long-term memory (24-h delay) (Fig. EV2C–F). We then tested behavioral despair with the Tail Suspension Test (TST). Male mice overexpressing miR-708-5p in the hippocampus exhibited a significant decrease in immobility during the TST (Fig. 3F), indicative of reduced behavioral despair, an effect that was not observed in female mice (Fig. 3G). Furthermore, we observed a strong, although non-significant, trend for elevated anxiety-related compulsive behavior, assessed by the Marble Burying Test (MBT), in miR-708-5p overexpressing male mice (Fig. EV2G). Interestingly, neither male nor female mice with miR-708-5p overexpression showed significant differences in saccharin preference compared to control groups, suggesting no alteration in anhedonia-like behavior (Appendix Fig. S1E,F). We then used the elevated plus maze (EPM) to assess innate anxiety-like behavior, but no significant changes were found in the time spent in the open or closed arms, neither in males nor in female mice (Appendix Fig. S1G–J). Additionally, miR-708-5p overexpression did not affect passive avoidance in male mice (Appendix Fig. S2A). Locomotor activity assessed with the Open Field Test (OFT), remained unchanged, ruling out changes in locomotion as a confounding factor in our behavioral tests (Appendix Fig. S2B,C). At the end of the behavioral cohorts, hippocampal tissue analysis confirmed a robust miR-708-5p overexpression, when compared to control AAV-infected mice (Fig. 3H; Appendix Fig. S2D,E). Together, our results provide strong evidence that hippocampal overexpression of miR-708-5p leads to distinct MD-associated behaviors, including impaired recognition memory, reduced behavioral despair and elevated compulsive behavior. This behavioral "signature" is consistent with an "anti-depressant" function of miR-708-5p, for instance during the manic-like state of BD.

## miR-708-5p directly targets Neuronatin (Nnat), an ER-resident protein involved in calcium homeostasis

As a next step, we explored the mechanisms underlying miR-708-5p-dependent regulation of MD-associated behavior. miRNAs regulate gene expression through binding to mRNAs, predominantly leading to their degradation or inhibition of translation (Bartel, 2018). Therefore, we reasoned that the analysis of the hippocampal transcriptome of miR-708-5p overexpressing mice could inform us about potential miR-708-5p downstream targets mediating behavioral effects. PolyA-RNA-sequencing revealed 118 differentially expressed genes (DEG) (p-value < 0.01; FDR < 0.001) between the hippocampus of hp708 and hpCTL infected mice

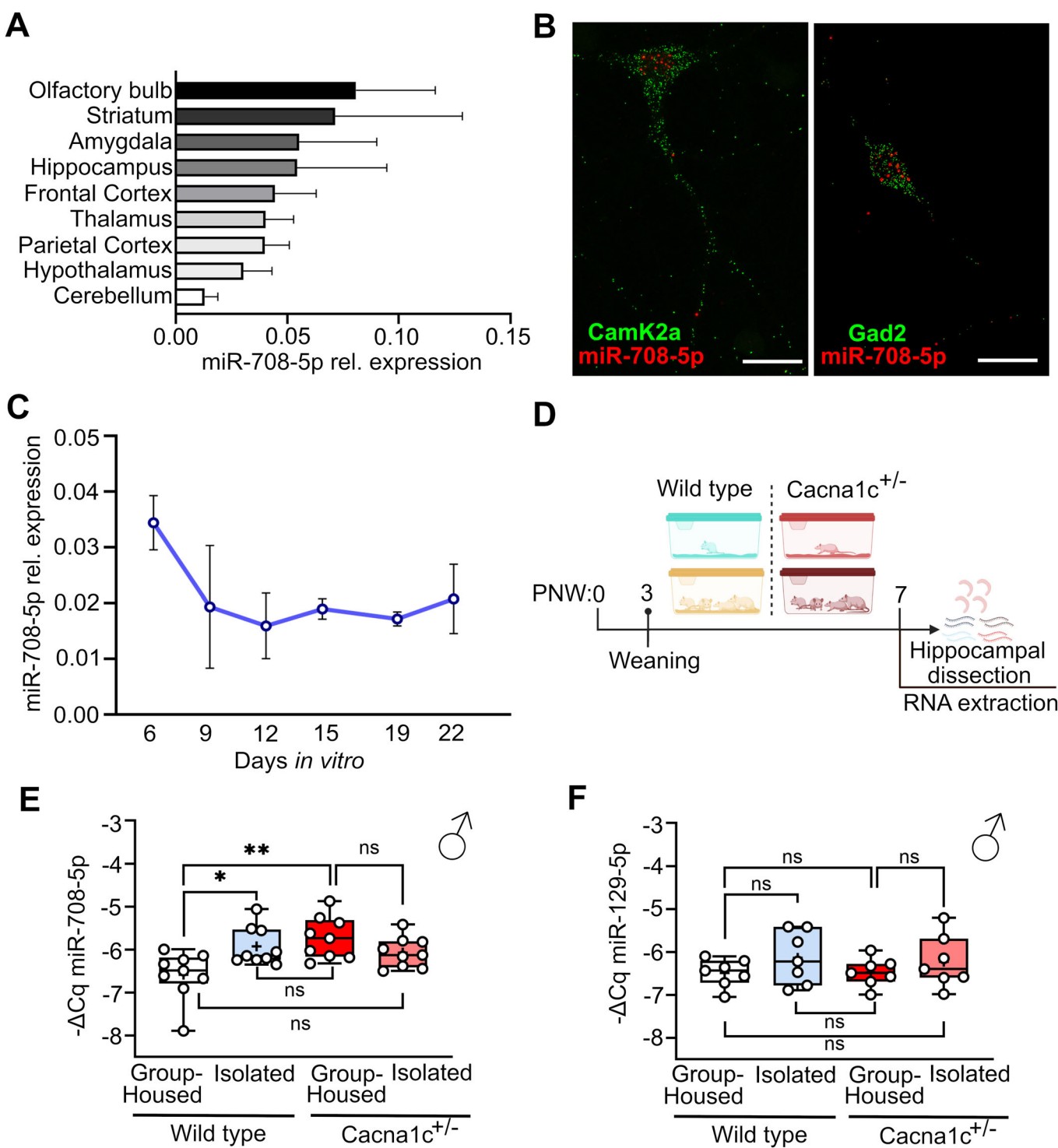

(Fig. 4A). Sixteen DEGs were significantly downregulated and contained predicted miR-708-5p binding sites (Fig. EV3A), making them strong candidates for direct targets of miR-708-5p. *Nnat* was of particular interest, since it has been previously identified as a direct target of miR-708-5p (Vatsa et al, 2019; Yang et al, 2015) and its 3′UTR contains a miR-708-5p perfect binding site (Fig. EV3B). The *Nnat* gene encodes for a small ER-resident protein which acts as an endogenous inhibitor of the SERCA calcium pump, thereby controlling intracellular calcium homeostasis which is disturbed especially in BD (Harrison et al, 2021). Moreover, within the mouse brain, *Nnat* is primarily expressed in glutamatergic CA1 and CA3 neurons based on published single-cell RNA-seq data (Fig. EV3A) (Yao et al, 2021), consistent with a functional interaction of miR-708-5p and *Nnat* in neurons. To further validate our RNA-seq data, we measured the levels of *Nnat* mRNA in the hippocampus of mice upon

**Figure 2.  miR-708-5p is expressed in rat hippocampal neurons and upregulated in the hippocampus of rat models of environmental or genetic risk for MDs.**

(A) miR-708-5p qPCR analysis using total RNA isolated from different regions of the adult female rat brain. Data are represented as bar graphs, mean ± SD ($n = 3$ animals). (B) Single-molecule fluorescence in situ hybridization performed in rat hippocampal neurons at 7 days in vitro using probes directed against miR-708-5p (in red). Probes against CamK2a (to identify excitatory neurons) or Gad2 (to identify inhibitory neurons) were also used (in green). Scale bar: 20 μm. (C) Relative expression of miR-708-5p in primary rat hippocampal neurons at different days in vitro ($n = 3$ biological replicates) normalized to U6 snRNA. Data are represented as mean ± SD. (D) Schematic of the experiment: wild type and $Cacna1c^{+/-}$ juvenile rats were either kept group-housed or socially social isolated upon weaning (3 postnatal weeks, PNW) and then euthanized for tissue collection at PNW 7. (E) miR-708-5p qPCR analysis of total RNA isolated from hippocampi of Wild type (WT) or $Cacna1c^{+/-}$ male juvenile rats that were either group-housed or socially isolated for four weeks ($n = 9$ each group). Linear mixed model, -DeltaCq ~ Genotype * Housing + (1| Cohort), followed by post hoc pairwise comparison with emmeans package (pairwise ~ Genotype * Housing); WT group-housed vs WT Isolated: $*p = 0.0265$; WT group-housed vs $Cacna1c^{+/-}$: $**p = 0.0026$; ns: non-significant. Data are represented as box plot with whiskers and data points (+: mean, line: median; whiskers: minimum and maximum values). (F) miR-129-5p expression was assessed as a control ($n = 7$ each group). Same statistics as (E); WT group-housed vs WT Isolated: ns; WT group-housed vs $Cacna1c^{+/-}$: ns. Data are represented as box plot with whiskers and data points (+: mean, line: median; whiskers: minimum and maximum values). Source data are available online for this figure.

miR-708-5p overexpression and detected a significant decrease compared to control-infected mice (Fig. 4B). So far, our expression analysis on hippocampal tissue did not allow us to distinguish between effects occurring in neuronal and non-neuronal cells. Therefore, we further extended our analysis to rat primary hippocampal neuron cultures. Like in the in vivo experiments, infection of rat neurons with hp708 resulted in a robust upregulation of miR-708-5p levels (Fig. 4C) and led to significant reductions in *Nnat* mRNA (Fig. 4D), and protein levels (Figs. 4E,F and EV3C), demonstrating that miR-708-5p represses endogenous *Nnat* expression in neurons. To prove that the downregulation of *Nnat* is due to a direct interaction of the miR-708-5p and *Nnat* mRNA via the predicted miR-708-5p binding site, we performed luciferase reporter gene assay in primary rat hippocampal neurons. Luciferase reporter genes were either fused to the wild-type 3'UTR of Nnat (Nnat wt) or to a 3'UTR of Nnat containing several point mutations expected to prevent miR-708-5p binding (*Nnat* mt). Whereas *Nnat* wt was efficiently downregulated by co-transfection of hp708-p, *Nnat* mt was insensitive to miR-708-5p overexpression (Fig. 4G, left). The results obtained from these assays therefore confirm that miR-708-5p directly suppresses *Nnat* expression through a 3'UTR interaction. In a complementary approach, we employed a target-directed miRNA degradation (TDMD) construct (sp708) to reduce endogenous miR-708-5p levels (Fig. EV3D,E). Here, we detected a significant increase in luciferase activity upon transfection with sp708 compared to control for *Nnat* wt, but not *Nnat* mt expressing neurons (Fig. 4G, right). In summary, our results from RNA-seq, qPCR and luciferase assays established *Nnat* as a direct target of miR-708-5p in rodent hippocampal neurons.

## Restoring Nnat expression in miR-708-5p overexpressing hippocampal neurons rescues BD-associated behavioral endophenotypes in mice

Building upon our identification of *Nnat* as a potential regulatory target of miR-708-5p in the hippocampus, we hypothesized that *Nnat* downregulation might underlie the observed MD-associated behavioral alterations elicited by miR-708-5p overexpression. To test this hypothesis, we engineered a construct which allows to simultaneously overexpress *Nnat* and miR-708-5p upon viral infection. This was achieved by inserting the coding sequence of *Nnat* downstream of the hSYN promoter, coupled with a P2A self-cleaving peptide; this fragment was followed by the EGFP coding sequence and the overexpressing miR-708-5p hairpin. As a control, we utilized a similar construct where the mCherry coding sequence replaced *Nnat* followed by EGFP and hp708 or EGFP and hpCTL

(Appendix Fig. S3A). rAAV obtained from these constructs was injected into the hippocampus of seven/eight weeks old male mice by stereotactic surgery, followed by a four-week recovery period before behavioral assessments as described in Fig. 3A. Behavioral analyses revealed that miR-708-5p overexpression alone, as shown above, impaired object discrimination ability in the NOR test (Fig. 5A; Appendix Fig. S3B). This was not observed in mice injected with the Nnat-P2A-hp708 construct, which, similar to the control-injected mice, were able to recognize the novel object (Fig. 5A; Appendix Fig. S3B). Furthermore, the total time spent exploring the objects was not affected in any of the conditions, ruling out this confounding factor (Appendix Fig. S3C). Interestingly, *Nnat* co-expression was able to rescue miR-708-5p overexpression-induced reductions in behavioral despair in the TST test (Fig. 5B). This finding was not due to changes in locomotor activity or anxiety, as all groups showed comparable distance walked and time spent in the center and periphery in the OFT (Appendix Fig. S3D,E). After the end of the cohort, qPCRs in hippocampal tissue confirmed the upregulation of miR-708-5p and the intended modulation of *Nnat* levels (Fig. 5C,D; Appendix Fig. S3F,G). Notably, the mCherry-P2A-hp708 construct led to a pronounced upregulation of miR-708-5p (Fig. 5C; Appendix Fig. S3F) and a corresponding decrease in *Nnat* levels, as opposed to the control and the Nnat-P2A-hp708 conditions, the latter of which restored *Nnat* expression to wild-type levels (Fig. 5D; Appendix Fig. S3G). Taken together, our results strongly suggest that *Nnat* is an important downstream target in mediating the effects on MD-associated behavioral endophenotypes caused by miR-708-5p overexpression in the mouse hippocampus.

## miR-708-5p levels negatively correlate with human cognitive processing and represent a potential biomarker for differential diagnosis in MDs

Our functional analysis in mice implies an important role of miR-708-5p in the regulation of MD-associated behaviors. To explore a potential link between miR-708-5p and human behavior, we harnessed our rich dataset of neuropsychological test results present within the FOR2107 cohort (Kircher et al, 2019). Given our previous results from mice (Fig. 3), we were specifically interested in correlations between miR-708-5p expression levels (measured in PBMCs) and performances in assessments covering the neurocognitive domains (attention/concentration; executive function, verbal and visuo-spatial memory (see Methods section). Interestingly, miR-708-5p expression

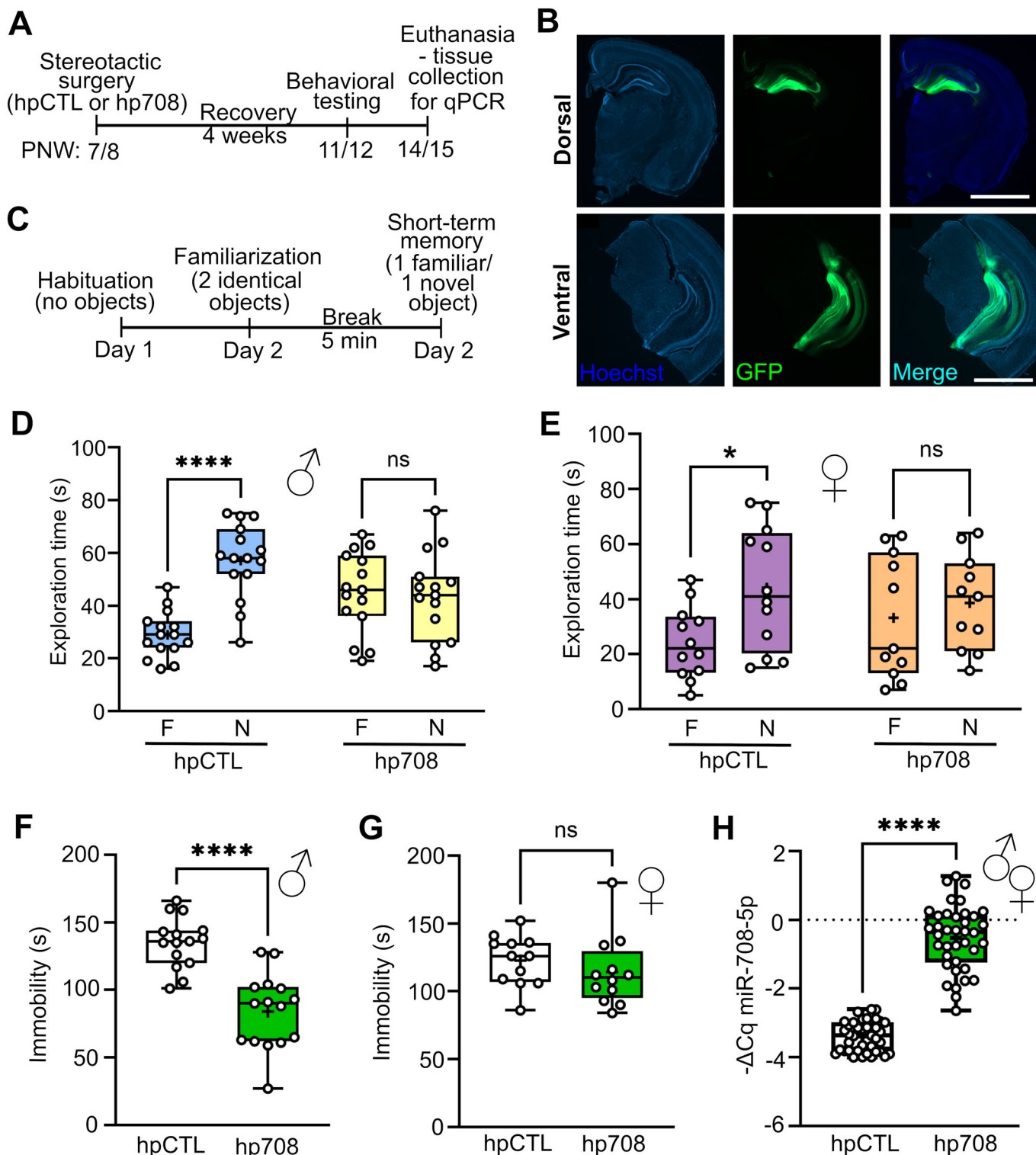

in a combined human sample ($n = 162$; both healthy and MD subjects of both sexes; Table 1) showed a significant negative correlation with the score from the D2 attention test (Fig. 6A), which suggests that high levels of miR-708-5p in human impair selective attention and cognitive processing. Significant negative correlations were also observed when considering males ($n = 90$) and females ($n = 72$) separately and were

mostly driven by BD and to a lesser extent MDD patients (Appendix Fig. S4A–D). These findings align well with our results from behavioral testing in mice, which revealed a negative role for miR-708-5p in novel object recognition memory.

Finally, we assessed the potential of miR-708-5p as a diagnostic biomarker in MD, Therefore, we measured the levels of miR-708-

Figure 3. miR-708-5p overexpression in the mouse hippocampus elicits MD-associated behavioral endophenotypes.

(A) Schematic timeline of the acute rAAV stereotactic injection of control (hpCTL) or miR-708 overexpression (hp708) virus into the dorsal and ventral hippocampus of mice at PNW 7/8, followed by behavioral testing at PNW 11/12. (B) Coronal brain section of mouse brains displaying intense GFP expression upon infection with hp708 virus in the dorsal (up) and ventral (down) hippocampus. Scale bar: 2000 μm. (C) Schematic representation of the Novel Object Recognition (NOR) test. Mice are habituated to the experimental box one day before the experiment. On the familiarization session, mice are exposed to two identical objects for 5 min. After a 5-min break (in which mice are returned to their cage), a short-term memory session takes place, in which mice are exposed to two objects: one object from the previous session, and one new object. Mice are then allowed to explore for 5 min. (D) Time (s) male mice injected with the indicated rAAV (hpCTL or hp708, $n = 15$ each) explored either the familiar (F) or the novel (N) object in the short-term memory session. Data are represented as box plot with whiskers and data points (+: mean, line: median; whiskers: minimum and maximum values). Two-way RM ANOVA: Novelty × Group, ****$p < 0.0001$; Novelty, ****$p < 0.0001$; Group, ns, $p = 0.8207$. Šídák's post hoc test, F vs N: hpCTL, ****$p < 0.0001$; hp708, ns, $p = 0.6975$. (E) Time (s) female mice injected with the indicated rAAV (hpCTL $n = 12$, or hp708, $n = 11$) explored either the familiar (F) or novel (N) object in the short-term memory session. Data are represented as box plot with whiskers and data points (+: mean, line: median; whiskers: minimum and maximum values). Two-way RM ANOVA: Novelty × Group, *$p < 0.0130$; Novelty, ***$p = 0.0001$; Group, ns, $p = 0.8143$. Šídák's post hoc test, F vs N: hpCTL, ****$p < 0.0001$; hp708, ns, $p = 0.3127$. (F) Time (s) male mice injected with the indicated rAAV (hpCTL or hp708, $n = 15$ each) spent immobile in the Tail Suspension Test. Data are represented as box plot with whiskers and data points (+: mean, line: median; whiskers: minimum and maximum values). Unpaired t-test, ****$p < 0.0001$. (G) Time (s) female mice injected with the indicated rAAV (hpCTL $n = 12$, or hp708, $n = 11$) spent immobile in the Tail Suspension Test. Data are represented as box plot with whiskers and data points (+: mean, line: median; whiskers: minimum and maximum values). Unpaired t-test, ns. (H) miR-708-5p qPCR analysis of total RNA isolated from hippocampus of all mice (males and females pooled) that underwent behavioral testing upon miR-708-5p overexpression. hpCTL $n = 39$, hp708 $n = 39$. Unpaired t-test, ****$p < 0.00001$. Data are represented as box plot with whiskers and data points (+: mean, line: median; whiskers: minimum and maximum values). See also: Fig. EV2, Appendix Figs. S1 and S2. Source data are available online for this figure.

5p in PBMCs obtained from MDD ($n = 42$) and BD ($n = 63$) patients (Fig. 6B; Table 1) by qPCR. miR-708-5p was significantly upregulated in female BD ($n = 26$) and MDD ($n = 18$) patients (Fig. 6C) compared to healthy controls (CTL, $n = 26$), after correcting for age and antidepressant treatment (Appendix Fig. S5A,B), whereby the effect was more pronounced in BD compared to MDD patients. A similar picture was observed in male patients, in which the increase was highly significant for BD subjects (Fig. 6D), after age and antidepressant treatment correction (Appendix Fig. S5C), and only trending for patients affected by MDD (Fig. 6D, Appendix Fig. S5D). Furthermore, miR-708-5p was significantly upregulated in both subtypes of BD in both female (Fig. EV4A) and male (Fig. EV4B) subjects. When separating BD samples into BD state sub-groups, the only significant BD state in which miR-708-5p was found to be upregulated in female subjects was mania (Fig. EV4C), although it only included two samples. In male subjects, a significant miR-708-5p upregulation was found in depressive and hypomanic BD states (Fig. EV4D).

We then investigated whether miR-708-5p expression correlated with depressive symptoms in MDD male and female subjects, as assessed with the Beck's depression inventory (BDI), but no significant or trending correlation could be observed (Appendix Fig. S5E,F). Moreover, the Young Mania Rating Scale (YMRS) score, a test used to assess manic-like symptomatology in patients with BD, was used to correlate with miR-708-5p expression levels. While no correlation was found in female BD patients (Appendix Fig. S6A), a trending positive correlation was found in male subjects affected by BD (Appendix Fig. S6B). Taken together, our results suggest that miR-708-5p is elevated in the peripheral blood of MD patients, with a particularly strong upregulation observed in BD males.

To strengthen miR-708-5p as a potential differential diagnostic marker for MD, we assessed whether its expression is changed in PBMCs obtained from patients suffering from a related psychiatric disorder. For that purpose, we chose schizophrenia (SCZ), given the high genetic similarities between SCZ and BD (Craddock et al, 2005). We performed qPCR in PBMCs from female ($n = 8$) and male ($n = 15$) SCZ patients but were unable to detect any significant change in miR-708-5p expression compared to CTL subjects ($n = 19$) (Fig. EV4E), showing that this miRNA is

dysregulated in MD, but not in another related psychiatric disorder.

One of the biggest challenges in mood disorder diagnostics is to distinguish BD and MDD patients, especially when BD patients are in a depressive phase (Nierenberg et al, 2023; Vieta et al, 2018). Based on our results, we demonstrated that miR-708-5p expression in PBMCs could be potentially used as a diagnostic tool to classify samples into BD or control group, and BD or Control and MDD group. To further test this, we performed receiver operating characteristic (ROC) curve analysis (Zweig and Campbell, 1993) and used miR-708-5p expression as predictor variable. We found that miR-708-5p alone performed better in male samples than in female samples when asked to discriminate BD patients from controls and MDD patients (Appendix Fig. S6C,D), consistent with our results above (Fig. 6C,D). We then performed the same analysis using the expression values of miR-499a-5p, which we previously found to be significantly upregulated in male BD subjects compared to healthy controls (Martins et al, 2022). Like miR-708-5p, miR-499a-5p performed better as a classifier in male subjects compared to female subjects (Appendix Fig. S6E,F). Furthermore, we performed another ROC curve analysis considering the expression levels of both miRNAs together and found that the combined expression of miR-708-5p and miR-499a-5p performs better (AUC = 0.914) in classifying male BD vs controls (Fig. 6E) than when the miRNAs are analyzed individually. In addition, the combination performs better (AUC = 0.906) also in classifying male BD vs controls and MDD male patients (Fig. 6F).

Taken together, our results show that the combined expression of miR-708-5p and miR-499a-5p in PBMCs shows a good predictive potential to distinguish BD patients from MDD patients and healthy controls, in particular in male patients.

## Discussion

In this study, we found that miR-708-5p is upregulated in healthy human subjects at high risk to develop MDs, in genetic and environmental rat MD models, as well as in MD patients.

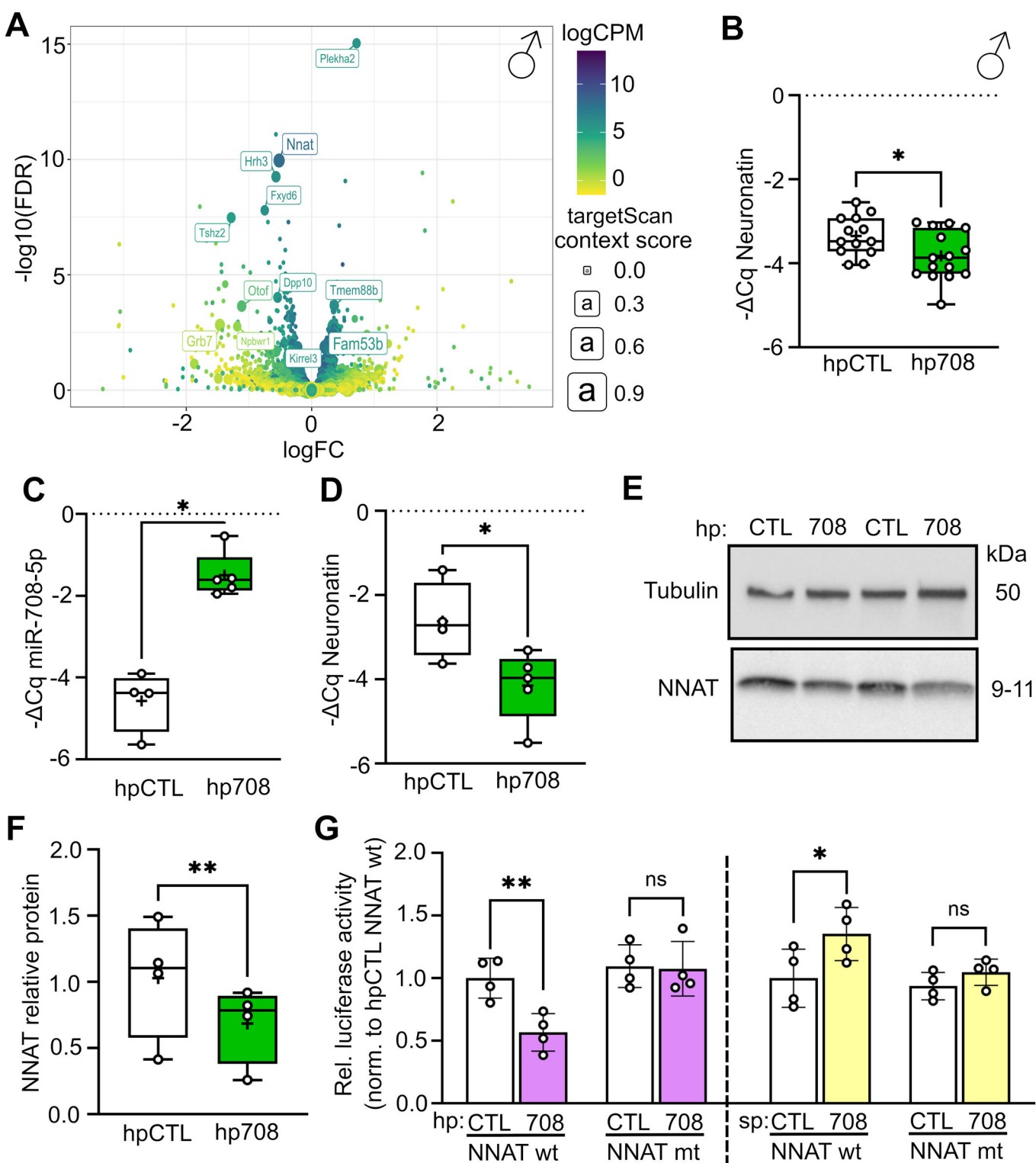

Furthermore, peripheral miR-708-5p expression was negatively correlated with cognitive processing. Mimicking miR-708-5p overexpression in the mouse hippocampus was sufficient to elicit MD-associated behavioral alterations, namely memory impairments, increased compulsive anxiety and reduced depression-like behavior. Finally, peripheral levels of miR-708-5p, together with the previously characterized BD-associated miR-499-5p, efficiently discriminated male BD from MDD patients and healthy subjects, emphasizing its utility as a diagnostic biomarker in MDs.

◄ **Figure 4. miR-708-5p directly targets *Nnat*, an ER-resident protein involved in calcium homeostasis.**

(A) Volcano plot depicting differentially expressed genes in the hippocampus of hp708 *vs* hpCTL injected male mice (*n* = 4 each group) based on polyA-RNA sequencing. Statistics: differential expression analysis (edgeR's quasi-likelihood model). (B) Neuronatin qPCR analysis of total RNA isolated from mouse hippocampi upon miR-708-5p overexpression. hpCTL *n* = 13, hp708 *n* = 15. Unpaired t-test, *$p$ = 0.0269. Data are represented as box plot with whiskers and data points (+: mean, line: median; whiskers: minimum and maximum values). (C) miR-708-5p qPCR analysis of total RNA isolated from rat primary hippocampal neurons (20 days in vitro) infected with hpCTL or hp708 (hpCTL *n* = 4, hp708 *n* = 5 biological replicates). Unpaired t-test, *$p$ = 0.0317. Data are represented as box plot with whiskers and data points (+: mean, line: median; whiskers: minimum and maximum values). (D) Neuronatin qPCR analysis of total RNA isolated from rat primary hippocampal neurons (20 days in vitro) infected with hpCTL or hp708 (hpCTL *n* = 4, hp708 *n* = 5 biological replicates). Unpaired t-test, *$p$ = 0.0159. Data are represented as box plot with whiskers and data points (+: mean, line: median; whiskers: minimum and maximum values). (E) Representative Western blot image of NNAT (lower panel) and Tubulin (upper panel) protein expression levels in primary hippocampal neurons (20 days in vitro) that were infected with hpCTL or hp708 at 2 days in vitro. (F) Quantification of the relative intensity of NNAT protein levels in hippocampal neurons. Tubulin was used as normalizer. Ratio-paired t-test, **$p$ = 0.0042 (*n* = 4 biological replicates per group). Data are represented as box plot with whiskers and data points (+: mean, line: median; whiskers: minimum and maximum values). (G) Relative luciferase activity of rat hippocampal neurons transfected with the indicated plasmid (left: control or 708 overexpression hairpins - hpCTL, hp708; right: control or 708 sponges – spCTL or sp708), and expressing either *Nnat* wild-type (wt) or miR-708-5p binding site mutant (mt) reporter genes. Data are represented as scattered dot plots with bar, mean ± SD (*n* = 4 biological replicates). Two-way ANOVA: main effect of the hairpin *$p$ = 0.0245, of the *Nnat* luciferase reporter **$p$ = 0.0050, and of the hairpin by *Nnat* luciferase reporter interaction $p$ < 0.0374. Sidak's post hoc test: **$p$ = 0.0092. Two-way ANOVA: main effect of the sponge *$p$ = 0.0211, no main effect of the *Nnat* luciferase reporter, or of the hairpin by Nnat luciferase reporter interaction. Sidak's post hoc test: *$p$ = 0.0287. See also: Fig. EV3. Source data are available online for this figure.

## miR-708-5p regulation by genetic and environmental risk factors

MDs are highly heritable psychiatric disorders. Genetic factors seem to explain about 35–45% of variance in the etiology of MDD and 65–70% of variance for BD (Coleman et al, 2020; Polderman et al, 2015). However, genetics alone cannot explain the degree to which MD is inherited in families, suggesting that environmental factors need to be considered as well. Childhood maltreatment is the most studied environmental stressor in the context of MDs (Aas et al, 2020; Nemeroff, 2016). Interestingly miR-708-5p was upregulated in the juvenile social isolation rat model, an animal model for childhood maltreatment (Seffer et al, 2015). Consistent with this result, a recent study reported miR-708-5p upregulation in the PFC upon seven days of chronic social defeat in vulnerable rats compared to resilient animals (Chen et al, 2015). Moreover, here we show that miR-708-5p levels are elevated in *Cacna1c*$^{+/-}$ rats, a well-established genetic model of psychiatric disorders (Bhat et al, 2012), even without environmental stressors. Taken together, these results suggest that genetic and environmental risk might impinge on common pathways triggering miR-708-5p expression. Furthermore, studies in the context of ovarian cancer revealed the presence of a glucocorticoid response element upstream of *ODZ4* (Lin et al, 2015), suggesting that the expression of the host gene and of miR-708-5p might be responsive to glucocorticoids. Moreover, studies in the context of ER stress suggest that miR-708-5p expression is induced by CHOP, a transcription factor involved in the Unfolded Protein Response (UPR) upon ER stress (Behrman et al, 2011). Furthermore, metformin-induced upregulation of miR-708-5p elicits *Nnat* downregulation in prostate cancer, leading to the expression of ER stress mediators. Finally, bisphenol A, a ubiquitous endocrine disruptive chemical, induces miR-708-5p in hypothalamic neurons in a CHOP-dependent manner, accompanied by *Nnat* downregulation (Nierenberg et al, 2023). Thus, we speculate that increased ER stress, which has been implicated in BD, is a critical mediator of miR-708-5p upregulation in response to genetic and environmental stress in neurons. In the future, miR-708-5p manipulation in the context of a GxE rodent model should help to test this hypothesis.

## miR-708-5p function in MD-associated behaviors

Modeling the complex symptomatology of MD in animal models is challenging and has so far only been partially achieved in mice in the context of miRNA manipulation. For example, the acute manipulation of miR-124 in the PFC of mice led to impaired social behavior and locomotor disturbances upon injection of psychostimulants (Namkung et al, 2023). The hippocampal overexpression of miR-499-5p in *Cacna1c*$^{+/-}$ rats caused memory impairments, recapitulating the cognitive deficits encountered in MD (Martins et al, 2022), including deficits in attention, memory and executive functions present during the different mood phases and during remission (Huang et al, 2023). Interestingly, memory impairments are also observed in environmental rodent models of bipolar disorder, such as in the ouabain (Valvassori et al, 2021) and in the amphetamine models (Fries et al, 2015). Moreover, cognitive impairments in BD patients are typically found on tests of attention, working and episodic memory, processing speed and executive function, with significant group differences of medium to large effect size compared to healthy comparison groups (Arts et al, 2008; Bourne et al, 2013; Mann-Wrobel et al, 2011; Robinson et al, 2006). Accordingly, we found that in humans miR-708-5p peripheral expression is negatively correlated with D2 scoring in the attention test (Camelo et al, 2017). Moreover, we found that both female and male mice overexpressing miR-708-5p in the hippocampus showed memory impairments. Furthermore, male, but not female, mice showed significantly decreased immobility in the TST. This test has been historically used to test antidepressant effects in mice but is also employed in mouse models of manic behavior. For example, BD mouse models of mania (Ankyrin-G conditional knockout, Clock19 deficient mice) show reduced immobility during TST (Zhu et al, 2017) and the FST (Roybal et al, 2007), suggesting that the reduced behavioral despair in our model could reflect a function of miR-708-5p in manic-like behavior. This is further supported by increased (although non-significant) marble burying, indicative of a compulsive response to anxiogenic stimuli. However, in our study, the antidepressant-like behavior was neither accompanied by increased exploratory behavior in the

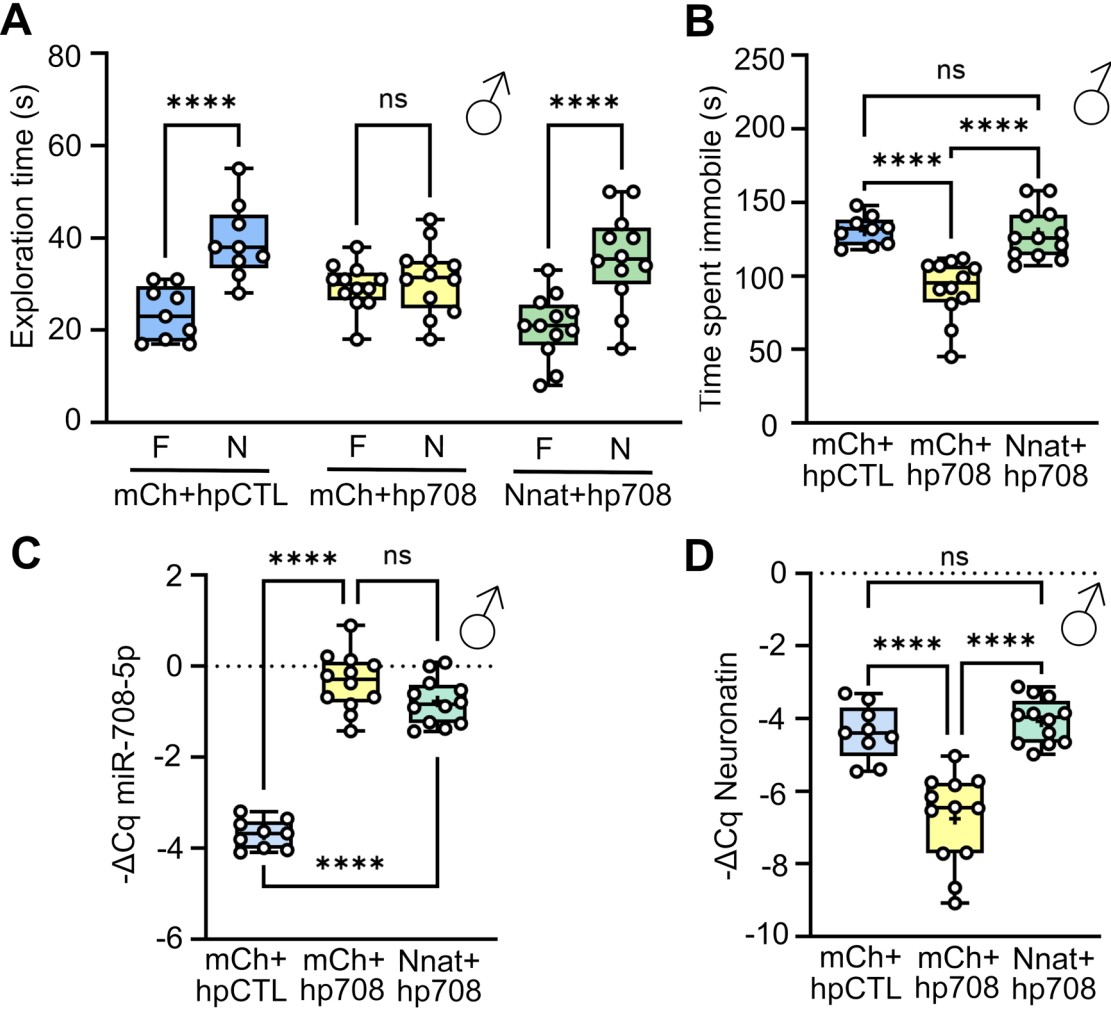

**Figure 5. Restoring *Nnat* expression in miR-708-5p overexpressing hippocampal neurons rescues MD-associated behaviors in mice.**

(A) Novel object recognition Test with 5 min break in between familiarization and short-term memory session. Time (s) male mice injected with the indicated rAAV (mCh +hpCTL $n = 9$, mCh+hp708 $n = 12$, or Nnat+hp708 $n = 12$) explored either the familiar (F) or novel (N) object. Data are represented as box plot with whiskers and data points (+: mean, line: median; whiskers: minimum and maximum values). Two-way RM ANOVA: Novelty × Group, ****$p < 0.0001$; Novelty, ****$p < 0.0001$; Group, ns, $p = 0.5765$. Šídák's post hoc test, F *vs* N: mCh+hpCTL, ****$p < 0.0001$; mCh+hp708, ns, $p = 0.6534$; Nnat+hp708, ****$p < 0.0001$. (B) Time (s) male mice injected with the indicated rAAV (mCh+hpCTL $n = 9$, mCh+hp708 $n = 12$, or Nnat+hp708 $n = 12$) spent immobile in the Tail Suspension Test. One-way ANOVA, Post hoc Tukey's Test: mCh+hpCTL *vs* mCh+hp708: ****$p < 0.0001$, mCh+hpCTL *vs* Nnat+hp708: ns, $p = 0.9617$, mCh+hp708 *vs* Nnat+hp708: ****$p < 0.0001$. Data are represented as box plot with whiskers and data points (+: mean, line: median; whiskers: minimum and maximum values). (C) miR-708-5p qPCR analysis of total RNA isolated from the hippocampus of male mice injected with mCh+hpCTL ($n = 9$), mCh+hp708 ($n = 12$), or Nnat+hp708 ($n = 12$) viruses. One-way ANOVA, Post hoc Tukey's Test: mCh +hpCTL *vs* mCh+hp708: ****$p < 0.0001$, mCh+hpCTL *vs* Nnat+hp708: ***$p < 0.0001$, mCh+hp708 *vs* Nnat+hp708: ns, $p = 0.1225$. Data are represented as box plot with whiskers and data points (+: mean, line: median; whiskers: minimum and maximum values). (D) Neuronatin qPCR analysis of total RNA isolated from the hippocampus of male mice injected with mCh+hpCTL ($n = 9$), mCh+hp708 ($n = 12$), or Nnat+hp708 ($n = 12$) viruses. One-way ANOVA, Post hoc Tukey's Test: mCh+hpCTL *vs* mCh +hp708: ***$p = 0.0003$, mCh+hpCTL *vs* Nnat+hp708: ns, $p = 0.7379$, mCh+hp708 *vs* Nnat+hp708: ****$p < 0.0001$. Data are represented as box plot with whiskers and data points (+: mean, line: median; whiskers: minimum and maximum values). See also: Appendix Fig. S3. Source data are available online for this figure.

OFT, nor increased risk-taking during EPM testing, both of which are usually observed in mouse models of mania. We consider two possible explanations for this discrepancy. First, contrary to most of the models reported in the literature, our animal model is characterized by an acute and selective overexpression of miR-708-5p in the hippocampus. Therefore, we might expect to observe endophenotypes dependent on the hippocampus (e.g., antidepressant-like behavior, cognition), but not those related to other brain regions, such as the amygdala, PFC, or cerebellum (e.g., risk-taking behavior, hyperactivity). Alternatively, the miR-708-5p/Nnat pathway might selectively control specific aspects of manic- and depression-like behavior. To distinguish between these possibilities, it will be important to study the role of miR-708-5p overexpression in other brain areas relevant for MD-associated behaviors in the future, in particular the PFC.

**Table 1. Data from healthy control, bipolar disorder and major depressive disorder subjects.**

|  | Control | BD | MDD | P Value |
|---|---|---|---|---|
| n (F/M) | 26/31 | 26/37 | 18/24 | N/A |
| Age ± S.D. | 29.4 ± 5.4/ 39.7 ± 13.4 | 29.9 ± 5.4/ 41.7 ± 11.3 | 29.6 ± 4.9/ 39.3.6 ± 14.7 | 0.94/ 0.75 |
| CTQ ± S.D. (%) | 40.7 ± 11.4 (53.8%) | 42.3 ± 13.2 (30.8%) | 50.2 ± 20.2 (61%) | 0.1067/ 0.4170 |
| Family history of MDs (%) | 8 (30.8%)/ 5 (16.1%) | 10 (38.5%)/ 10 (27.0%) | 5 (27.8%)/ 7 (29.1%) | N/A |
| HAMD ± S.D. | 2.7 ± 3.9/ 1.5 ± 2.5 | 7.5 ± 6.2/ 8.6 ± 6.4 | 14.8 ± 6.7/ 10.3 ± 7.7 | <0.0001 |
| YMRS ± S.D. | 0.5 ± 1.1/ 0.7 ± 1.7 | 2.4 ± 2.9/ 7.0 ± 8.2 | 1.4 ± 1.6/ 1.7 ± 2.2 | <0.01/ <0.0001 |
| BDI ± S.D. | 6.1 ± 6.1/ 4.5 ± 3.9 | 12.9 ± 11.1/ 12.9 ± 9.6 | 26.6 ± 11.0/ 16.6 ± 10.7 | <0.0001 |
| AD (%) | 0 (0%) | 7 (26.9%)/ 17 (45.95%) | 16 (88.9%)/ 17 (70.83%) | N/A |
| Antipsychotic (%) | 0 (0%) | 12 (46.15%)/ 19 (51.35%) | 7 (38.89%)/ 4 (16.67%) | N/A |
| Lithium (%) | 0 (0%) | 4 (15.38%)/ 12 (32.43%) | 1 (5.56%)/ 0 (0%) | N/A |
| Anticonvulsive (%) | 0 (0%) | 6 (23.08%)/ 10 (27.03%) | 1 (5.56%)/ 1 (4.17%) | N/A |
| Stimulants (%) | 0 (0%) | 0 (0%)/ 1 (2.70%) | 0 (0%)/ 2 (8.33%) | N/A |
| Benzodiazepine (%) | 0 (0%) | 0 (0%) | 1 (5.56%)/ 0 (0%) | N/A |
| Z substance (%) | 0 (0%) | 0 (0%)/ 1 (2.70%) | 1 (5.56%) | N/A |

Subjects (F = females, M = males) for miR-708-5p expression analysis on psychiatrically healthy controls (Control), Bipolar disorder patients (BD) or Major Depressive Disorder patients (MDD). One-way-ANOVA was performed to evaluate significant differences between groups.
*SD* standard deviation, *CTQ* Childhood Maltreatment Questionnaire, *HAMD* Hamilton Depression Rating Scale, *YMRS* Young Mania Rating Scale, *BDI* Beck's Depression Inventory, *AD* antidepressant use.

## Molecular and cellular mechanisms downstream of the miR-708-5p/Nnat interaction

Dysregulated calcium homeostasis has been previously implicated in the pathophysiology of MD, with a special emphasis on BD (Harrison et al, 2021). Our results from this and a previous study (Martins et al, 2022) are consistent with disrupted calcium flux via L-type calcium channels, e.g., due to *Cacna1c* mutation, as a major underlying cause. However, ER calcium dynamics emerges as another major player. For example, store operated calcium entry (SOCE) is dysregulated in BD-induced pluripotent stem cells, leading to earlier neuronal differentiation and abnormal neurite outgrowth (Hewitt et al, 2023). Furthermore, altered expression of the miR-708-5p target Nnat has been associated with defective intracellular and ER calcium levels (Sharma et al, 2013; Vatsa et al, 2019; Zou et al, 2023). NNAT is a small ER membrane protein which acts as an antagonist of the sarco-endoplasmic reticulum calcium ATPase (SERCA) pump, thereby interfering with calcium

re-uptake into the ER (Braun et al, 2021). Consistently, it has been reported that miR-708-5p-mediated downregulation of *Nnat* causes lower basal cytoplasmic calcium levels (Vatsa et al, 2019). On the other hand, excessive ER calcium re-uptake could result in ER calcium overload, which in turn leads to calcium leakage to the cytoplasm and mitochondria (Daverkausen-Fischer and Prols, 2022), with negative consequences for calcium signaling and cell health, as exemplified in Alzheimer's disease (Bezprozvanny and Mattson, 2008). Clearly, further studies are warranted on the impact of aberrant miR-708-5p/Nnat signaling on intraneuronal calcium homeostasis in the context of MD.

## miR-708-5p as a potential biomarker in BD

Our findings indicate an upregulation of miR-708-5p in PBMCs of females at risk of MDs and in patients with MDD and BD. How altered miR-708-5p levels in PBMCs are linked to human brain states is currently unknown, but one potential route could involve the release of miR-708 encapsulated in extracellular vesicles (EVs) from neurons into the bloodstream. While the crossing of EVs through the BBB has been reported in the literature (Shi et al, 2019), whether EVs can be taken up by PBMCs is not yet fully understood.

A previous study reported a downregulated expression of miR-708-5p in leukocytes of women in depressive state compared to women in remission (Banach et al, 2017). These results are in line with our detected expression difference between female euthymic patients and depressive patients. For male samples, on the other hand, our data suggest a role of miR-708-5p specifically in the manic phase of the disorder, as indicated from the YMRS correlation and the behavioral characterization. Although SCZ and BD were reported to have the highest genetic correlation among psychiatric disorders (Craddock et al, 2005), we found that miR-708-5p expression was unchanged in peripheral samples of patients diagnosed with SCZ. This suggests a rather specific expression pattern of miR-708-5p in the spectrum of MDs. Within MDs, miR-708-5p upregulation was most pronounced in BD. This is impressively illustrated by our ROC curve analysis, which shows that miR-708-5p expression, when combined with miR-499-5p, effectively distinguishes BD patients not only from healthy controls, but also from MDD patients. Although miR-708-5p is significantly upregulated in both male and female BD patient samples, the degree of increase is higher in males compared to females (8.8-fold vs. 4.23-fold change). Moreover, based on our ROC curve analysis, miR-708-5p expression discriminated male BD patients better from control and MDD compared to female patients. Together, these observations suggest a potential sex-specific role of miR-708-5p in BD, which also aligns with our results from mouse behavior. Sex differences in BD manifest across various aspects, ranging from clinical symptoms to the progression of the disorder. Males typically encounter their first manic episode at a younger age compared to females, who are more likely to experience a depressive episode at the onset of BD (Kennedy et al, 2005). In contrast, females experience a higher incidence of depressive episodes and hypomania, leading to a more frequent diagnosis of BD type II (Diflorio and Jones, 2010). Females are also more susceptible to mixed

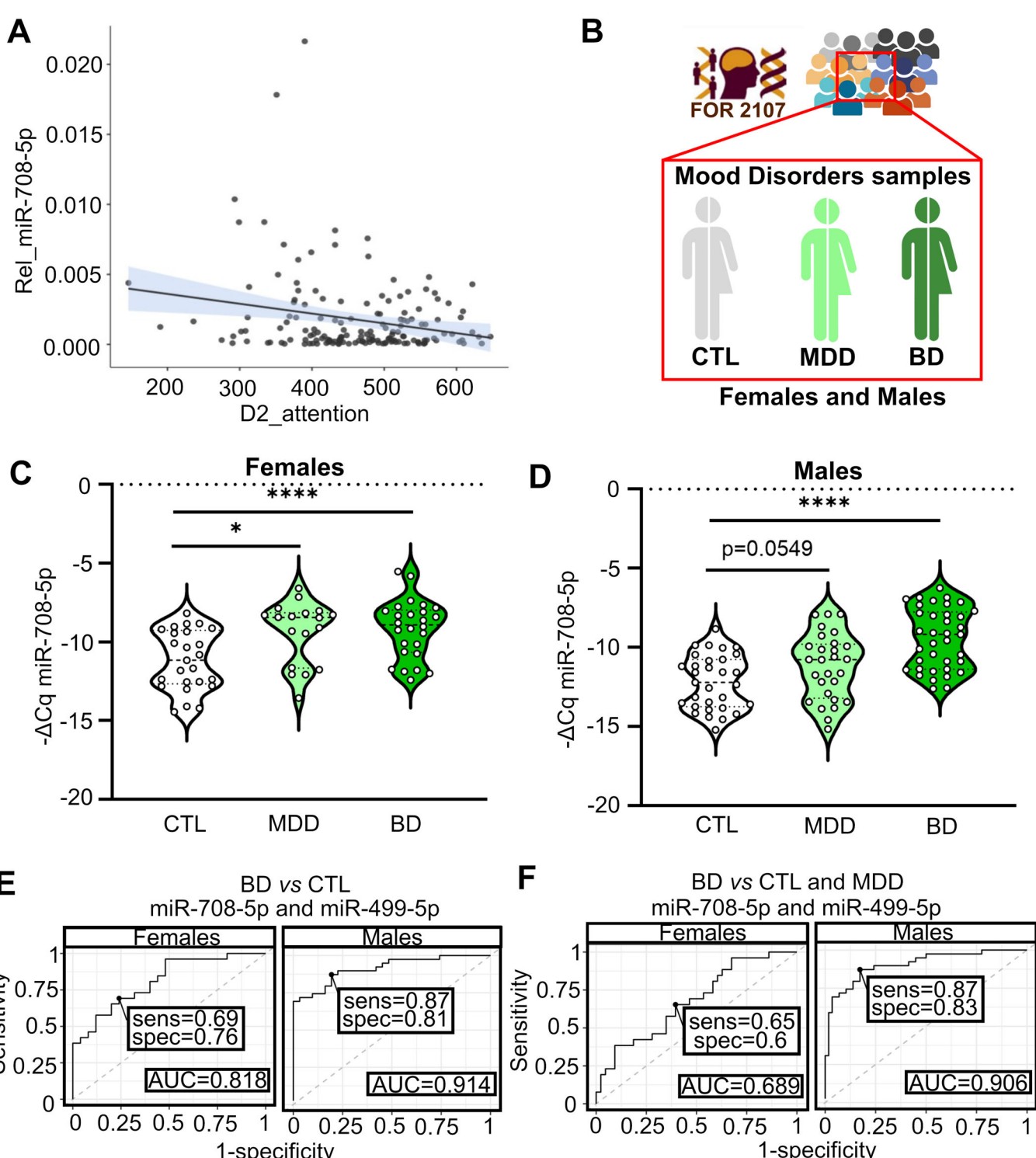

episodes (Arnold et al, 2000), rapid cycling of mood phases (Tondo and Baldessarini, 1998), and show a greater likelihood of attempting suicide (Clements et al, 2013). Thus, it is tempting to speculate that miR-708-5p might play an important role in male-specific aspects of BD, e.g., the development of more common and intense manic episodes. In this regard, miRNAs have been previously implicated in the sexually dimorphic control

of circadian, cholinergic, and neurokine pathways in BD (Lobentanzer et al, 2019).

Taken together, we propose that miR-708-5p represents a promising candidate for the development of a biomarker that helps to stratify MD patients based on disease entity, specific phases of the disease and sex, which should greatly help with diagnosis and therapy.

**Figure 6.  miR-708-5p levels negatively correlate with human cognitive processing and represent a potential biomarker for differential diagnosis in MDs.**

(A) Pearson correlation plot of miR-708-5p peripheral levels and attention (measured by the d2 test) in all human participants (female and male, CTL + MDD + BD pooled, $n = 162$), **$p = 0.005$, $r = -0.223$. (B) Blood PBMCs from male and female patients diagnosed with Major Depressive Disorder (MDD) or Bipolar Disorder (BD) (Mood Disorder samples) were obtained from the FOR2107 cohort, and RNA was extracted, as described in Fig. 1A. (C) miR-708-5p qPCR analysis of total RNA isolated from PBMCs of female patients diagnosed with BD or MDD (CTL $n = 26$, MDD $n = 18$, BD $n = 26$). Wilcoxon rank-sum test after correction for age and antidepressant treatment; linear model of the form -DeltaCq ~ Group + Age + Antidepressant treatment. CTL vs MDD, *$p < 0.05$; CTL vs BD, ****$p < 0.0005$. Data are presented as violin plots with median, quartiles and data points. (D) miR-708-5p qPCR analysis of total RNA isolated from PBMCs of male patients diagnosed with BD or MDD (CTL $n = 31$, MDD $n = 24$, BD $n = 37$). Wilcoxon rank-sum test after correction for age and antidepressant treatment; linear model of the form -DeltaCq ~ Group + Age + Antidepressant treatment. CTL vs MDD, $p = 0.0549$; CTL vs BD, ****$p < 0.0005$. Data are presented as violin plots with median, quartiles and data points. (E) ROC curve of combined miR-708-5p and miR-499-5p expression in female (left) and male (right) subjects to discriminate BD vs control samples. The indicated thresholds are the closest point to the optimal (i.e. top-left). (F) ROC curve of combined miR-708-5p and miR-499-5p expression in female (left) and male (right) subjects to discriminate BD vs control and MDD samples. The indicated thresholds are the closest point to the optimal (i.e. top-left). See also: Fig. EV4, Appendix Figs. S4, S5 and S6. Source data are available online for this figure.

# Methods

### Reagents and tools table

| Reagent/Resource | Reference or Source | Identifier or Catalog Number |
|---|---|---|
| **Experimental models** | | |
| C57BL/6 mice | Janvier Laboratories (France) | *Mus musculus* |
| Primary hippocampal and cortical neurons from embryonic rats | Janvier Laboratories (France) | *Rattus norvegicus* |
| Human PBMCs | This manuscript | *Homo sapiens* |
| **Recombinant DNA** | | |
| rAAV-hSyn-EGFP | Addgene | *#114213* |
| pmirGLO dual-luciferase expression vector | Promega | E1330 |
| **Oligonucleotides and other sequence-based reagents** | | |
| miR-708-5p Taqman qPCR probe | ThermoFisher | 4427975 |
| miR-16-5p Taqman qPCR probe | ThermoFisher | 4427975 |
| miR-30a-5p Taqman qPCR probe | ThermoFisher | 4427975 |
| miR-1248-5p Taqman qPCR probe | ThermoFisher | 4427975 |
| miR-129-5p Taqman qPCR probe | ThermoFisher | 4427975 |
| U6 snRNA Taqman qPCR probe | ThermoFisher | 4427975/001973 |
| For primers used for cloning, see Appendix Table S2 | This paper | N/A |
| **Chemicals, Enzymes and other reagents** | | |
| TRIzol TM Reagent | Thermo Fisher | 15596026 |
| Glycogen | Ambion | AM9510 |
| **Software** | | |
| Prism 10 | GraphPad | https://www.graphpad.com |

| Reagent/Resource | Reference or Source | Identifier or Catalog Number |
|---|---|---|
| Zen Microscopy Software | Zeiss | https://www.zeiss.com/microscopy/en/products/software/zeiss-zen.html |
| edgeR | Bioconductor | http://bioconductor.org/packages/edgeR/ |
| Fiji ImageJ | NIH | https://imagej.net/ |
| Inkscape | Inkscape | https://inkscape.org/ |
| Image Lab 6 | Bio-Rad | http://www.bio-rad.com/en-us/sku/1709690-image-lab-software |
| CFX Maestro | Bio-Rad | https://www.bio-rad.com/en-ch/product/cfx-maestro-software-for-cfx-real-time-pcr-instruments |
| Endnote 21 | Clarivate Analytics | http://endnote.com |
| **Other** | | |
| Lipofectamine 2000 | Thermo Fisher | 11668019 |
| LeukoLOCK | Thermo Fisher | AM1923 |
| *mir*Vana™ | Thermo Fisher | AM1560 |
| QuantiGene ViewRNA miRNA Cell Assay Kit | Thermo Fisher | QVCM0001 |
| Phusion Hot Start Polymerase | Thermo Fisher | F549S |
| Tissue-Tek O.C.T Compound | Sakura Finetek Europe | 4583 |
| Aqua Poly Mount | Chemie Brunschwig | POL18606-20 |
| Complete Protease Inhibitor Cocktail EDTA-free | Sigma-Aldrich | P8340 |
| 4x Laemmli Sample Buffer | Bio-Rad | 1610747 |
| Mini-PROTEAN® TGX™ Precast Protein Gel | Bio-Rad | 4561094 |
| Clarity™ Western ECL Substrate | Bio-Rad | 1705060 |

| Reagent/Resource | Reference or Source | Identifier or Catalog Number |
|---|---|---|
| TURBO DNase enzyme | Thermo Fisher | AM2238 |
| iScript cDNA synthesis kit | Bio-Rad | 1708891 |
| iTaq Universal SYBR Green Supermix | Bio-Rad | 1725121 |
| TaqMan MicroRNA Reverse Transcription Kit | Thermo Fisher | 4366597 |
| TaqMan Universal PCR Master Mix | Thermo Fisher | 4304437 |
| MidiPrep Kit | Macherey Nagel | 740410 |
| Passive lysis buffer (luciferase) | Promega | E194A |
| Aqua-Poly/Mount | Chemie Brunschwig | POL18606-20 |
| Trans-Blot Turbo system | Bio-Rad | 1704158 |
| Clarity™ Western ECL Substrate | Bio-Rad | 1705060 |
| DEPC Nuclease-free H₂O | Ambion | AM9906 |

## Methods and protocols

### Human study

**Recruitment of participants:** The study involved participants with BD (26 females, 37 males), MDD (18 females, 24 males), and healthy individuals (26 females, 31 males), including psychiatrically healthy subjects that had a history of childhood maltreatment ($n = 17$) or a genetic predisposition to MDs ($n = 18$) or no risk ($n = 18$). These participants were recruited from the University of Marburg and the University of Münster in Germany, as part of the FOR2107 cohort (Kircher et al, 2019). Diagnoses were made using the SCID-I interview (Wittchen et al, 1997), and adapted to DSM-IV criteria, excluding those with substance abuse, severe neurological, or other significant medical conditions. Healthy controls were screened similarly, with inclusion in the maltreatment study if at least one of the subclasses of the Childhood Trauma Questionnaire (CTQ) reached the maltreatment threshold (Emotional Abuse ≥ 10, Physical Abuse ≥ 8, Sexual Abuse ≥ 8, Emotional Neglect ≥ 15, and Physical Neglect ≥ 8) (Walker et al, 1999). All participants underwent a comprehensive neuropsychological test battery, including the d2 test of attention (Brickenkamp, 2022). All necessary ethical approvals were obtained (ethics committees of the Medical Faculties of the Universities of Münster (2014-422-b-S) and Marburg (AZ: 07/14)), and participants consented to the study, which complied with the Declaration of Helsinki and the Belmont Report. Demographic and clinical data are outlined in Table 1 and in Appendix Table S1.

**Human peripheral blood mononuclear cell (PBMC) sample processing:** PBMCs were obtained from 10 mL of whole blood using the LeukoLOCK technology (Thermo Scientific) at the Biomaterialbank Marburg, Germany. After sample randomization, total RNA extraction from PBMCs was carried out using the mirVana™ kit (Thermo Fisher) or TRIzol™ Reagent (Thermo Fisher) following the manufacturer's protocol.

**PBMCs processing for small RNA-sequencing:** For the Small RNA-sequencing, RNA was extracted with mirVana™ kit (Thermo Fisher) and DNase treated, and the sequencing was carried out at the Functional Genomic Center Zurich (FGCZ, https://fgcz.ch/).

**Small RNA-analysis:** Short RNA reads were processed using the ncPro 1.6.4 pipeline (Chen et al, 2012) using the hg19 annotated (based on miRBase version 21). Only mature miRNA reads (accepting +2 bp on either end) were considered for downstream analysis. Differential expression analysis was then performed using limma/voom 3.46.0 (Ritchie et al, 2015), with a model including a covariate to correct for the two technical batches.

### Rodent studies—rat primary neuron cultures and in vivo

**Rat in vivo study:** All animal experiments on rats were conducted in accordance with the National Institutes of Health Guidelines for the Care and Use of Laboratory Animals and were subject to prior authorization by the local government (MR 20/35 Nr. 19/2014 and G48/2019; Tierschutzbehörde, Regierungspräsidium Giessen, Germany). Constitutive heterozygous $Cacna1c^{+/-}$ animal breeding and the juvenile social isolation paradigm were conducted as described (Martins et al, 2022).

**Rat primary neuron cultures:** In vitro experiments were approved by the local cantonal authorities (ZH027/21). In short, primary neuronal cultures were prepared from hippocampus or cortex of embryonic day 18 Sprague-Dawley rat embryos (Janvier Laboratories), as described previously (Martins et al, 2022; Narayanan et al, 2024).

**Primary neuron transfection:** Primary hippocampal neurons were transfected with Lipofectamine™ 2000 Reagent (Thermo Fisher) and plasmid DNA constructs as previously described (Martins et al, 2022; Narayanan et al, 2024).

**Primary neuron viral infection:** Primary hippocampal neurons were infected with recombinant adeno-associated virus (rAAV) containing hpCTL or hp708 constructs on the second day in vitro, by adding 300 μL of a mixture of the virus with 300 μL of NBP+ per well on a 24-well plate. Cells were used for downstream analysis on the 20th day in vitro.

**Luciferase assay:** Hippocampal neurons at 6 days in vitro were transfected with 100 ng of the Nnat-3'UTR luciferase reporters and 500 ng of hpCTL or hp708, spCTL or sp708. After 7 days from transfection, the cells were lysed, luciferase assay was conducted using a GloMax R96 Microplate Luminometer (Promega), as previously described (Martins et al, 2022). The relative luciferase activity was determined by calculating the ratio of the Firefly signal to the Renilla signal.

**In situ hybridization:** Single-molecule fluorescence in situ hybridization (smFISH) for miRNA detection in hippocampal neuron cultures was conducted using the QuantiGene ViewRNA miRNA Cell Assay Kit (Thermo Fisher) following the manufacturer protocol with minor adjustments. Probes for hsa-miR-708-5p (Alexa Fluor 546, Thermo Fisher), Nnat (Alexa Fluor 546), CamK2 (Alexa Fluor 488) and Gad2 (Alexa Fluor 488) were used for the assay.

**Plasmid design and rAAVs preparation:** For rAAV-mediated overexpression of miR-708-5p, the chimeric miR-708 hairpin was generated by polynucleotide cloning. A detailed description and characterization of this system have been published (Christensen et al, 2010). To knockdown miR-708-5p, a sponge plasmid was constructed by inserting six TDMD sites predicted to bind miR-

708-5p in the 3'UTR of eGFP on rAAV-hSyn-EGFP. The TDMD prediction was done on the webtool ScanMIR (https://ethz-ins.org/scanMiR/). To perform luciferase activity analysis, the wild type and mutated *Nnat* 3'UTR was inserted into a pmirGLO dual-luciferase expression vector (Promega). Viral vectors were produced by the Viral Vector Facility (VVF) of the Neuroscience Center Zurich (https://www.vvf.uzh.ch). Oligos used for cloning are listed in Appendix Table S2.

### Rodent studies—mouse in vivo

**Husbandry and housing:** All animal experiments on mice were conducted in accordance with Switzerland's animal protection laws and received approval from local cantonal authorities (ZH194/21). Mice were housed collectively in cages designed for 2 to 4 individuals, with unrestricted access to food and water. The animal facility maintained an inverted light-dark cycle of 12 h, with behavioral assessments conducted during the dark phase.

**Stereotactic surgeries and post-operative care:** Stereotactic brain injections were conducted on 2-month-old C57BL/6JRj wild-type mice. The mice were anesthetized with 5% isoflurane in oxygen (1 L/min) and positioned on a stereotactic frame. After anesthesia induction and before the surgical incision, mice were transferred to a heated plate and administered 1.5–2% isoflurane in 200 mL/min oxygen through a mouth/nose mask. To ensure their well-being, animals received a subcutaneous injection of 5 mg/kg Meloxicam for analgesia, and vitamin A was applied to prevent eye dryness. Subsequently, the heads were shaved, cleaned, and an incision was made. Bregma and lambda were identified, and bilateral injections were performed at specific coordinates from bregma: for dorsal hippocampus, the coordinates used were AP: −2.1 mm; ML: ±1.5 mm; DV: −1.7 mm and for ventral hippocampus, the coordinates used were AP: −3.3 mm; ML: ±2.7 mm; DV: 3.7 mm. Each rAAV virus was injected in a volume of 1 µL at each injection site using a thin capillary, with an infusion time of 1 min. After a 2-min period for virus diffusion, the capillary was slowly removed. Local analgesia was achieved by suturing the skin and applying Lidocain and Bupivacain drops (2 mg/kg each) at the wound site. The mice were given time to recover from the surgery before being returned to group housing. A second subcutaneous injection of Meloxicam (5 mg/kg in saline) was administered 12 h post-surgery, and paracetamol was added to their drinking water for the subsequent 48 h. Postoperative health checks were conducted over the three days following the surgery.

**Behavioral testing:** Prior to experimentation, mice underwent daily handling lasting 5 min each over one week. On the day of each test, mice were individually housed and given a 20-min acclimatization period in a holding cage before the task. After the test, mice were then returned to their original group housing with littermates. The equipment was cleaned between trials using a detergent solution (10 ml/L Dr. Schnell AG). Behavioral assays were administered in a sequential order, progressing from the least to the most stressful. Tests were either analyzed automatically or assessed by an experimenter who was blinded to the experimental groups.

**Open field test:** Mice were placed in an open field box with dim yellow light and white noise (size: L 45 × W 45 × H 40 cm, TSE System, Bad Homburg, Germany), and left to explore it for 60 min. The experiment was recorded, and the TSE VideoMot2 analyzer software (TSE Systems, Bad Homburg, Germany) was used for the automated analysis of the distance traveled and time spent in the center of the field.

**Elevated plus maze test:** The mouse was placed in the center of the elevated plus maze, a cross-shaped apparatus with two open arms and two closed arms, each arm measuring L 65 × W 5.5 cm and elevated 62 cm from the ground illuminated by 20 lux white light. Behavior was recorded on video for 5 min and analysis was automated using Noldus Ethovision.

**Saccharin preference test:** Mice were habituated in their standard group housing to the presence of two bottles for 48 h before the test. The test was performed by single housing the test animals, and two bottles of tap water or 0.1% Saccharin solution were given to them for 48 h. The weight of the bottles was recorded at the beginning of the test and every 12 h around the light change (09:00–09:15 am/pm). The position of the bottle was exchanged each time to avoid side preference. The animals were re-grouped after the test ended.

**Novel object recognition (NOR) test:** The test was performed as previously described (Daswani et al, 2022), with slight modifications: 1. Twenty-four hours prior to testing, the animals were habituated to the arena used for the test; 2. The break between familiarization and novelty introduction rounds was extended to 5 min (short-term memory) or 24 h (long-term memory). Animals of two independent cohorts were used for the short-term and long-term novel object recognition memory tests to rule out that the results were confounded due to repeated exposure to the objects.

**Marble burying test:** The test was conducted as previously described (Levone et al, 2021). Shortly, mice were placed on a large box with 5 cm-deep bedding and with 20 glass marbles symmetrically placed and left undisturbed for 30 min. At the end of the test, the number of marbles more than two-thirds buried were counted.

**Passive avoidance test:** The test was conducted using the Passive Avoidance 2-Compartment light-dark arena of the TSE System, with white light set to 500 lux. On day one, the animal was placed into the light compartment while the door to the dark one was open and it was allowed to explore; once it passed to the dark compartment, the door closed, and the animal received a 2-s foot shock of 0.3 mA after three seconds. On day two, the animal was placed again into the light compartment, and, after 15 s, the door was automatically opened. The latency to enter the dark compartment was measured.

**Tail suspension test:** The mouse was hang by its tail for a duration of 6 min using a tape affixed to a suspension metal bar, elevated approximately 50 cm from the ground. Before initiating the experiment, a climb-stopper, following a published procedure (Can et al, 2012), was placed at the mouse tail base to prevent climbing. The sessions were recorded on video, and the duration of immobility was subsequently quantified.

**Tissue collection and histology:** At the end of the experimental cohorts, animals were sacrificed by cervical dislocation and one hippocampus was collected on an ice-cold glass plate and subsequentially snap-frozen for RNA extraction (Trizol protocol) and gene expression analysis; one brain hemisphere was put in a 4% paraformaldehyde solution overnight, then cryoprotected through 30% sucrose solution, frozen at −80 °C and coronally sectioned (50 µm thick) at a cryostat for histological assessments. Coronal sections were stained with Hoechst 33342 (1:2000) for 5 min, mounted on glass slides (Menzel-Gläser Superfrost+), air-dried,

and mounted with Aqua Poly Mount medium (Chemie Brunsch-wig). Fourteen-bit grayscale images of GFP and Hoechst were acquired using a widefield microscope (Axio ObserverZ1/7, Zeiss) with tile scan, 5x objective, a pixel size of 1.17 μm, and an image size dependent on the section's size.

### Other molecular protocols

**RT-qPCR:** Total RNA extraction from rodent brain tissue, PBMC samples, and primary hippocampal cultures was performed using TRIzol™ Reagent (Thermo Fisher), and total RNA was subsequently extracted following the manufacturer's guidelines. The RNA was subsequentially treated with TURBO DNase enzyme (Thermo Fisher). For miRNA detection, the RNA was reverse transcribed with TaqMan MicroRNA Reverse Transcription Kit (Thermo Fisher) and the TaqMan Universal PCR Master Mix (Thermo Fisher) was used to perform RT-qPCR, according to the manufacturer's instructions. For the analysis of mRNA, RNA was reverse transcribed with the iScript cDNA synthesis kit (Bio-Rad), and the iTaq SYBR Green Supermix with ROX (Bio-Rad) was used to perform RT-qPCR on the CFX384 Real-Time System (Bio-Rad).

**PolyA-RNA sequencing sample preparation:** Male mice underwent stereotactic surgery for the injection of the rAAVs hpCTL or hp708 ($n = 4$ per group), as described above. Four weeks after surgery, mice were euthanized, the whole hippocampi were dissected and RNA extracted with the TRIzol™ Reagent (Thermo Fisher). The RNA was then treated with Turbo DNase enzyme. 600 ng of DNase-treated RNA was used for High Throughput Transcriptome sequencing. Libraries were prepared with Illumina TrueSeq mRNA protocol, and the transcriptome sequencing was run on Illumina Novaseq 6000. The transcriptome sequencing was carried out by the Functional Genomic Center Zurich (FGCZ: https://fgcz.ch/).

**PolyA-RNA sequencing analysis:** Reads were mapped to the GRCm39 genome with STAR 2.7.8a (Dobin et al, 2013) using the GENCODE M26 annotation as reference and quantified at the gene-level using featureCounts 1.6.4 (Liao et al, 2014). Genes were filtered using edgeR's filterByExpr function before differential expression analysis with edgeR 3.32.1 (Robinson et al, 2010) using likelihood ratio tests with 3 surrogate variables estimated using sva 3.38.0 (Leek et al, 2012). TargetScan 7 (Agarwal et al, 2015) was used to predict miRNA targets. For gene expression across different brain cell types (Fig. EV2A), the Allen 10X+smartSeq taxonomy (Yao et al, 2021) and per-cluster expression was used, aggregating non-neuronal cell types into broader classes.

**Western blot:** Proteins from primary hippocampal neurons were isolated using ice cold RIPA lysis buffer (10 mM Tris, pH 7.4, 150 mM NaCl, 10 mM EDTA, 2.5 mM EGTA, 1% Triton X-100, 0.1% SDS, 1% sodium deoxycholate, 10 mM NaF, 5 mM $Na_4P_2O_7$, 0.1 mM $Na_3VO_4$), supplemented with Complete Protease Inhibitor Cocktail EDTA-free (Sigma-Aldrich). 20 μg of protein was mixed with 4xLaemmli Sample Buffer (Bio-Rad) and were run on a 4–20% Mini-PROTEAN® TGX™ Precast Protein Gels (Bio-Rad). Proteins were transferred on a nitrocellulose membrane and blocked for 2 h at room temperature in blocking solution (2% non-fatty milk diluted in TBS-Tween 0.1% – TBS-T) and incubated in primary antibody, in blocking solution, for 48 h at 4 °C (rabbit anti-NNAT, 1:1000, Abcam ab27266; rabbit anti-Tubulin, 1:2000, Cell Signaling 2125s). Membranes were washed 5x times in blocking solution and incubated in HRP (horseradish peroxidase)-

conjugated secondary antibody (1:10,000) in blocking solution for 1 h. Membranes were washed 5x in TBS-T, developed with the Clarity™ Western ECL Substrate (Bio-Rad) and visualized with the ChemiDocTM MP, Imaging System (Bio-Rad).

### Statistical analysis

Statistical tests were performed using GraphPad Prism version 10.0 for Windows (GraphPad Software, San Diego, CA, USA) or R. For data sets depicting human samples, violin plots were used. The number of independent experiments is indicated in the plots. Box plots represent median (box: two quantiles around the median; whiskers: minimum and maximum value; points superimposed on the graph: individual values). Normally distributed data were tested using two-sided Student's t-test or ANOVA followed by Tukey post hoc test. Nonnormal data were analyzed with the nonparametric test Mann–Whitney U-test. Novel Object Recognition was analyzed using Two-way ANOVA followed by Šídák's multiple comparisons test. Data distribution was tested with the Shapiro-Wilk test. Correlations were calculated using the Spearman correlation coefficient with two-tailed analysis. Significant changes in the BD and MDD patient group were determined by pairwise comparison using a nonparametric Wilcoxon rank-sum test. To control for the effect of additional factors, a linear model of the form $-\Delta Cq \sim$ Group + Sex + Age + Antidepressant treatment was used in R studio. $P < 0.05$ was considered statistically significant. The detailed parameters ($n$, $p$ value, test) for the statistical assessment of the data are provided in the figure legends. For ROC curves generation, to test a combination of the two miRNAs in classifying bipolar disorder patients from both controls and MDD patients, we first scaled the deltaCq values of each miRNA (to bring them to a comparable scale), and simply summed them up for each patient in order to produce the Receiver Operating Characteristic (ROC) curves (Fig. 6E,F; Appendix Fig. 6C–F). For single miRNAs, the deltaCq values themselves were used to segregate the groups. The indicated thresholds are the closest point to the optimal (top-left), and the reported areas under the curves were computed using the PRROC 1.3.1 package. Correlational analyses were conducted to examine the relationship between relative miR-708 expression and cognitive performance. Prior to analysis, normality of the data was assessed using Shapiro-Wilk tests. Given that all data were found to be normally distributed, bivariate Pearson correlation tests were employed. These statistical analyses were conducted using Jamovi (v2.3) and R (v4.1).

### Graphics

Graphs were produced using GraphPad Prism 10 software or R. Some diagrams were created with BioRender.com (Fig. 1A, Fig. 2D, Fig. 6B, Fig. EV2A, Fig. EV3E, Appendix Fig. 3A and Synopsis Figure).

## Data availability

RNA sequencing data were deposited to Gene Expression Omnibus (GEO): small RNA sequencing (Fig. 1B–E): GSE261287; and RNA sequencing (Fig. 4A): GSE261288. Links: https://www.ncbi.nlm.nih.gov/geo/query/acc.cgi?acc=GSE261287 and https://www.ncbi.nlm.nih.gov/geo/query/acc.cgi?acc=GSE261288, respectively.

The source data of this paper are collected in the following database record: biostudies:S-SCDT-10_1038-S44319-025-00410-y.

## Peer review information

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

## Acknowledgements

We greatly acknowledge the technical support in the preparation of primary hippocampal cultures by Cristina Furler, Tatjana Wüst and Dr. Roberto Fiore. We also thank Darren Kelly for support in designing the TDMD construct, and David Colameo and Emanuel Sonder for support in the statistical analysis. RNA sequencing was performed at the Functional Genomics Center Zurich (FGCZ) of the University Zurich and ETH Zurich. The work in the lab of GS was supported by grants from ETH Zurich (ETH-24 18-2 Grant (NeuroSno)) and the Swiss National Science Foundation (SNSF 310030E_179651, 32NE30_189486, 310030_205064/1). This work was further funded by the German Research Foundation (DFG grants FOR2107 KI588/14-1, and KI588/14-2, and KI588/20-1, KI588/22-1 to TK; grant FOR2107 DA1151/5-1, DA1151/5-2, DA1151/9-1, DA1151/10-1, DA1151/11-1 to UD; grant FOR2107 SCHR 1136/3-1 to GS; grant DFG WO 1732/4-1 and DFG WO 1732/4-2 to MW; grant DFG 559/14-1 and DFG 559/14-2 to RS), as well as the Fonds Wetenschappelijk Onderzoek – Vlaanderen (FWO; Research Foundation – Flanders) through a senior project to MW (G0C0522N), and the Interdisciplinary Center for Clinical Research (IZKF) of the medical faculty of Münster (grant Dan3/022/22 to UD). Biosamples and corresponding data were sampled, processed, and stored in the Marburg Biobank CBBMR.

## Author contributions

**Carlotta Gilardi**: Investigation; Writing—original draft. **Helena C Martins**: Investigation. **Brunno Rocha Levone**: Investigation; Writing—original draft; Writing—review and editing. **Alessandra Lo Bianco**: Investigation. **Silvia Bicker**: Investigation. **Pierre-Luc Germain**: Formal analysis. **Fridolin Gross**: Formal analysis. **Ayse Özge Sungur**: Investigation. **Theresa M Kisko**: Investigation. **Frederike Stein**: Data curation; Formal analysis. **Susanne Meinert**: Data curation. **Rainer K W Schwarting**: Supervision. **Markus Wöhr**: Supervision; Project administration. **Udo Dannlowski**: Supervision. **Tilo Kircher**: Supervision; Project administration. **Gerhard Schratt**: Conceptualization; Supervision; Writing—original draft; Project administration; Writing—review and editing.

Source data underlying figure panels in this paper may have individual authorship assigned. Where available, figure panel/source data authorship is listed in the following database record: biostudies:S-SCDT-10_1038-S44319-025-00410-y.

## Disclosure and competing interests statement

The authors declare no competing interests.

# Expanded View Figures

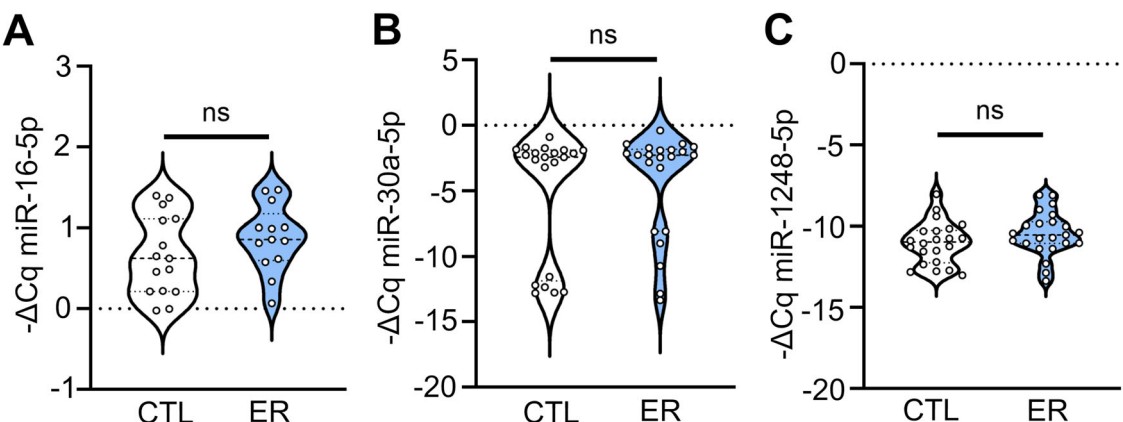

**Figure EV1. Unrelated miRNAs tested by qPCR as negative controls for the CTL *vs* ER small-RNA sequencing.**

(A) miR-16-5p qPCR analysis of total RNA isolated from PBMCs of CTL ($n=15$) and ER ($n=13$) subjects. Unpaired t-test, ns. Data are presented as violin plots with median, quartiles and data points. (B) miR-30a-5p qPCR analysis of total RNA isolated from PBMCs of CTL ($n=15$) and ER ($n=16$) subjects. Unpaired t-test, ns. Data are presented as violin plots with median, quartiles and data points. (C) miR-1248-5p qPCR analysis of total RNA isolated from PBMCs of CTL ($n=16$) and ER ($n=17$) subjects. Unpaired t-test, ns. Data are presented as violin plots with median, quartiles and data points. Source data are available online for this figure.

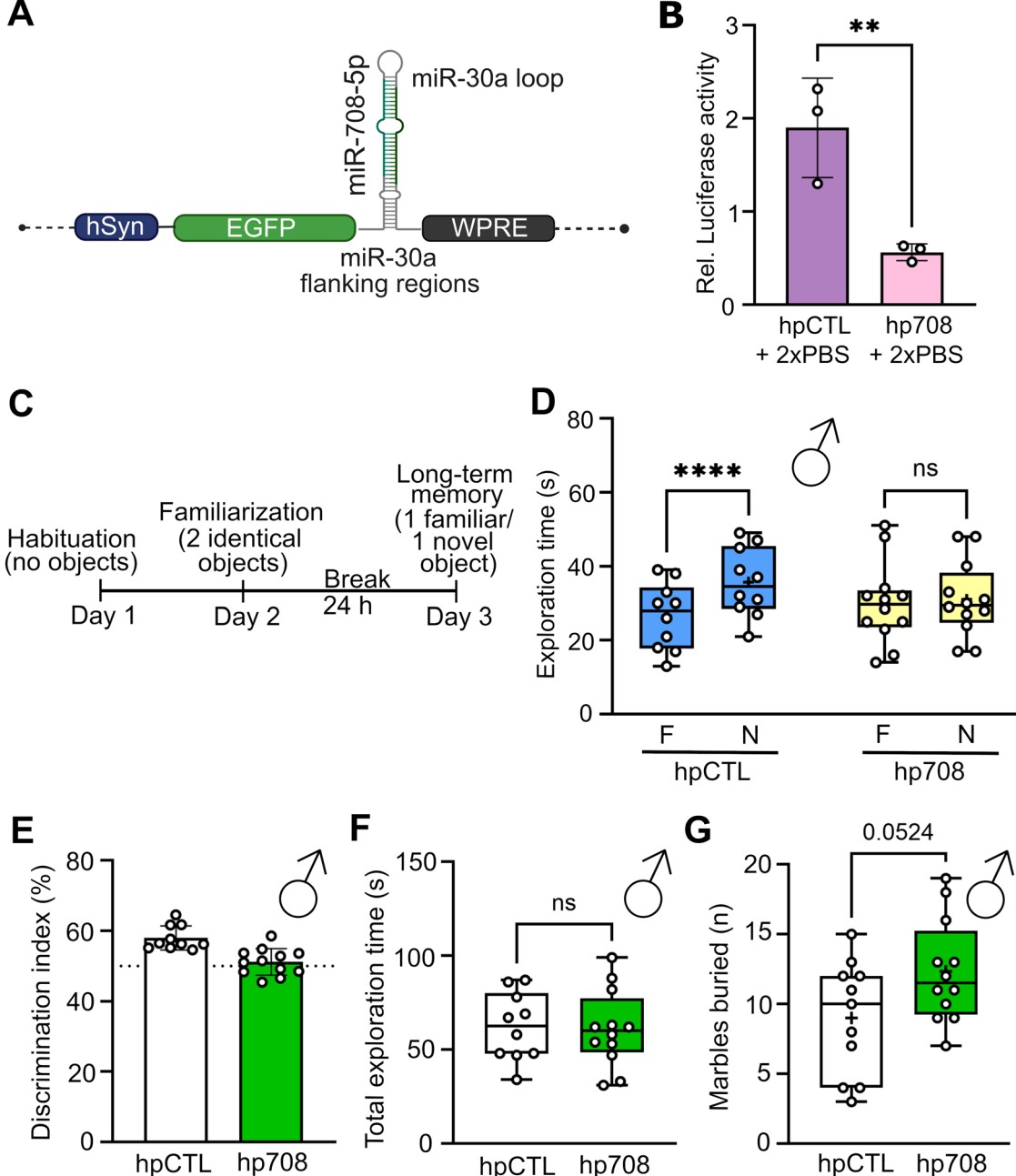

**Figure EV2. Total time exploring objects (novel plus familiar) and discrimination index for the novel object test.**

Related to Fig. 3. (**A**) Schematic representation of the miR-708-5p overexpressing hairpin (hp708), placed in the 3'-UTR of EGFP. (**B**) Relative luciferase activity of rat hippocampal neurons transfected with the indicated plasmid (hpCTL or hp708) and a luciferase reporter expressing 2x Perfect Binding sites (PBS) for miR-708-5p. Data are represented as scattered dot plots with bar, mean ± SD ($n = 3$ biological replicates). Ratio paired t test, two-tailed, **$p = 0.0057$. (**C**) Schematic representation of the Novel Object Recognition (NOR) test for long-term memory. The break between the familiarization and the long-term memory sessions is of 24 h. (**D**) Time (s) male mice injected with the indicated rAAV (hpCTL $n = 10$, hp708, $n = 12$) explored either the familiar (F) or novel (N) object. Data are represented as box plot with whiskers and data points (+: mean, line: median; whiskers: minimum and maximum values). Two-way RM ANOVA: Novelty × Group, ***$p = 0.0004$; Novelty, ****$p < 0.0001$; Group, ns, $p = 0.8830$. Šídák's post hoc test, F vs N: hpCTL, ****$p < 0.0001$; hp708, ns, $p = 0.6735$. (**E**) Discrimination index calculated as time spent exploring novel object/time spent exploring novel and familiar objects for male mice injected with the indicated rAAV (hpCTL $n = 12$, hp708, $n = 11$) in the Novel Object Recognition long-term session. Data are represented as scattered dot plots with bar, mean ± SD. (**F**) Time (s) male mice injected with the indicated rAAV (hpCTL $n = 12$, hp708, $n = 11$) explored the familiar (F) and novel (N) object in the Novel Object Recognition long-term session. Data are represented as box plot with whiskers and data points (+: mean, line: median; whiskers: minimum and maximum values). Unpaired t-test, ns, $p = 0.8869$. (**G**) Number of marbles male mice injected with the indicated rAAV (hpCTL $n = 11$, hp708, $n = 12$) buried in 30 min. Data are represented as box plot with whiskers and data points (+: mean, line: median; whiskers: minimum and maximum values). Unpaired t-test, $p = 0.0524$. Source data are available online for this figure.

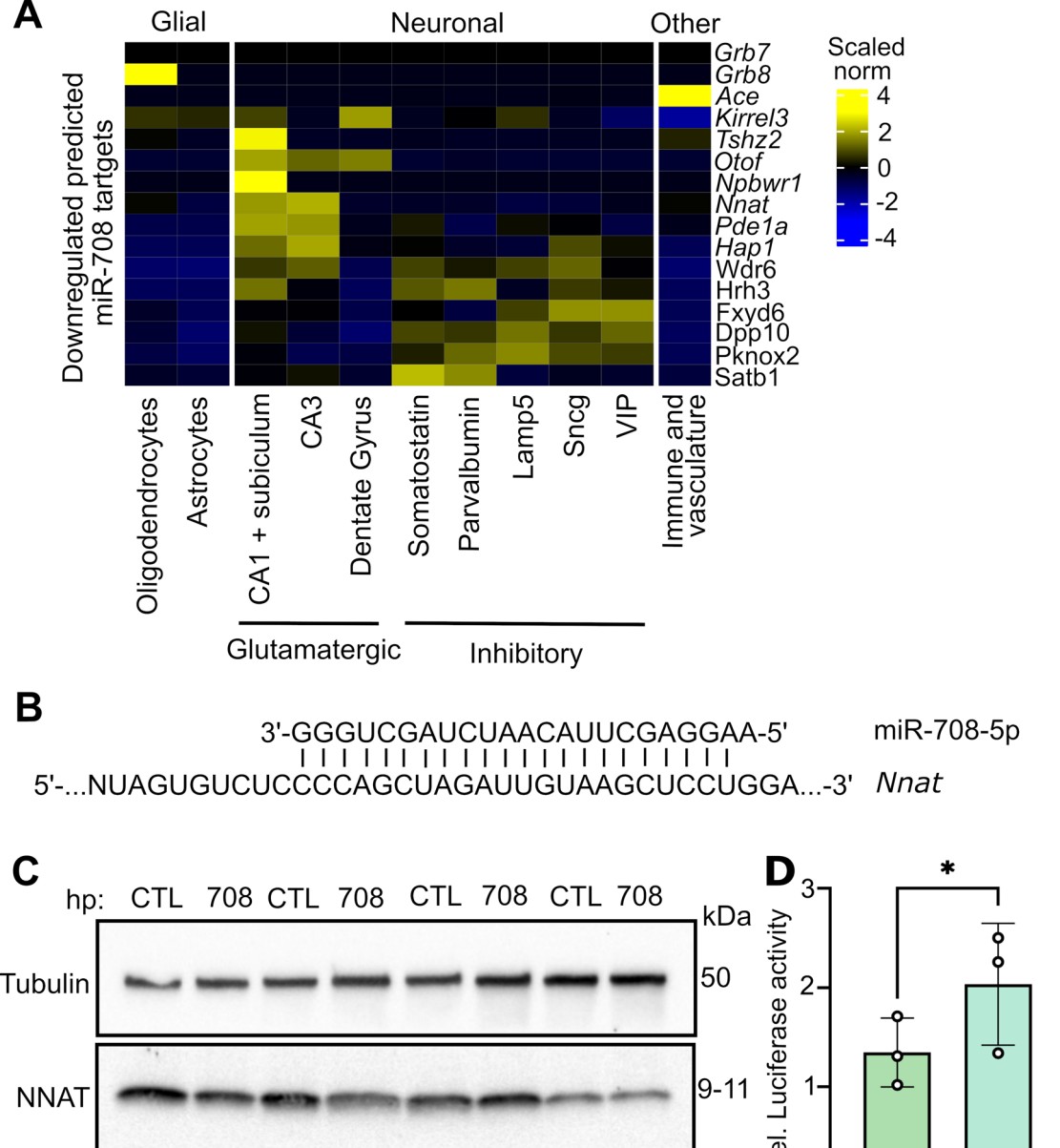

**Figure EV3.  Validation of *Nnat* as a miR-708-5p target.**

Related to Fig. 4. (A) Heatmap displaying miR-708-5p predicted targets which are significantly downregulated (from Fig. 4A), as well as their expression in different cell types based on single-cell RNA-seq data (Allen 10X+smartSeq taxonomy). (B) Nucleotide base pairing depicting the perfect binding of miR-708-5p with the mouse *Nnat* 3′-UTR. (C) Full Western blot image (Fig. 4E) of NNAT (lower panel) and Tubulin (upper panel) protein expression levels in hippocampal neurons (20 days in vitro) that were infected with hpCTL or hp708 at 2 days in vitro. (D) Relative luciferase activity of rat hippocampal neurons transfected with the indicated plasmid (sponge control: spCTL, miR-708-5p sponge: sp708) and a luciferase reporter expressing 2x Perfect Binding Sites (PBS) for miR-708-5p. Data are represented as scattered dot plots with bar, mean ± SD ($n = 3$ biological replicates). Ratio paired t-test, two-tailed, $*p = 0.0372$. (E) Schematic representation of the construct to knock-down miR-708-5p via six TDMD sites, used in Fig. 4G (right) and Fig. EV3D. Source data are available online for this figure.

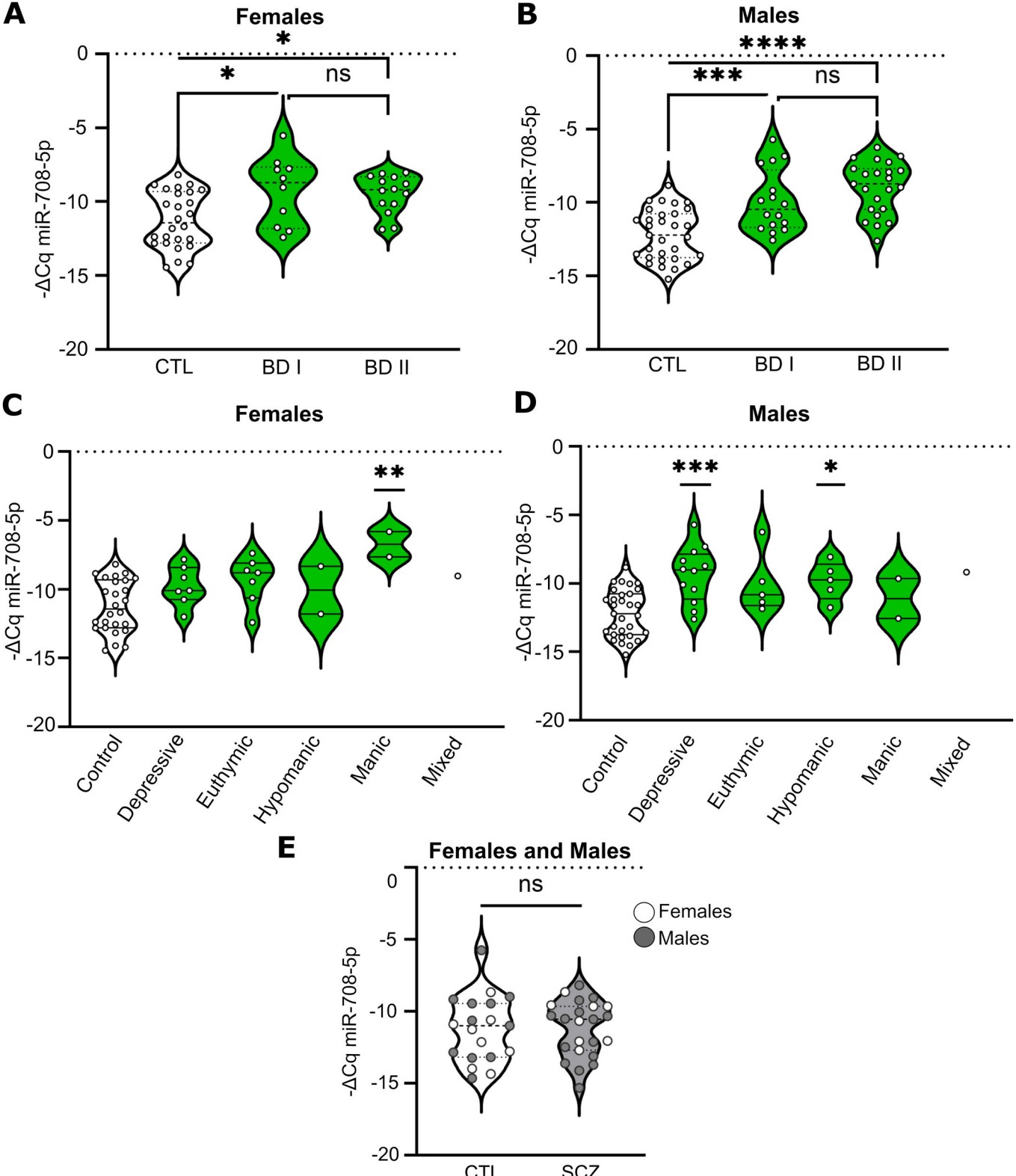

◀

**Figure EV4.  miR-708-5p expression levels in the PBMCs of subjects with different bipolar types and mood states.**

Related to Fig. 6. (A) miR-708-5p expression levels in PBMCs of female subjects with Bipolar disorder type I (BD I) or type II (BD II) (CTL, $n = 26$; BD I, $n = 10$; BD II, $n = 14$). Kruskal Wallis Test, Post hoc Dunn's: Control *vs* BD I: *$p = 0.0253$, Control *vs* BD II: *$p = 0.0196$, BD I *vs* BD II: ns, $p > 0.9999$. Data are presented as violin plots with median, quartiles, and data points. (B) miR-708-5p expression levels in PBMCs of male subjects with BD I or BD II (CTL $n = 31$; BD I, $n = 16$; BD II, $n = 23$). One-way ANOVA, Post hoc Tukey's test: Control *vs* BD I: ***$p = 0.0005$, Control *vs* BD II: ****$p = 0.0001$, BD I *vs* BD II: ns, $p = 0.2690$. Data are presented as violin plots with median, quartiles, and data points. (C) miR-708-5p qPCR analysis of total RNA isolated from PBMCs of healthy control female subjects (control, $n = 26$) and BD subjects in different mood states (depressive, $n = 7$; euthymic, $n = 7$; hypomanic, $n = 25$; manic, $n = 2$; mixed, $n = 1$). Kruskal Wallis Test, Post hoc Dunn's: Control *vs* Manic, *$p = 0.0300$. Data are presented as violin plots with median, quartiles, and data points. (D) miR-708-5p qPCR analysis of total RNA isolated from PBMCs of healthy control male subjects (control, $n = 31$) and BD subjects in different mood states (depressive, $n = 12$; euthymic, $n = 5$; hypomanic, $n = 5$; manic, $n = 2$; mixed, $n = 1$). One-way ANOVA, Post hoc Dunnett's test: Control *vs* Depressive, ***$p = 0.0002$; Control *vs* Hypomanic, *$p = 0.0490$. Data are presented as violin plots with median, quartiles, and data points. (E) miR-708-5p qPCR analysis of total RNA isolated from PBMCs of male and female patients diagnosed with Schizophrenia (male: CTL $n = 11$, SCZ $n = 15$; female: CTL $n = 8$, SCZ $n = 8$). Two-way ANOVA: Group × Sex, ns, $p = 0.4904$; Group, ns, $p = 0.2383$, Sex, ns, $p = 0.4904$. Tukey's post hoc test, ns. Data are presented as violin plots with median, quartiles, and data points. Source data are available online for this figure.

                                                                                       