## [Peer Review File · EMBO Reports]

miR-708-5p is elevated in bipolar patients and can induce mood disorder-associated behavior in mice

Carlotta Gilardi, Helena Martins, Brunno Rocha Levone, Alessandra Lo Bianco, Silvia Bicker, Pierre-Luc Germain, Fridolin Gross, Ayse Sungur, Theresa Kisko, Frederike Stein, Susanne Meinert, Rainer Schwarting, Markus Wöhr, Udo Dannlowski, Tilo Kircher, and Gerhard Schratt

Corresponding author(s): Gerhard Schratt (gerhard.schratt@hest.ethz.ch)

Review Timeline:

Transfer Date:	4th Nov 24
Editorial Decision:	12th Nov 24
Revision Received:	8th Jan 25
Editorial Decision:	14th Feb 25
Revision Received:	21st Feb 25
Accepted:	24th Feb 25

Editor: Esther Schnapp

Transaction Report: This manuscript was transferred to EMBO reports following peer review at Review Commons.

**Review
COMMONS**

Review #1

1. Evidence, reproducibility and clarity:

Evidence, reproducibility and clarity (Required)

****Summary:****

Mood disorders (MD), including bipolar disorder (BD) and major depressive disorder (MDD), are serious mental health conditions caused by a complex interplay of genetic and environmental risk factors (GxE). However, little is known about how miRNA interactions contribute to these diseases, which limits the use of miRNAs as potential biomarkers for MD diagnosis. The study by Gilardi and colleagues aims to identify MD-associated mature miRNAs (miRs) from peripheral blood mononuclear cells (PBMCs) in a large human cohort of healthy individuals at high GxE risk for developing MD, using small RNA sequencing (RNA-seq). This unbiased, back-translational approach led to the identification of BD-associated miR-708-5p, which was further validated using luciferase and qPCR assays. The functional role of miR-708-5p was characterized at the molecular and behavioral levels in vivo and in vitro in rodent models through viral overexpression in the mouse hippocampus. Finally, the diagnostic potential of miR-708-5p was assessed in MD patient subgroups.

****Major comments:**** The manuscript is well-structured, and the findings presented here provide translational value for MD-associated miRNAs. The identified candidate, miR-708-5p, and its predicted binding target, the endoplasmic reticulum (ER) resident protein Neuronatin (Nnat), were carefully validated and confirmed as direct targets of miR-708-5p. Furthermore, comprehensive behavioral assays were conducted in mice overexpressing miR-708-5p in the hippocampus using AAV constructs, demonstrating a causal and robust MD-associated behavioral phenotype. Notably, the authors elegantly showed that by using an engineered viral construct that simultaneously overexpresses both Nnat and miR-708, they were able to restore Nnat expression in the hippocampus of miR-708-5p overexpressing adult mice, rescuing BD-associated behavioral phenotypes.

The major concern with this study is the lack of follow-up functional characterization of the identified miR-708-5p target, Nnat, in the rodent model, particularly its role in calcium homeostasis processes. Additionally, there is no evidence provided to show that this gene is upregulated in patients suffering from BD. It would be valuable to explore whether aberrant miR-708/Nnat signaling impacts intraneuronal calcium homeostasis in this psychiatric disorder, as this could have implications for developing new therapeutic interventions. However, such an exploration may be beyond the scope of the present study, which aims primarily to identify novel potential biomarkers for MD.

****Minor Comments:****

The manuscript should be carefully checked for reference formatting, as there are instances of mixed formats (numbers and names) and missing details in the figure legends that should be consistent with the figure titles used in the manuscript. Variations between the figure titles and the corresponding manuscript text should be addressed.

Introduction: The behavioral assays mentioned in the introduction (e.g., lines 72-73) should be spelled out in full before using abbreviations, as not every reader may have a behavioral background.

Supplementary Materials and Methods: The age of the 3-month-old mice used for stereotactic surgeries (see page 2) does not align with the 7/8-week-old mice mentioned in the main text. Additionally, in the Statistical Analysis section, the Y-maze is referenced but not shown in the manuscript.

Results: During the unbiased small RNA sequencing assay, six mature miRNAs were identified in the GxE groups. While it is reasonable to focus on miR-708-5p due to its recent association with BD, it would be interesting to briefly discuss the other miRNA candidates given the unbiased nature of the approach. Is there literature or in silico analysis (e.g., miRNA-related GO terms) that could be relevant to share? For better clarity, I would recommend highlighting miR-708 in the tables shown in Fig. 1B and C to provide quick orientation for the reader.

Fig. 2E: Since other behavioral panels include a sex symbol/icon to identify the sex of the animals used, it would be consistent to include the male icon here as well. The same applies to panels in Fig. 5D and E.

Fig. 3: Is there a reason why females were not tested in the NOR assay after a 24-hour break? Perhaps panel 3G could be moved to the supplements to maintain consistency, as males and females are otherwise tested in the other assays shown.

It would be helpful to edit the schematic panel in Fig. 3A to add "qPCR" at the timeline's end, indicating when animals were measured for miR-708-5p expression levels in the hippocampus. I would suggest moving Fig. 3C to the end, as the qPCR was performed after the behavioral assays were completed.

Fig. 3B: Are higher magnification images from the injected hippocampi available, including additional neuronal and astrocyte markers (e.g., NeuN, GFAP), to demonstrate in which hippocampal cells the virus is expressed?

I would recommend moving the WB panel in Fig. 4H to the supplements or focusing on a smaller representative area (e.g., the first four lines) to demonstrate the downregulation of the Nnat protein. I am not convinced of the relevance of the Fig. 6E panel in this study, which primarily deals with MD, as miR-708-5p expression levels in CTL and schizophrenia (SCZ) patient PBMCs were analyzed.

Given the high genetic similarities between SCZ and BD, the sample size for SCZ is on the lower end (see females CTL n=8 vs. SCZ n=8), particularly if the data were to be presented separately by sex as done with the MD patients (see Fig. 6C-D). Therefore, I am not entirely convinced that miR-708-5p is unaffected in SCZ patients; the sample size may simply not be sufficient to detect a difference in expression levels, especially considering the heterogeneity of SCZ. I suggest moving panel Fig. 6E to the supplements and briefly discussing the lack of findings in SCZ patients in the main text.

Supplemental Fig. 1C-E: It is unclear from the graph or figure legend what these three graphs specifically refer to. Are the X-axis values animal numbers, and do graphs C-E represent different cohorts used for the qPCR assay? Additionally, in Suppl. Fig. 2J, please clarify the difference between this group of animals and the one shown in Suppl. Fig. 2H in the figure legend description.

2. Significance:

Significance (Required)

This translational study is well-structured and meticulously conducted, demonstrating the role of miR-708-5p in regulating MD-associated behavioral phenotypes in rodents. It represents a first step toward deepening our understanding of the specific functions of miR-708-5p in mood disorders (MD), its potential as a diagnostic biomarker, and insights into biological mechanisms that may offer translational value for MD therapeutics in the future. Overall, the study is informative and valuable, particularly for scientists in translational and psychiatric research who seek to develop novel diagnostic biomarkers for bipolar disorder (BD).

Audience: The study by Gilardi et al. will be of great interest to scientists from both basic and translational/clinical neuroscience.

Own Field of Expertise: Neurodevelopmental and psychiatric diseases, genetics, behavior, miRNA.

3. How much time do you estimate the authors will need to complete the suggested revisions:

Estimated time to Complete Revisions (Required)

(Decision Recommendation)

Cannot tell / Not applicable

4. Review Commons values the work of reviewers and encourages them to get credit for their work. Select 'Yes' below to register your reviewing activity at Web of Science Reviewer Recognition Service (formerly Publons); note that the content of your review will not be visible on Web of Science.

No

Review #2

1. Evidence, reproducibility and clarity:

Evidence, reproducibility and clarity (Required)

****Summary:****

To address a critical gap in identifying potential biomarkers detectable in blood for individuals predisposed to psychiatric conditions, the current study analyzed blood samples from individuals at risk for and diagnosed with mood disorders, with molecular characterization in rodent models and in vitro.

Authors identify elevated miR-708 levels in the blood of female participants at high risk for mood disorders due to environmental and genetic factors. In female rats, miR-708 expression was identified to be expressed in different brain regions. The study focuses on miR-708 expression in the

hippocampus, localizing the expression to the neurons and altered levels in male rat model for environmental or genetic risk for behavioral abnormalities.

Given the upregulation of miR-708 in both rat model and humans at risk, miR-708 was upregulated with viral vectors in adult mouse hippocampus and subsequently tested for differences on a behavioral level. In the tail-susception test male mice with high hippocampal levels of miR-708 showed reduced immobility and lack of novel object exploration compared to familiar, whereas in female mice the test could be conclusive when tested within the same test session with a 5-minute interval.

Differential expression analysis of high-throughput gene expression measure in the hippocampus between mice with control or virally upregulated levels of miR-708 identified neuronatin gene to be altered and contain the matching sequence for miR-708. In vitro study confirmed the direct regulation of neuronatin RNA and protein by miR-708. In male mice with modulated miR-708 dependent neuronatin downregulation showed neuronatin-level dependent novel object exploration.

Applying the principle of miR-708 involvement in molecular pathways and behavioral abnormalities in animals, blood levels of miR-708 were measured in clinical depression and bipolar disorder populations. A negative association between D2 attention scores and miR-708 levels was observed in the full cohort. Greater upregulated levels of miR-708 in blood were observed for the group with bipolar disorder diagnosis in males and females. Furthermore, a combination of miR-708 and a previously identified microRNA involved in bipolar disorder, miR-499a, showed high sensitivity and specificity of differentiating individuals with bipolar disorder and healthy controls, with better discrimination in males.

This study is commendable for its ambitious approach and extensive efforts, particularly in the challenging field of microRNAs as biomarkers in psychiatric research. However, despite the authors' endeavors, the presentation of data and results have several major caveats that undermine confidence in their conclusions. The study jumps inconsistently from different species, sex, and models. There is also a need for additional control conditions, such as including another microRNA, brain region and human behavioral scores, which are essential in presenting results that are not fitted to a desired outcome.

****Major comments:****

1. The study describes the availability of both male and female participants' blood, however the analysis was only pursued in females (lines 326 to 328). Based on the table provided, it is not clear that the sample composition is better for females and thus serves as a strong justification for differential expression solely based on females.
2. In line 78 social communication is used to describe mood disorder associated endophenotypes, which is a seemingly critical measure not included in the study. Justification for the measure of solely memory and attention aspect needs to be strengthened. The link between *Cacna1c*, hippocampus and memory has been previously described in the literature, however the relation between memory deficits and bipolar disorder as the main feature of the endophenotype is missing

clarity in the manuscript.

3. In the introduction section, expected behaviors in animal models for MD-associated behavioral endophenotypes (similar to lines 426 and 642) and potential differences for bipolar versus depression animal models should be better explained. As is, Figure 3 does not provide convincing experimental evidence for miR-708 role in MD-associated behaviors, where the data in Figure 3D presents that animals with upregulated miR-708 show less immobility, indicative of reduced despair-like behavior compared to control mice and no difference in anhedonia or in anxiety. As such, the statement in the abstract (line 41) that there is a causal role cannot be made.

Additionally, the use of female mice in Figure 3 but not female rats in Figure 2 is unclear.

4. The focus on the hippocampus is not well described, as authors state in line 650 that their approach is contrary to most models reported in the literature. The study does not appear to differentiate between the different subsections of the hippocampus, which are well known to have different functional roles. Method description is often lacking, for example, it is not clear what hippocampal section punches were taken for subsequent sequencing experiment in Figure 4. The sex and number of animals injected with either hp708 and hpCTL and used for sequencing is not indicated anywhere.

5. The introduction of miR-499a-5p in Figure 6F & G comes as a surprise.

6. Line 605 refers to a literature source that identified miR-708 upregulation in the PFC in rats susceptible to stress-induced depressive like behaviors, however the authors do not consider PFC or provide the reason for not pursuing miR-708 expression in the PFC in the manuscript.

7. The manuscript needs an explanation why mononuclear cells instead of other blood fractions was used. The link between altered levels of miR-708 in the PBMCs and the hippocampus is not clear.

****Minor comments:****

- In addition to the information in the legends, the figures themselves need to indicate the sex or other information that the graphs are presenting.

- Fig 1D would benefit from inclusion of another microRNA expression in ER and GR, without the need for a qPCR, in addition to sequencing expression levels.

- For Figure 1, why are the FDR adjusted p-values not included as part of the tables? Additionally, the top DE miRNAs such as miR-100 and miR-412 are not looked at or explained why they haven't been investigated further, given the stated unbiased approach objective.

- For Figure 2, since the sex of the animal is not consistent throughout, need to add a symbol next to the graph. For line 356 there is a jump from looking at the expression in the whole brain to focusing on the hippocampus, need a clear justification at the beginning (e.g. lines 388-390).

- Fig 2F would benefit from another brain region and another miR as controls. Did not explain why double hit in genetic and environmental risk factors seem to cancel each other. The study does not show differences between genetic versus environmental risks in humans or rodent models, perhaps this can be addressed in the discussion.

- Need to expand abbreviation at first use, such as in lines 56, 72-74

- Line 316, "miRNAs causally involved in MD etiology" is not a factual statement and should be toned down, especially given that this section describes "healthy" at-risk participants.

- Have the authors considered replicating the upregulated levels of miR-708 in the blood of rats, similar to humans? Association between blood and hippocampal/brain levels would add a stronger argument for the methodological approach.
- Figure 3C does not normalize to control, this needs to be re-graphed. Why are there several different colors for individual animal dots?
- Figure 3G, the inconsistency in not presenting data for females should be addressed.
- Line 448 refers to a published single-cell RNA seq data which is lacking a reference
- Is there a literature source that discusses the connection between the D2 attention test in humans and the object recognition test in rodents (line 523 with 529)? Is there support between D2 test and bipolar disorder? Supplying these references in the introduction would be beneficial to introduce the focus of the experiments pursued here.
- The object recognition involves 5 minutes interval and a 24 hours interval, it is not clear whether the same animal is exposed to both intervals and the same "novel" object? In which case, could it be that the repeated exposure leads to the inconclusive results in females?
- Fig 6A - the D2 task alone is not sufficient, need similar test and unrelated measure to confidently conclude the association.
- Some grammatical editing is needed, for example lines 386, 387, 717
- The methods section needs reorganization as some descriptives are not mentioned at first use as in the results (e.g. line 146 needs to expand on the sequencing details because the link is insufficient), and the division between "human study" and "animal study" are not consistent (e.g. rat culture method description under human section).

2. Significance:

Significance (Required)

The introduction presents a compelling objective: an unbiased, back-translational approach to identify microRNAs associated with mood disorders from a large cohort of healthy individuals at high genetic and environmental risk. However, the results do not align with this objective. The authors do not demonstrate an unbiased approach to identifying target miRNAs, and the analysis falls short of showcasing risk prediction in non-diagnosed individuals. Instead, the study begins with at-risk individuals and concludes by demonstrating diagnostic ability in already diagnosed individuals, which is counterintuitive. Furthermore, the microRNA previously implicated in bipolar disorder by the authors is only mentioned at the end, without clarifying how this study relates to their prior findings. Addressing these concerns will significantly strengthen the interpretation of the results and their significance for the field.

3. How much time do you estimate the authors will need to complete the suggested revisions:

Estimated time to Complete Revisions (Required)

(Decision Recommendation)

More than 6 months

4. Review Commons values the work of reviewers and encourages them to get credit for their work. Select 'Yes' below to register your reviewing activity at Web of Science Reviewer

Recognition Service (formerly Publons); note that the content of your review will not be visible on Web of Science.

Yes

Revision Plan

Manuscript number: RC- 2024-02694

Corresponding author(s): Schratt, Gerhard

1. General Statements [optional]

Not applicable.

2. Description of the planned revisions

Reviewer #1 (Evidence, reproducibility and clarity (Required)):

Summary:

Mood disorders (MD), including bipolar disorder (BD) and major depressive disorder (MDD), are serious mental health conditions caused by a complex interplay of genetic and environmental risk factors (GxE). However, little is known about how miRNA interactions contribute to these diseases, which limits the use of miRNAs as potential biomarkers for MD diagnosis. The study by Gilardi and colleagues aims to identify MD-associated mature miRNAs (miRs) from peripheral blood mononuclear cells (PBMCs) in a large human cohort of healthy individuals at high GxE risk for developing MD, using small RNA sequencing (RNA-seq). This unbiased, back-translational approach led to the identification of BD-associated miR-708-5p, which was further validated using luciferase and qPCR assays. The functional role of miR-708-5p was characterized at the molecular and behavioral levels in vivo and in vitro in rodent models through viral overexpression in the mouse hippocampus. Finally, the diagnostic potential of miR-708-5p was assessed in MD patient subgroups.

Major comments: The manuscript is well-structured, and the findings presented here provide translational value for MD-associated miRNAs. The identified candidate, miR-708-5p, and its predicted binding target, the endoplasmic reticulum (ER) resident protein Neuronatin (Nnat), were carefully validated and confirmed as direct targets of miR-708-5p. Furthermore, comprehensive behavioral assays were conducted in mice overexpressing miR-708-5p in the hippocampus using AAV constructs, demonstrating a causal and robust MD-associated behavioral phenotype. Notably, the authors elegantly showed that by using an engineered viral construct that simultaneously overexpresses both Nnat and miR-708, they were able to restore Nnat expression in the hippocampus of miR-708-5p overexpressing adult mice, rescuing BD-associated behavioral phenotypes.

- We are happy that this reviewer emphasizes the high translational value of our study and further highlights the significance of our target validation and mouse behavioural approach

The major concern with this study is the lack of follow-up functional characterization of

Revision Plan

the identified miR-708-5p target, Nnat, in the rodent model, particularly its role in calcium homeostasis processes. Additionally, there is no evidence provided to show that this gene is upregulated in patients suffering from BD. It would be valuable to explore whether aberrant miR-708/Nnat signaling impacts intraneuronal calcium homeostasis in this psychiatric disorder, as this could have implications for developing new therapeutic interventions. However, such an exploration may be beyond the scope of the present study, which aims primarily to identify novel potential biomarkers for MD.

- We agree with this reviewer that in this study, we have not explored the molecular mechanisms downstream of Neuronatin, e.g., the role of miR-708/Nnat signaling in intraneuronal calcium homeostasis processes. However, as correctly pointed out by this reviewer, the major emphasis of this paper was on the identification of microRNAs as potential biomarkers for human mood disorders. In addition, we have provided evidence for a functional involvement of the miR-708/Nnat pathway in mood-disorder associated behavioural endophenotypes in mice. Therefore, further mechanistic studies on Nnat in neuronal calcium homeostasis are beyond the scope of the current study and will form the basis for a separate manuscript.

Minor Comments:

The manuscript should be carefully checked for reference formatting, as there are instances of mixed formats (numbers and names) and missing details in the figure legends that should be consistent with the figure titles used in the manuscript. Variations between the figure titles and the corresponding manuscript text should be addressed.

- In a revised version, we will fix all issues related to reference formatting and figure legends/titles.

Introduction: The behavioral assays mentioned in the introduction (e.g., lines 72-73) should be spelled out in full before using abbreviations, as not every reader may have a behavioral background.

- We will spell out all terms before using abbreviations in the revised paper.
- Supplementary Materials and Methods: The age of the 3-month-old mice used for stereotactic surgeries (see page 2) does not align with the 7/8-week-old mice mentioned in the main text. Additionally, in the Statistical Analysis section, the Y-maze is referenced but not shown in the manuscript.
- Mice were usually 2 months old at the time of surgery. We will check for consistency in the revised manuscript.

Results: During the unbiased small RNA sequencing assay, six mature miRNAs were identified in the GxE groups. While it is reasonable to focus on miR-708-5p due to its recent association with BD, it would be interesting to briefly discuss the other miRNA candidates given the unbiased nature of the approach. Is there literature or in silico analysis (e.g., miRNA-related GO terms) that could be relevant to share? For better clarity, I would recommend highlighting miR-708 in the tables shown in Fig. 1B and C to provide quick orientation for the reader.

- We will include available references for the other candidate GxE miRNAs identified in the small RNA sequencing screen. Furthermore, we will highlight miR-708 in the tables for better clarity.

Revision Plan

Fig. 2E: Since other behavioral panels include a sex symbol/icon to identify the sex of the animals used, it would be consistent to include the male icon here as well. The same applies to panels in Fig. 5D and E.

- We will include the sex symbols in the revised manuscript when appropriate.

Fig. 3: Is there a reason why females were not tested in the NOR assay after a 24-hour break? Perhaps panel 3G could be moved to the supplements to maintain consistency, as males and females are otherwise tested in the other assays shown.

- Females were not tested in the 24-hour break paradigm since in the initially used 5-min short term memory paradigm (Fig. 3H, I), males displayed a much stronger memory impairment compared to females. For consistency, we will move Fig. 3G to the supplement of the revised manuscript.

It would be helpful to edit the schematic panel in Fig. 3A to add "qPCR" at the timeline's end, indicating when animals were measured for miR-708-5p expression levels in the hippocampus.

- We will edit the schematic panel as suggested by the reviewer.

I would suggest moving Fig. 3C to the end, as the qPCR was performed after the behavioral assays were completed.

- We will move Fig. 3C to the end of Fig. 3 as suggested by the reviewer.

Fig. 3B: Are higher magnification images from the injected hippocampi available, including additional neuronal and astrocyte markers (e.g., NeuN, GFAP), to demonstrate in which hippocampal cells the virus is expressed?

- Unfortunately, we do not have any higher magnification images of the injected hippocampi. Experiments would have to be repeated to obtain such images, which in our view is beyond the scope.

I would recommend moving the WB panel in Fig. 4H to the supplements or focusing on a smaller representative area (e.g., the first four lines) to demonstrate the downregulation of the Nnat protein.

- As suggested by the reviewer, we will focus on a smaller representative area of the WB in the main figure (Fig. 4H) and show the entire blot in the supplement of the revised manuscript.

I am not convinced of the relevance of the Fig. 6E panel in this study, which primarily deals with MD, as miR-708-5p expression levels in CTL and schizophrenia (SCZ) patient PBMCs were analyzed. Given the high genetic similarities between SCZ and BD, the sample size for SCZ is on the lower end (see females CTL n=8 vs. SCZ n=8), particularly if the data were to be presented separately by sex as done with the MD patients (see Fig. 6C-D). Therefore, I am not entirely convinced that miR-708-5p is unaffected in SCZ patients; the sample size may simply not be sufficient to detect a difference in expression levels, especially considering the heterogeneity of SCZ. I suggest moving panel Fig. 6E to the supplements and briefly discussing the lack of findings in SCZ patients in the main text.

- We feel strongly that showing the data on SCZ is important, since it nicely demonstrates the specificity of the miR-708-5p upregulation in BD compared to the genetically similar SCZ. However, we also agree with this reviewer that the data on SCZ might be slightly underpowered to make strong statements. Therefore, as suggested by the reviewer, we will move the SCZ data to the supplement of the revised manuscript.

Supplemental Fig. 1C-E: It is unclear from the graph or figure legend what these three graphs specifically refer to. Are the X-axis values animal numbers, and do graphs C-E represent different cohorts used for the qPCR assay? Additionally, in Suppl. Fig. 2J,

Revision Plan

please clarify the difference between this group of animals and the one shown in Suppl. Fig. 2H in the figure legend description.

- We will clarify the axis labels in the revised manuscript.

Reviewer #1 (Significance (Required)):

Significance: This translational study is well-structured and meticulously conducted, demonstrating the role of miR-708-5p in regulating MD-associated behavioral phenotypes in rodents. It represents a first step toward deepening our understanding of the specific functions of miR-708-5p in mood disorders (MD), its potential as a diagnostic biomarker, and insights into biological mechanisms that may offer translational value for MD therapeutics in the future. Overall, the study is informative and valuable, particularly for scientists in translational and psychiatric research who seek to develop novel diagnostic biomarkers for bipolar disorder (BD).

Audience: The study by Gilardi et al. will be of great interest to scientists from both basic and translational/clinical neuroscience.

Own Field of Expertise: Neurodevelopmental and psychiatric diseases, genetics, behavior, miRNA.

Reviewer #2 (Evidence, reproducibility and clarity (Required)):

Summary:

To address a critical gap in identifying potential biomarkers detectable in blood for individuals predisposed to psychiatric conditions, the current study analyzed blood samples from individuals at risk for and diagnosed with mood disorders, with molecular characterization in rodent models and in vitro.

Authors identify elevated miR-708 levels in the blood of female participants at high risk for mood disorders due to environmental and genetic factors. In female rats, miR-708 expression was identified to be expressed in different brain regions. The study focuses on miR-708 expression in the hippocampus, localizing the expression to the neurons and altered levels in male rat model for environmental or genetic risk for behavioral abnormalities.

Given the upregulation of miR-708 in both rat model and humans at risk, miR-708 was upregulated with viral vectors in adult mouse hippocampus and subsequently tested for differences on a behavioral level. In the tail-susception test male mice with high hippocampal levels of miR-708 showed reduced immobility and lack of novel object exploration compared to familiar, whereas in female mice the test could be conclusive when tested within the same test session with a 5-minute interval.

Differential expression analysis of high-throughput gene expression measure in the hippocampus between mice with control or virally upregulated levels of miR-708 identified neuronatin gene to be altered and contain the matching sequence for miR-708. In vitro study confirmed the direct regulation of neuronatin RNA and protein by miR-708. In male mice with modulated miR-708 dependent neuronatin downregulation showed neuronatin-level dependent novel object exploration.

Applying the principle of miR-708 involvement in molecular pathways and behavioral abnormalities in animals, blood levels of miR-708 were measured in clinical depression and bipolar disorder populations. A negative association between D2 attention scores and miR-708 levels was observed in the full cohort. Greater upregulated levels of miR-708 in blood were observed for the group with bipolar disorder diagnosis in males and females. Furthermore, a combination of miR-708 and a previously identified microRNA involved in bipolar disorder, miR-499a, showed high sensitivity and specificity of differentiating individuals with bipolar disorder and healthy controls, with better discrimination in males.

This study is commendable for its ambitious approach and extensive efforts, particularly in the challenging field of microRNAs as biomarkers in psychiatric research. However, despite the authors' endeavors, the presentation of data and results have several major caveats that undermine confidence in their conclusions. The study jumps inconsistently from different species, sex, and models. There is also a need for additional control conditions, such as including another microRNA, brain region and human behavioral scores, which are essential in presenting results that are not fitted to a desired outcome.

- We are happy that this reviewer acknowledges our ambitious approach and extensive efforts towards microRNA biomarker discovery in psychiatric research. As outlined in more detail below, we are further confident that we can address in full all the remaining concerns of this reviewer with respect to the selection of species, sex, models and brain regions in our revised manuscript.

Major comments:

1. The study describes the availability of both male and female participants' blood, however the analysis was only pursued in females (lines 326 to 328). Based on the table provided, it is not clear that the sample composition is better for females and thus serves as a strong justification for differential expression solely based on females.

- For our RNA sequencing screen, we used healthy individuals with high genetic or environmental risk to develop MD. For these groups (in contrast to the BD and MDD patient groups), it was in fact impossible to form a large, homogenous subgroup for males (suppl. Table 1 only provides the data for female subjects!). This was due to the presence of many confounding factors (e.g., smoking) in the male subjects which led to their exclusion from the study.

2. In line 78 social communication is used to describe mood disorder associated endophenotypes, which is a seemingly critical measure not included in the study. Justification for the measure of solely memory and attention aspect needs to be strengthened. The link between *Cacna1c*, hippocampus and memory has been previously described in the literature, however the relation between memory deficits and bipolar disorder as the main feature of the endophenotype is missing clarity in the manuscript.

- We would like to point out that, in addition to the memory and attention aspects mentioned by this reviewer, we have also interrogated behavioural despair (using TST), anxiety (EPM, open field), anhedonia (sucrose preference) and compulsive/manic behaviour (using marble burying) in our comprehensive behavioural test battery. Upon

miR-708-5p, we observed significant changes in memory (Fig. 3F-I), but also reduced behavioural despair (Fig. 3D) and a strong trend towards increased compulsive/manic-like behaviour (Suppl. Fig. 1F). Therefore, our results suggest that miR-708 overexpression not solely impacts memory, but also other behaviours related to bipolar disorder. Nevertheless, we will provide further literature describing the well-known association between memory deficits and bipolar disorder in the revised manuscript.

3. In the introduction section, expected behaviors in animal models for MD-associated behavioral endophenotypes (similar to lines 426 and 642) and potential differences for bipolar versus depression animal models should be better explained. As is, Figure 3 does not provide convincing experimental evidence for miR-708 role in MD-associated behaviors, where the data in Figure 3D presents that animals with upregulated miR-708 show less immobility, indicative of reduced despair-like behavior compared to control mice and no difference in anhedonia or in anxiety. As such, the statement in the abstract (line 41) that there is a causal role cannot be made. Additionally, the use of female mice in Figure 3 but not female rats in Figure 2 is unclear.

- According to the suggestion of this reviewer, we will now better explain which animal behaviours can be expected in models of unipolar vs. bipolar depression. In particular, we will mention that reduced behavioural despair and increased compulsive behaviour, as observed in miR-708 overexpressing mice, have been repeatedly found in mice modeling the manic component of bipolar disorder and provide corresponding references. Although miR-708 overexpressing mice clearly show several endophenotypes related to bipolar disorder (reduced despair, increased compulsivity, impaired short-term memory), we will tone down our statement in the abstract to: “ectopic overexpression of miR-708-5p in the hippocampus of adult male mice is sufficient to elicit MD-associated behavioural endophenotypes, suggesting that elevation of miR-708 in the brain might have a causal role in important aspects of MD”

4. The focus on the hippocampus is not well described, as authors state in line 650 that their approach is contrary to most models reported in the literature. The study does not appear to differentiate between the different subsections of the hippocampus, which are well known to have different functional roles. Method description is often lacking, for example, it is not clear what hippocampal section punches were taken for subsequent sequencing experiment in Figure 4. The sex and number of animals injected with either hp708 and hpCTL and used for sequencing is not indicated anywhere.

- The original focus on the hippocampus was due to its high relevance for emotional control, including the stress response and memory, as has been shown by multiple studies (corresponding references will be provided). Due to the high sample numbers which had to be processed for the different conditions, we were not able to further differentiate between hippocampal subsections (e.g., dorsal vs. ventral). We apologize for the lack of methodological descriptions and will introduce them in the revised manuscript.

5. The introduction of miR-499a-5p in Figure 6F & G comes as a surprise.

- miR-499-5p is not the main focus of this study, but was extensively characterized in a previous paper (Martins et al., EMBO Reports, 2022). However, due to its known role in

Revision Plan

BD, we included it to further improve the diagnostic discrimination between bipolar and unipolar depression. We are happy to include a more thorough description of miR-499-5p in the revised manuscript if needed.

6. Line 605 refers to a literature source that identified miR-708 upregulation in the PFC in rats susceptible to stress-induced depressive like behaviors, however the authors do not consider PFC or provide the reason for not pursuing miR-708 expression in the PFC in the manuscript.

- for reasons outlined under 4., we focused on the hippocampus in the current study. However, we will acknowledge that the PFC is a brain region equally important in the context of MD in the revised manuscript. In fact, interactions between the PFC, hippocampus and amygdala play a major role in important aspects of MD, such as the discrimination between dangerous and safe cues. Therefore, including the PFC in future studies would be highly informative, and we will discuss this in the revised manuscript.

7. The manuscript needs an explanation why mononuclear cells instead of other blood fractions was used. The link between altered levels of miR-708 in the PBMCs and the hippocampus is not clear.

- PBMCs represent a very reliable source of microRNAs, and the correlation between brain expression and PBMCs is often higher as between brain and plasma or serum (e.g, Kos et al., Front. Mol. Neurosci. 2022). The exact link between altered miR-708 levels in brain cells and PBMCs is currently unknown, but one potential route could involve the release of miR-708 encapsulated in extracellular vesicles (EVs) from neurons into the bloodstream, which are subsequently taken up by PBMCs. Crossing of EVs through the BBB and subsequent uptake by monocytes has been reported in the literature. However, describing the full mechanisms of miR-708 regulation in different cell types represents a completely new research avenue and is beyond the scope of the current paper.

Minor comments:

- o In addition to the information in the legends, the figures themselves need to indicate the sex or other information that the graphs are presenting.
- We will indicate sex on the respective graphs (see also our comment to reviewer 1)
 - o Fig 1D would benefit from inclusion of another microRNA expression in ER and GR, without the need for a qPCR, in addition to sequencing expression levels.
- We have already interrogated additional microRNAs and will provide the data in the revised manuscript.
 - o For Figure 1, why are the FDR adjusted pvalues not included as part of the tables? Additionally, the top DE miRNAs such as miR-100 and miR-412 are not looked at or explained why they haven't been investigated further, given the stated unbiased approach objective.
- We will provide the FDR values in the tables. As already indicated above, we will furthermore discuss a potential function of other differentially expressed miRNAs from the screen, such as miR-100 and miR-412. The reasons for focusing on miR-708-5p have already been stated in the results part and include its known regulation by cellular stress and its known genetic association with bipolar disorder.

Revision Plan

- o For Figure 2, since the sex of the animal is not consistent throughout, need to add a symbol next to the graph. For line 356 there is a jump from looking at the expression in the whole brain to focusing on the hippocampus, need a clear justification at the beginning (e.g. lines 388-390).
- We will add sex symbols to all relevant panels and provide a justification for the focus on the hippocampus (see also our comments on reviewer 1).
- o Fig 2F would benefit from another brain region and another miR as controls.
- We have already measured other unrelated microRNAs in rats and will include this data in the revised paper. Regarding different brain regions, we did not collect them from the rat GxE cohort and are therefore unable to provide this additional data. Generating a new cohort simply for collecting additional tissue would constitute a high burden with regard to animal welfare, finances and time. This in our view outweighs the expected added value from such experiments, and we therefore decided not to perform additional animal experiments.
- Did not explain why double hit in genetic and environmental risk factors seem to cancel each other. The study does not show differences between genetic versus environmental risks in humans or rodent models, perhaps this can be addressed in the discussion.
- In our opinion, the animal and human data are highly consistent. In both models, genetic and environmental risk for developing MD lead to increased miR-708-5p expression (cf. Fig. 1D and 2F). The combination of both GxE factors could only be performed in the rat model (condition Canca1c/social isolation), due to the limited number of available participants with both a genetic and environmental predisposition. Contrary to the comment of the reviewer, GxE risk factors do not cancel each other in the rat model, but lead to miR-708-5p levels comparable to the G and E conditions alone (Fig. 2F, fourth bar). Such a non-additive effect would be expected if both G and E risk factors impinge on the same pathway, e.g., intracellular calcium homeostasis.
- o Need to expand abbreviation at first use, such as in lines 56, 72-74
- Abbreviations will be explained at first use.
- o Line 316, "miRNAs causally involved in MD etiology" is not a factual statement and should be toned down, especially given that this section describes "healthy" at-risk participants.
- We agree with the comment of the reviewer and will tone down this statement.
- o Have the authors considered replicating the upregulated levels of miR-708 in the blood of rats, similar to humans? Association between blood and hippocampal/brain levels would add a stronger argument for the methodological approach.
- We agree that correlating brain and blood miR-708-5p levels in the rat model would be powerful. However, obtaining blood samples from rats would require the generation of large additional animal cohorts, which in our view are beyond the scope of the present study (see also our comments above).
- o Figure 3C does not normalize to control, this needs to be re-graphed. Why are there several different colors for individual animal dots?
- The values for miR-708-5p are presented relative to the U6 housekeeping RNA, so we do not see the necessity to further normalize this data. The different colors represent animals of the two sexes for each condition, we will clarify this in the respective figure legend.
- o Figure 3G, the inconsistency in not presenting data for females should be addressed.
- Please see our comment on Reviewer 1 regarding this issue.
- o Line 448 refers to a published single-cell RNA seq data which is lacking a reference

Revision Plan

- **We will include the missing reference.**
 - o Is there a literature source that discusses the connection between the D2 attention test in humans and the object recognition test in rodents (line 523 with 529)? Is there support between D2 test and bipolar disorder? Supplying these references in the introduction would be beneficial to introduce the focus of the experiments pursued here.
- **We are not aware of any literature source discussing a link between the D2 attention and the object recognition test. However, there are multiple studies describing impaired performance of bipolar disorder patients in the D2 attention test. We will provide the respective references in the introduction of the revised manuscript.**
 - o The object recognition involves 5 minutes interval and a 24 hours interval, it is not clear whether the same animal is exposed to both intervals and the same "novel" object? In which case, could it be that the repeated exposure leads to the inconclusive results in females?
- **Animals of two independent cohorts were used for the short-term (5min) or long-term (24h) novel object recognition memory tests. Therefore, we can rule out that the results were confounded due to repeated exposure to the objects. We will add this information to the revised version of the manuscript.**
 - o Fig 6A - the D2 task alone is not sufficient, need similar test and unrelated measure to confidently conclude the association.
 - o Some grammatical editing is needed, for example lines 386, 387, 717
- **We have fixed these grammatical errors.**
 - o The methods section needs reorganization as some descriptives are not mentioned at first use as in the results (e.g. line 146 needs to expand on the sequencing details because the link is insufficient), and the division between "human study" and "animal study" are not consistent (e.g. rat culture method description under human section).
- **We will re-organize the methods section according to the suggestions of the reviewer.**

Reviewer #2 (Significance (Required)):

The introduction presents a compelling objective: an unbiased, back-translational approach to identify microRNAs associated with mood disorders from a large cohort of healthy individuals at high genetic and environmental risk. However, the results do not align with this objective. The authors do not demonstrate an unbiased approach to identifying target miRNAs, and the analysis falls short of showcasing risk prediction in non-diagnosed individuals. Instead, the study begins with at-risk individuals and concludes by demonstrating diagnostic ability in already diagnosed individuals, which is counterintuitive. Furthermore, the microRNA previously implicated in bipolar disorder by the authors is only mentioned at the end, without clarifying how this study relates to their prior findings. Addressing these concerns will significantly strengthen the interpretation of the results and their significance for the field.

- We agree with the reviewer that our main objective was to identify microRNAs associated with mood disorders using a back-translational approach. However, we want to stress that this was performed in a highly unbiased manner by small RNA-sequencing from blood samples of healthy human subjects with a high risk for mood disorders. Our approach led to the identification of GxE regulated microRNAs, one of which was subsequently shown to regulate BD-associated behaviour in mice and, consistently, to be upregulated in BD patients. In contrast, risk prediction in non-diagnosed individuals would require a longitudinal monitoring of probands over several years and was therefore not the focus of the current study. With regard to the other BD-associated miRNA, miR-499-5p, we are happy to provide a more in-depth discussion of our previous paper (Martins et al., EMBO Rep. 2022) in the introduction of the revised manuscript.

3. Description of the revisions that have already been incorporated in the transferred manuscript

Not applicable.

4. Description of analyses that authors prefer not to carry out

Reviewer 1:

“The major concern with this study is the lack of follow-up functional characterization of the identified miR-708-5p target, Nnat, in the rodent model, particularly its role in calcium homeostasis processes.” As outlined above, we feel that a detailed mechanistic characterization of Nnat function in calcium homeostasis in neurons is clearly beyond the scope of the present study. First, the main focus of the paper is on the role of miR-708-5p in behavioural endophenotypes related to mood disorder, as well as its potential use as a biomarker for differential diagnosis. In this regard, we have already provided data for the functional relevance of miR-708-5p dependent regulation of Nnat. Second, our lab is currently not set up to perform high-resolution calcium imaging experiments in the neuronal cytosol and ER. The time required for Establishing and performing such experiments will significantly exceed 2-3 months which is usually allocated to a paper revision.

Reviewer 2: “The study jumps inconsistently from different species, sex, and models. There is also a need for additional control conditions, such as including another microRNA, brain region and human behavioral scores, which are essential in presenting results that are not fitted to a desired outcome.” We have now explained the rationales for using different species, sex, and models in our experiments. The only major issue we are not able to address is the inclusion of another brain region for our animal experiments. While we clearly see the advantage of measuring miR-708-5p levels in different brain regions and the blood, there is currently neither additional brain tissue nor blood samples available, which means that entirely new animal cohorts (mice, rats) would have to be generated to obtain these samples. This would likely take

Revision Plan

several months, especially since we and our collaborators working with rats (M. Wöhr, Leuven) would have to file entirely new animal licenses. In addition, generating large new animal cohorts solely for tissue collection (brain, blood), especially those with a high severity degree (e.g., juvenile social isolation in rats), would not be compliant with the RRR principles of animal research.

Dear Gerhard,

Thank you for the submission of your manuscript with referee reports and revision plan to EMBO reports. I have looked at all files now and agree that your study is interesting and a good fit for our journal, and I also agree with your revision plan.

I am thus happy to invite you to revise your manuscript with the understanding that the referee concerns must be addressed and their suggestions taken on board. Please address all referee concerns in a complete point-by-point response. Acceptance of the manuscript will depend on a positive outcome of a second round of review. It is EMBO reports policy to allow a single round of major revision only and acceptance or rejection of the manuscript will therefore depend on the completeness of your responses included in the next, final version of the manuscript.

I understand that no further major experimentation is required for the revisions and I would thus like to suggest that you re-submit your manuscript as soon as possible.

- 1) A data availability section providing access to data deposited in public databases is missing. If you have not deposited any data, please add a sentence to the data availability section that explains that.
- 2) Your manuscript contains statistics and error bars based on $n=2$. Please use scatter blots in these cases. No statistics should be calculated if $n=2$.

5) a complete author checklist, which you can download from our author guidelines . Please insert information in the checklist that is also reflected in the manuscript. The completed author checklist will also be part of the RPF.

6) Please note that all corresponding authors are required to supply an ORCID ID for their name upon submission of a revised manuscript (. Please find instructions on how to link your ORCID ID to your account in our manuscript tracking system in our Author guidelines

EMBO reports

- the name of the statistical test used to generate error bars and P values,
- the number (n) of independent experiments (please specify technical or biological replicates) underlying each data point,
- the nature of the bars and error bars (s.d., s.e.m.),
- If the data are obtained from n Program fragment delivered error ``Can't locate object method "less" via package "than" (perhaps you forgot to load "than"?) at //ejpvfs23/sites23b/embor_www/letters/embor_decision_rc_revise_and_rereview.txt line 56.' 2, use scatter blots showing the individual data points.

12) All Materials and Methods need to be described in the main text using our 'Structured Methods' format, which is required for all research articles. According to this format, the Methods section includes a separate file Reagents and Tools Table (listing key reagents, experimental models, software and relevant equipment and including their sources and relevant identifiers) and a Methods and Protocols section describing the methods using a step-by-step protocol format. The aim is to facilitate adoption of the methodologies across labs. More information on how to adhere to this format as well as a downloadable template (.docx) for the Reagents and Tools Table can be found in our author guidelines:

An example of a Method paper with Structured Methods can be found here: <https://www.embopress.org/doi/full/10.1038/s44320-024-00037-6#sec-4>

I look forward to seeing a revised form of your manuscript when it is ready.

Point-by-point response letter

Reviewer #1 (Evidence, reproducibility and clarity (Required)):

Summary:

Mood disorders (MD), including bipolar disorder (BD) and major depressive disorder (MDD), are serious mental health conditions caused by a complex interplay of genetic and environmental risk factors (GxE). However, little is known about how miRNA interactions contribute to these diseases, which limits the use of miRNAs as potential biomarkers for MD diagnosis. The study by Gilardi and colleagues aims to identify MD-associated mature miRNAs (miRs) from peripheral blood mononuclear cells (PBMCs) in a large human cohort of healthy individuals at high GxE risk for developing MD, using small RNA sequencing (RNA-seq). This unbiased, back-translational approach led to the identification of BD-associated miR-708-5p, which was further validated using luciferase and qPCR assays. The functional role of miR-708-5p was characterized at the molecular and behavioral levels in vivo and in vitro in rodent models through viral overexpression in the mouse hippocampus. Finally, the diagnostic potential of miR-708-5p was assessed in MD patient subgroups.

Major comments: The manuscript is well-structured, and the findings presented here provide translational value for MD-associated miRNAs. The identified candidate, miR-708-5p, and its predicted binding target, the endoplasmic reticulum (ER) resident protein Neuronatin (Nnat), were carefully validated and confirmed as direct targets of miR-708-5p. Furthermore, comprehensive behavioral assays were conducted in mice overexpressing miR-708-5p in the hippocampus using AAV constructs, demonstrating a causal and robust MD-associated behavioral phenotype. Notably, the authors elegantly showed that by using an engineered viral construct that simultaneously overexpresses both Nnat and miR-708, they were able to restore Nnat expression in the hippocampus of miR-708-5p overexpressing adult mice, rescuing BD-associated behavioral phenotypes.

- We are happy that this reviewer emphasizes the high translational value of our study and further highlights the significance of our target validation and mouse behavioural approach

The major concern with this study is the lack of follow-up functional characterization of the identified miR-708-5p target, Nnat, in the rodent model, particularly its role in calcium homeostasis processes. Additionally, there is no evidence provided to show that this gene is upregulated in patients suffering from BD. It would be valuable to explore whether aberrant miR-708/Nnat signaling impacts intraneuronal calcium homeostasis in this psychiatric disorder, as this could have implications for developing new therapeutic interventions. However, such an exploration may be beyond the scope of the present study, which aims primarily to identify novel potential biomarkers for MD.

- We agree with this reviewer that in this study, we have not explored the molecular mechanisms downstream of Neuronatin, e.g., the role of miR-708/Nnat signaling in intraneuronal calcium homeostasis processes. However, as correctly pointed out by this reviewer, the major emphasis of this paper was on the identification of microRNAs as potential biomarkers for human mood disorders. In addition, we have provided evidence for a functional involvement of the miR-708/Nnat pathway in mood-disorder associated behavioural endophenotypes in mice. Therefore, further

mechanistic studies on Nnat in neuronal calcium homeostasis are beyond the scope of the current study and will form the basis for a separate manuscript.

Minor Comments:

The manuscript should be carefully checked for reference formatting, as there are instances of mixed formats (numbers and names) and missing details in the figure legends that should be consistent with the figure titles used in the manuscript. Variations between the figure titles and the corresponding manuscript text should be addressed.

- In the revised version, we have fixed all issues related to reference formatting and figure legends/titles.

Introduction: The behavioral assays mentioned in the introduction (e.g., lines 72-73) should be spelled out in full before using abbreviations, as not every reader may have a behavioral background.

- We have spelled out all terms before using abbreviations in the revised paper.

Supplementary Materials and Methods: The age of the 3-month-old mice used for stereotactic surgeries (see page 2) does not align with the 7/8-week-old mice mentioned in the main text. Additionally, in the Statistical Analysis section, the Y-maze is referenced but not shown in the manuscript.

- Mice were usually 2 months old at the time of surgery. We have checked for consistency in the revised manuscript.

Results: During the unbiased small RNA sequencing assay, six mature miRNAs were identified in the GxE groups. While it is reasonable to focus on miR-708-5p due to its recent association with BD, it would be interesting to briefly discuss the other miRNA candidates given the unbiased nature of the approach. Is there literature or in silico analysis (e.g., miRNA-related GO terms) that could be relevant to share? For better clarity, I would recommend highlighting miR-708 in the tables shown in Fig. 1B and C to provide quick orientation for the reader.

- We have included available references for two other candidate GxE miRNAs, miR-100-5p and miR-501-3p, identified in the small RNA sequencing screen in the results part (p.5). Furthermore, we have highlighted miR-708 in the tables (Fig. 1B, D) for better clarity.

Fig. 2E: Since other behavioral panels include a sex symbol/icon to identify the sex of the animals used, it would be consistent to include the male icon here as well. The same applies to panels in Fig. 5D and E.

- We have included the sex symbols in the revised manuscript when appropriate.

Fig. 3: Is there a reason why females were not tested in the NOR assay after a 24-hour break? Perhaps panel 3G could be moved to the supplements to maintain consistency, as males and females are otherwise tested in the other assays shown.

- Females were not tested in the 24-hour break paradigm since in the initially used 5-min short term memory paradigm (Fig. 3H, I), males displayed a much stronger

memory impairment compared to females. For consistency, we have moved the original Fig. 3G to the expanded view (EV2) of the revised manuscript. It would be helpful to edit the schematic panel in Fig. 3A to add "qPCR" at the timeline's end, indicating when animals were measured for miR-708-5p expression levels in the hippocampus.

- We have edited the schematic panel as suggested by the reviewer.

I would suggest moving Fig. 3C to the end, as the qPCR was performed after the behavioral assays were completed.

- We have moved original Fig. 3C to the end of Fig. 3 (new Fig. 3H) as suggested by the reviewer.

Fig. 3B: Are higher magnification images from the injected hippocampi available, including additional neuronal and astrocyte markers (e.g., NeuN, GFAP), to demonstrate in which hippocampal cells the virus is expressed?

- Unfortunately, we do not have any higher magnification images of the injected hippocampi. Experiments would have to be repeated to obtain such images, which in our view is beyond the scope of this study.

I would recommend moving the WB panel in Fig. 4H to the supplements or focusing on a smaller representative area (e.g., the first four lines) to demonstrate the downregulation of the Nnat protein.

- As suggested by the reviewer, we have focused on a smaller representative area of the WB in the main figure (new Fig. 4E) and show the entire blot in the expanded view (EV3C) of the revised manuscript.

I am not convinced of the relevance of the Fig. 6E panel in this study, which primarily deals with MD, as miR-708-5p expression levels in CTL and schizophrenia (SCZ) patient PBMCs were analyzed. Given the high genetic similarities between SCZ and BD, the sample size for SCZ is on the lower end (see females CTL n=8 vs. SCZ n=8), particularly if the data were to be presented separately by sex as done with the MD patients (see Fig. 6C-D). Therefore, I am not entirely convinced that miR-708-5p is unaffected in SCZ patients; the sample size may simply not be sufficient to detect a difference in expression levels, especially considering the heterogeneity of SCZ. I suggest moving panel Fig. 6E to the supplements and briefly discussing the lack of findings in SCZ patients in the main text.

- We feel strongly that showing the data on SCZ is important, since it nicely demonstrates the specificity of the miR-708-5p upregulation in BD compared to the genetically similar SCZ. However, we also agree with this reviewer that the data on SCZ might be slightly underpowered to make strong statements. Therefore, as suggested by the reviewer, we will move the SCZ data to the expanded view (EV4) of the revised manuscript.

Supplemental Fig. 1C-E: It is unclear from the graph or figure legend what these three graphs specifically refer to. Are the X-axis values animal numbers, and do graphs C-E represent different cohorts used for the qPCR assay? Additionally, in Suppl. Fig. 2J, please clarify the difference between this group of animals and the one shown in Suppl. Fig. 2H in the figure legend description.

- We have clarified the axis labels in the revised manuscript and have merged the graphs containing data from male and female mice (new Annex Fig. S2D-E).

Reviewer #1 (Significance (Required)):

Significance: This translational study is well-structured and meticulously conducted, demonstrating the role of miR-708-5p in regulating MD-associated behavioral phenotypes in rodents. It represents a first step toward deepening our understanding of the specific functions of miR-708-5p in mood disorders (MD), its potential as a diagnostic biomarker, and insights into biological mechanisms that may offer translational value for MD therapeutics in the future. Overall, the study is informative and valuable, particularly for scientists in translational and psychiatric research who seek to develop novel diagnostic biomarkers for bipolar disorder (BD).

Audience: The study by Gilardi et al. will be of great interest to scientists from both basic and translational/clinical neuroscience.

Own Field of Expertise: Neurodevelopmental and psychiatric diseases, genetics, behavior, miRNA.

Reviewer #2 (Evidence, reproducibility and clarity (Required)):

Summary:

To address a critical gap in identifying potential biomarkers detectable in blood for individuals predisposed to psychiatric conditions, the current study analyzed blood samples from individuals at risk for and diagnosed with mood disorders, with molecular characterization in rodent models and in vitro.

Authors identify elevated miR-708 levels in the blood of female participants at high risk for mood disorders due to environmental and genetic factors. In female rats, miR-708 expression was identified to be expressed in different brain regions. The study focuses on miR-708 expression in the hippocampus, localizing the expression to the neurons and altered levels in male rat model for environmental or genetic risk for behavioral abnormalities.

Given the upregulation of miR-708 in both rat model and humans at risk, miR-708 was upregulated with viral vectors in adult mouse hippocampus and subsequently tested for differences on a behavioral level. In the tail-suspension test male mice with high hippocampal levels of miR-708 showed reduced immobility and lack of novel object exploration compared to familiar, whereas in female mice the test could be conclusive when tested within the same test session with a 5-minute interval.

Differential expression analysis of high-throughput gene expression measure in the hippocampus between mice with control or virally upregulated levels of miR-708 identified neuronatin gene to be altered and contain the matching sequence for miR-708. In vitro study confirmed the direct regulation of neuronatin RNA and protein by miR-708. In male mice with modulated miR-708 dependent neuronatin downregulation showed neuronatin-level dependent novel object exploration.

Applying the principle of miR-708 involvement in molecular pathways and behavioral abnormalities in animals, blood levels of miR-708 were measured in clinical depression and bipolar disorder populations. A negative association between D2 attention scores and miR-708 levels was observed in the full cohort. Greater upregulated levels of miR-708 in blood were observed for the group with bipolar disorder diagnosis in males and females. Furthermore, a combination of miR-708 and a previously identified microRNA involved in bipolar disorder, miR-499a, showed

high sensitivity and specificity of differentiating individuals with bipolar disorder and healthy controls, with better discrimination in males.

This study is commendable for its ambitious approach and extensive efforts, particularly in the challenging field of microRNAs as biomarkers in psychiatric research. However, despite the authors' endeavors, the presentation of data and results have several major caveats that undermine confidence in their conclusions. The study jumps inconsistently from different species, sex, and models. There is also a need for additional control conditions, such as including another microRNA, brain region and human behavioral scores, which are essential in presenting results that are not fitted to a desired outcome.

- We are happy that this reviewer acknowledges our ambitious approach and extensive efforts towards microRNA biomarker discovery in psychiatric research. As outlined in more detail below, we are further confident that we have addressed in full all the remaining concerns of this reviewer with respect to the selection of species, sex, models and brain regions in our revised manuscript.

Major comments:

1. The study describes the availability of both male and female participants' blood, however the analysis was only pursued in females (lines 326 to 328). Based on the table provided, it is not clear that the sample composition is better for females and thus serves as a strong justification for differential expression solely based on females.

- For our RNA sequencing screen, we used healthy individuals with high genetic or environmental risk to develop MD. For these groups (in contrast to the BD and MDD patient groups), it was in fact impossible to form a large, homogenous subgroup for males (suppl. Table 1 only provides the data for female subjects!). This was due to the presence of many confounding factors (e.g., smoking) in the male subjects which led to their exclusion from the study.

2. In line 78 social communication is used to describe mood disorder associated endophenotypes, which is a seemingly critical measure not included in the study. Justification for the measure of solely memory and attention aspect needs to be strengthened. The link between *Cacna1c*, hippocampus and memory has been previously described in the literature, however the relation between memory deficits and bipolar disorder as the main feature of the endophenotype is missing clarity in the manuscript.

- We would like to point out that, in addition to the memory and attention aspects mentioned by this reviewer, we have also interrogated behavioural despair (using TST), anxiety (EPM, open field), anhedonia (sucrose preference) and compulsive/manic behaviour (using marble burying) in our comprehensive behavioural test battery. Upon miR-708-5p, we observed significant changes in memory (Fig. 3F-I), but also reduced behavioural despair (Fig. 3D) and a strong trend towards increased compulsive/manic-like behaviour (Suppl. Fig. 1F). Therefore, our results suggest that miR-708 overexpression not solely impacts memory, but also other behaviours related to bipolar disorder. Nevertheless, we have now provided further literature describing the well-known association between memory deficits and bipolar disorder in the discussion of the revised manuscript (p.14-15).

3. In the introduction section, expected behaviors in animal models for MD-associated behavioral endophenotypes (similar to lines 426 and 642) and potential differences for

bipolar versus depression animal models should be better explained. As is, Figure 3 does not provide convincing experimental evidence for miR-708 role in MD-associated behaviors, where the data in Figure 3D presents that animals with upregulated miR-708 show less immobility, indicative of reduced despair-like behavior compared to control mice and no difference in anhedonia or in anxiety. As such, the statement in the abstract (line 41) that there is a causal role cannot be made. Additionally, the use of female mice in Figure 3 but not female rats in Figure 2 is unclear.

- According to the suggestion of this reviewer, we have now better explained which animal behaviours can be expected in models of unipolar vs. bipolar depression. In particular, we have mentioned that reduced behavioural despair and increased compulsive behaviour, as observed in miR-708 overexpressing mice, have been repeatedly found in mice modeling the manic component of bipolar disorder and provide corresponding references. Although miR-708 overexpressing mice clearly show several endophenotypes related to bipolar disorder (reduced despair, increased compulsivity, impaired short-term memory), we have toned down our statement in the abstract regarding a causal role of miR-708-5p as suggested by this reviewer.

4. The focus on the hippocampus is not well described, as authors state in line 650 that their approach is contrary to most models reported in the literature. The study does not appear to differentiate between the different subsections of the hippocampus, which are well known to have different functional roles. Method description is often lacking, for example, it is not clear what hippocampal section punches were taken for subsequent sequencing experiment in Figure 4. The sex and number of animals injected with either hp708 and hpCTL and used for sequencing is not indicated anywhere.

- The original focus on the hippocampus was due to its high relevance for emotional control, including the stress response and memory, as has been shown by multiple studies (corresponding references will be provided). Due to the high sample numbers which had to be processed for the different conditions, we were not able to further differentiate between hippocampal subsections (e.g., dorsal vs. ventral). We apologize for the lack of methodological descriptions and have now introduced them in the revised manuscript to the methods section and figure legends.

5. The introduction of miR-499a-5p in Figure 6F & G comes as a surprise.

- miR-499-5p is not the main focus of this study, but was extensively characterized in a previous paper (Martins et al., EMBO Reports, 2022). However, due to its known role in BD, we included it to further improve the diagnostic discrimination between bipolar and unipolar depression. We have now added additional information about miR-499-5p in the introduction of the revised paper (p.4).

6. Line 605 refers to a literature source that identified miR-708 upregulation in the PFC in rats susceptible to stress-induced depressive like behaviors, however the authors do not consider PFC or provide the reason for not pursuing miR-708 expression in the PFC in the manuscript.

- for reasons outlined under 4., we focused on the hippocampus in the current study. However, we acknowledge that the PFC is a brain region equally important in the context of MD in the revised manuscript. In fact, interactions between the PFC,

hippocampus and amygdala play a major role in important aspects of MD, such as the discrimination between dangerous and safe cues. Therefore, including the PFC in future studies would be highly informative, and we have now discussed this in the revised manuscript (p. 15).

7. The manuscript needs an explanation why mononuclear cells instead of other blood fractions was used. The link between altered levels of miR-708 in the PBMCs and the hippocampus is not clear.

- PBMCs represent a very reliable source of microRNAs, and the correlation between brain expression and PBMCs is often higher as between brain and plasma or serum (e.g, Kos et al., *Front. Mol. Neurosci.* 2022). The exact link between altered miR-708 levels in brain cells and PBMCs is currently unknown, but one potential route could involve the release of miR-708 encapsulated in extracellular vesicles (EVs) from neurons into the bloodstream, which are subsequently taken up by PBMCs. Crossing of EVs through the BBB and subsequent uptake by monocytes has been reported in the literature. However, describing the full mechanisms of miR-708 regulation in different cell types represents a completely new research avenue and is beyond the scope of the current paper.

Minor comments:

o In addition to the information in the legends, the figures themselves need to indicate the sex or other information that the graphs are presenting.

- We have indicated sex on the respective graphs (see also our comment to reviewer 1)

o Fig 1D would benefit from inclusion of another microRNA expression in ER and GR, without the need for a qPCR, in addition to sequencing expression levels.

- We have now interrogated three additional microRNAs (miR-16, miR-30a, miR-1248) by qPCR and found that none of these significantly differed in their expression levels between control and ER subjects (new Fig. EV1).

For Figure 1, why are the FDR adjusted p-values not included as part of the tables? Additionally, the top DE miRNAs such as miR-100 and miR-412 are not looked at or explained why they haven't been investigated further, given the stated unbiased approach objective.

- For the RNA-sequencing experiments, we had to use a relaxed cut-off ($p < 0.05$ instead of $FDR < 0.05$) due to the high variability of human PBMC expression data. We therefore provided p-values instead of FDR in Figs. 1C and 1E. Furthermore, we provide a rationale for our focusing on miR-708-5p, based on its previous association with bipolar disorder and stress (p. 5). We included two additional references for two of the top DE miRNAs (miR-100-5p, miR-501-3p).

o For Figure 2, since the sex of the animal is not consistent throughout, need to add a symbol next to the graph. For line 356 there is a jump from looking at the expression in the whole brain to focusing on the hippocampus, need a clear justification at the beginning (e.g. lines 388-390).

- We have added sex symbols to all relevant panels and provide a justification for the focus on the hippocampus (see also our comments on reviewer 1).

o Fig 2F would benefit from another brain region and another miR as controls.

- We have now measured an unrelated microRNA (miR-129-5p) in the hippocampus of the original rat GxE cohort and included this data in the revised paper (new Fig. 2F). Regarding different brain regions, we did not collect them from the rat GxE cohort and are therefore unable to provide this additional data. Generating a new cohort simply for collecting additional tissue would constitute a high burden with regard to animal welfare, finances and time. This in our view outweighs the expected added value from such experiments, and we therefore decided not to perform additional animal experiments.

Did not explain why double hit in genetic and environmental risk factors seem to cancel each other. The study does not show differences between genetic versus environmental risks in humans or rodent models, perhaps this can be addressed in the discussion.

- In our opinion, the animal and human data are highly consistent. In both models, genetic and environmental risk for developing MD lead to increased miR-708-5p expression (cf. Fig. 1F and 2E). The combination of both GxE factors could only be performed in the rat model (condition Canca1c/social isolation), due to the limited number of available participants with both a genetic and environmental predisposition. Contrary to the comment of the reviewer, GxE risk factors do not cancel each other in the rat model, but lead to miR-708-5p levels comparable to the G and E conditions alone (Fig. 2E, fourth bar). Such a non-additive effect would be expected if both G and E risk factors impinge on the same pathway, e.g., intracellular calcium homeostasis.

o Need to expand abbreviation at first use, such as in lines 56, 72-74

- Abbreviations have now been explained at first use.

o Line 316, "miRNAs causally involved in MD etiology" is not a factual statement and should be toned down, especially given that this section describes "healthy" at-risk participants.

- We agree with the comment of the reviewer and have now toned down this statement to: "We hypothesized that miRNAs whose expression correlates with environmental (ER) and genetic risk (GR) for MD in human subjects might be good candidates for molecules with a functional role in MD pathophysiology." (p. 5).

o Have the authors considered replicating the upregulated levels of miR-708 in the blood of rats, similar to humans? Association between blood and hippocampal/brain levels would add a stronger argument for the methodological approach.

- We agree that correlating brain and blood miR-708-5p levels in the rat model would be powerful. However, obtaining blood samples from rats would require the generation of large additional animal cohorts, which in our view are beyond the scope of the present study (see also our comments above).

o Figure 3C does not normalize to control, this needs to be re-graphed. Why are there several different colors for individual animal dots?

- In the original figure 3C (new Fig. 3D), The values for miR-708-5p are presented relative to the U6 housekeeping RNA, so we do not see the necessity to further normalize this data. The different colors in the original figure represented animals derived from different cohorts. To avoid further confusion, we have now omitted

these labels in the new Fig. 3D.

o Figure 3G, the inconsistency in not presenting data for females should be addressed.

➤ Please see our comment on Reviewer 1 regarding this issue.

o Line 448 refers to a published single-cell RNA seq data which is lacking a reference

➤ We have now included the missing reference.

o Is there a literature source that discusses the connection between the D2 attention test in humans and the object recognition test in rodents (line 523 with 529)? Is there support between D2 test and bipolar disorder? Supplying these references in the introduction would be beneficial to introduce the focus of the experiments pursued here.

➤ We are not aware of any literature source discussing a link between the D2 attention and the object recognition test. However, there are multiple studies describing impaired performance of bipolar disorder patients in the D2 attention test. We have added respective references in the revised manuscript on p. 15 (e.g., Carnelo et al., 2017).

o The object recognition involves 5 minutes interval and a 24 hours interval, it is not clear whether the same animal is exposed to both intervals and the same "novel" object? In which case, could it be that the repeated exposure leads to the inconclusive results in females?

➤ Animals of two independent cohorts were used for the short-term (5min) or long-term (24h) novel object recognition memory tests. Therefore, we can rule out that the results were confounded due to repeated exposure to the objects. We have now added this information to the methods section of the revised manuscript.

o Fig 6A - the D2 task alone is not sufficient, need similar test and unrelated measure to confidently conclude the association.

➤ A standard psychological test battery was performed on the large FOR2107 cohort, which did not include any additional tests similar to the D2 task. Since these tests were done sometimes many years ago, recruiting back the same individuals for performing additional tests would be rather challenging and beyond the scope of this manuscript.

o Some grammatical editing is needed, for example lines 386, 387, 717

➤ We have fixed these grammatical errors.

o The methods section needs reorganization as some descriptives are not mentioned at first use as in the results (e.g. line 146 needs to expand on the sequencing details because the link is insufficient), and the division between "human study" and "animal study" are not consistent (e.g. rat culture method description under human section).

➤ We have re-organized the methods section according to the suggestions of the reviewer.

Reviewer #2 (Significance (Required)):

The introduction presents a compelling objective: an unbiased, back-translational approach to identify microRNAs associated with mood disorders from a large cohort of healthy individuals at high genetic and environmental risk. However, the results do not align with this objective. The authors do not demonstrate an unbiased approach to identifying target miRNAs, and the analysis falls short of showcasing risk prediction in non-diagnosed individuals. Instead, the study begins with at-risk individuals and concludes by demonstrating diagnostic ability in already diagnosed individuals, which is counterintuitive. Furthermore, the microRNA previously implicated in bipolar disorder by the authors is only mentioned at the end, without clarifying how this study relates to their prior findings. Addressing these concerns will significantly strengthen the interpretation of the results and their significance for the field.

➤ We agree with the reviewer that our main objective was to identify microRNAs associated with mood disorders using a back-translational approach. However, we want to stress that this was performed in a highly unbiased manner by small RNA-sequencing from blood samples of healthy human subjects with a high risk for mood disorders. Our approach led to the identification of GxE regulated microRNAs, one of which was subsequently shown to regulate BD-associated behaviour in mice and, consistently, to be upregulated in BD patients. In contrast, risk prediction in non-diagnosed individuals would require a longitudinal monitoring of probands over several years and was therefore not the focus of the current study. With regard to the other BD-associated miRNA, miR-499-5p, we have now provided more information about findings from our previous paper in the introduction of the revised manuscript.

Dear Gerhard,

Thank you for the submission of your revised manuscript. We have now received the enclosed report from referee 1 (prev referee 2). This referee still has some more points that will need to be addressed in the ms text before we can proceed with the official acceptance of your ms.

Please co-submit a final point-by-point response with your final ms files. I do agree with referee 1 that it is important to clearly explain in the main ms text why only female data are shown in Figure 1 and only male data in Figure 2. I also agree that the conclusions based on the data shown in Fig 2E need to be amended or better explained. Please also tone down the conclusions in the last sentence of the abstract.

A few editorial requests will also need to be addressed:

- The DATA AVAILABILITY SECTION needs to be placed after the Methods. The Methods section needs to be called just "Methods". Please provide the specific URLs for GSE261287, GSE261288 datasets in the data availability statement.
- The funding info grant Dan3/022/22 is missing in our online submission system. Please add this when you upload the final ms files.
- Please remove the authors credits from the manuscript text. All credits need to be entered during online ms submission.
- The methods in the APPENDIX file should be added to the main manuscript, if they describe methods relevant for the main ms. If they only apply to the Appendix figures, the methods can stay in the Appendix. However, page numbers need to be added to the table of content of the Appendix file.
- The SOURCE DATA need to be uploaded as one file per main figure; the EV and Appendix Figure source data can stay zipped as one file.
- The REAGENT TABLE should be uploaded as a separate file.
- FIGURE CALLOUTS: Table S1 on p.5 should be corrected to Appendix Table S1 and Fig. S1 etc on p. 7 and following should be Appendix Figure S1 etc.; the panels for Fig EV1 should be called out.
- The BioRender reference should be removed from the Acknowledgements and added as a separate section to the Methods using the following format:
Graphics:
(some of the... OR Figure #... OR synopsis) Graphics were created with BioRender.com.
- Table 1 should be placed between the main figure legends and the expanded view figure legends.
- Please provide the exact p values (as reasonable) in the legends of figures 3D, E, F, H; 5A-D; 6C, D; EV2 D.
- Please indicate the statistical test used for data analysis in the legends of figures 1B, C, D, E; 4A.

I would like to suggest some minor changes to the title and abstract (that needs to be written in present tense). Please let me know whether you agree with the following:

miR-708-5p is elevated in bipolar patients and can induce mood disorder-associated behavior in mice

Mood disorders (MDs) are caused by an interplay of genetic and environmental (GxE) risk factors. However, molecular pathways engaged by GxE risk factors are poorly understood. Using small-RNA sequencing of peripheral blood mononuclear cells (PBMCs), we show that the bipolar disorder (BD)-associated microRNA miR-708-5p is upregulated in healthy human subjects with a high genetic or environmental predisposition for MDs. miR-708-5p is further upregulated in the hippocampus of rats that underwent juvenile social isolation, a model of early life stress. Hippocampal overexpression of miR-708-5p in adult male mice is sufficient to elicit MD-associated behavioural endophenotypes. We further show that miR-708-5p directly targets Neuronatin (Nnat), an endoplasmic reticulum protein. Restoring Nnat expression in the hippocampus of miR-708-5p-overexpressing mice rescues miR-708-5p-dependent behavioral phenotypes. Finally, miR-708-5p is upregulated in PBMCs from patients diagnosed with MD. Peripheral miR-708-5p expression allows to differentiate male BD patients from patients suffering from major depressive disorder (MDD). In summary, we describe a potential functional role for the miR-708-5p/Nnat pathway in MD etiology and identify miR-708-5p as a potential biomarker for the differential diagnosis of MDs.

EMBO press papers are accompanied online by A) a short (1-2 sentences) summary of the findings and their significance, B) 2-3 bullet points highlighting key results and C) a synopsis image that is exactly 550 pixels wide and 200-600 pixels high (the height is variable). The synopsis image should provide a sketch of the major findings, like a graphical abstract. Please note that text needs to be readable at the final size. Please send us this information along with the final manuscript.

Best wishes,
Esther

Referee #1:

Having thoroughly read the manuscript, my suggestion for the authors going forward is to majorly reorganize the main result presentation, prioritizing consistency. The truly strong findings are presented in Figures 4 - 6 and are suggested to be the central focus. The following limitations are hindering the manuscript in its present form:

o Figure 1

- miR-708-5p is not a top hit in the high-throughput analysis and is subsequently chosen on the basis of a priory hypothesis, while the earlier rationale provided is that of an unbiased approach;
- only female subjects from the cohort were considered, which was not a central aim, with the discrepancy not being clearly explained in the methods (comment #1 below);

o Figure 2

- the male rat model with both the genetic risk and environmental hindrance (isolated *Cacna1c*^{+/-} group) does not show differential expression to control;
- no statistical differences are detected in female rats;

o Figure 3

- the study solely investigates miR-708-5p in the hippocampus without indicating a link between them;
- the study suggests that hippocampal overexpression of miR-708-5p may be associated with MD-related behaviors in the mouse model, based on findings that appear to align with an "anti-depressant" function of miR-708-5p, particularly during a manic-like state. However, the results do not provide compelling evidence for a specific link to bipolar disorder (BD). It is possible that the microRNA is enhancing resiliency to stress, potentially at the expense of memory and attention. Additionally, the author's conclusion is not supported by the limited relationship between miR-708-5p and the Young Mania Rating Scale (YMRS) score.

The major comments from original review:

#1 As previously noted, it is unclear whether using only female subjects from the same cohort for analysis in Figure 1 is more appropriate, while sex composition does not appear to be a concern for Figure 6. It would be helpful if the authors could clarify the total number of participants in the cohort after applying the exclusion and selection criteria, and provide subsections with additional details about the subjects described in Table 1 and Supplementary Table 1.

#2-4 When considering bipolar disorder, memory and attention may not be the most prominent deficits that distinguish this psychiatric condition from others. The rationale for focusing on memory/attention, the hippocampus, and the link to this microRNA appears somewhat unclear. For example, in the results section, the transition from observing expression in regions associated with mood disorders and cognitive function, such as the amygdala, frontal cortex, and hippocampus, to solely assessing miR-708-5p expression in rat hippocampal primary neurons introduces a gap. Additionally, while the authors observe effects such as amelioration of despair (as measured by the tail-suspension test) and trending compulsive behavior, these effects are only present in males, which complicates the interpretation of the results that begins with female subjects.

#5 Introducing the link between miR-499-5p and miR-708-5p functions in the hippocampus, for example by manipulating the two microRNAs in the animal model concurrently, would have been a more logical precedent to the ROC curve analysis with two candidates. The characterization of miR-499-5p in the other publication is consistent with the use of the rat model and the hippocampus rationale that extends between the animal model and clinical cohort. In the current manuscript, a lot of inconsistencies are introduced with different species, sex, and models which challenge the cohesiveness of the story.

#7 The choice for mononuclear blood cells suggests that there is either a hypothesized involvement of the immune-related response or altered microRNA biogenesis mechanism in those cells specifically. The author's suggestion of a potential route involving the "release of miR-708 encapsulated in extracellular vesicles (EVs) from neurons into the bloodstream, which are subsequently taken up by PBMCs" seems speculative, given the uncertainty around the likelihood of such an occurrence. This

raises the question of why microRNA expression isn't measured directly in EVs or in whole blood. Regarding the relaxed statistical cut-off in the RNA-sequencing experiment, the authors commented an acknowledgement of the "high variability of human PBMC expression data," which adds to the question of the origin of microRNAs in PBMC fractions. While the characterization of circulating microRNAs is valuable in psychiatric research, the rationale provided for this specific approach lacks clarity.

In reference to Figure 2E, I had previously noted that the combined genetic and environmental risk factors seemed to cancel each other out. The authors responded by stating that "GxE risk factors do not cancel each other in the rat model, but lead to miR-708-5p levels comparable to the G and E conditions alone (Fig. 2E, fourth bar)." In their reply the authors explain that such a non-additive effect would be expected if both G and E risk factors impact the same pathway. However, in the Discussion section the authors note "these results suggest that genetic and environmental risk might impinge on common pathways triggering miR-708-5p expression," and it is not fully explained why the fourth bar is not differentially expressed compared to the first bar. The results section also seems inconsistent with the statement that miR-708-5p upregulation is independent of social housing conditions, which appears contradictory based on the "ns" indication between the first and fourth bars. Additionally, a point to consider is the use of relative expression scales between Figures 2A and 2E. U6 may not always be an ideal control for normalizing microRNA expression, and the negative values across all gene expression scales suggest this could be a potential issue.

The expanded view figures are raw data in excel format. The data should be presented in an easily interpretable figure to aid the reviewers and readers interpreting the results.

As a final note, the authors make several strong claims throughout the manuscript, such as "robust," "comprehensive," "unbiased," and "strong evidence," which are not always fully supported by the results. While it is understandable that adding more samples and experiments to strengthen certain aspects may not be feasible, the authors are encouraged to consider moderating these interpretations.

Referee #1:

Having thoroughly read the manuscript, my suggestion for the authors going forward is to majorly reorganize the main result presentation, prioritizing consistency. The truly strong findings are presented in Figures 4 - 6 and are suggested to be the central focus. The following limitations are hindering the manuscript in its present form:

Figure 1:

- miR-708-5p is not a top hit in the high-throughput analysis and is subsequently chosen on the basis of a priory hypothesis, while the earlier rationale provided is that of an unbiased approach;

- We have provided a rationale for choosing miR-708-5p for further analysis on p. 5-6 of the revised manuscript ("However, miR-708-5p was of specific interest, since its expression was previously shown to be induced by various forms of cellular stress (Behrman *et al*, 2011; Lin *et al*, 2015; McIlwraith *et al*, 2022; Rodriguez-Comas *et al*, 2017; Yang *et al*, 2015) and it had been associated with BD (Forstner *et al.*, 2015). We therefore decided to focus on miR-708-5p for our further studies.").

- only female subjects from the cohort were considered, which was not a central aim, with the discrepancy not being clearly explained in the methods (comment #1 below);

- We have provided the rationale for only considering female subjects from the cohort on p. 5 of the revised manuscript ("Only samples obtained from female subjects were considered, since we were unable to form homogeneous GR and ER groups for males due to an overall lower representation of male samples in the cohort.")

Figure 2

*- the male rat model with both the genetic risk and environmental hindrance (isolated *Cacna1c*^{+/-} group) does not show differential expression to control;*

- We agree that the finding that miR-708-5p levels in isolated *Cacna1c* het rats are not significantly increased compared to those in group-housed WT rats

(Fig. 2d) is surprising. However, we also want to stress that there is a strong trend towards increased levels ($p=0.11$), suggesting that the response to early life stress is attenuated in the context of *Cacna1c* heterozygosity. The molecular mechanisms underlying this attenuation are a topic for future studies. We have now rephrased our conclusions regarding this figure on p.7 of the revised manuscript.

- *no statistical differences are detected in female rats;*

- Female rats were not included in our study

Figure 3

- *the study solely investigates miR-708-5p in the hippocampus without indicating a link between them;*

- We have provided a rationale for focusing on the hippocampus on p.7 of the revised manuscript (“The hippocampus was chosen for functional manipulation since it represents a key brain structure for cognitive and emotional processing, and our previous data indicates strong effects of juvenile social isolation on hippocampal miRNA expression (Martins *et al.*, 2022; Valluy *et al.*, 2015) (Fig. 2E)”.

- *the study suggests that hippocampal overexpression of miR-708-5p may be associated with MD-related behaviors in the mouse model, based on findings that appear to align with an "anti-depressant" function of miR-708-5p, particularly during a manic-like state. However, the results do not provide compelling evidence for a specific link to bipolar disorder (BD). It is possible that the microRNA is enhancing resiliency to stress, potentially at the expense of memory and attention. Additionally, the author's conclusion is not supported by the limited relationship between miR-708-5p and the Young Mania Rating Scale (YMRS) score.*

- We acknowledge that in our mouse model, we were only able to recapitulate certain endophenotypes associated with bipolar disorders, such as altered behavioural despair and impaired short-term memory. We provide a thorough discussion of the possible reasons for the absence of other BD-associated behavioural endophenotypes on p.15-16 of the revised manuscript. (“However, in our study, the antidepressant-like behavior was neither

accompanied by increased exploratory behavior in the OFT, nor increased risk-taking during EPM testing, both of which are usually observed in mouse models of mania. We consider two possible explanations for this discrepancy. First, contrary to most of the models reported in the literature, our animal model is characterized by an acute and selective overexpression of miR-708-5p in the hippocampus. Therefore, we might expect to observe endophenotypes dependent on the hippocampus (e.g., antidepressant-like behavior, cognition), but not those related to other brain regions, such as the amygdala, PFC, or cerebellum (e.g., risk-taking behavior, hyperactivity). Alternatively, the miR-708-5p/Nnat pathway might selectively control specific aspects of manic- and depression-like behavior. To distinguish between these possibilities, it will be important to study the role of miR-708-5p overexpression in other brain areas relevant for MD-associated behaviors in the future, in particular the PFC.”).

The major comments from original review:

#1 As previously noted, it is unclear whether using only female subjects from the same cohort for analysis in Figure 1 is more appropriate, while sex composition does not appear to be a concern for Figure 6. It would be helpful if the authors could clarify the total number of participants in the cohort after applying the exclusion and selection criteria, and provide subsections with additional details about the subjects described in Table 1 and Supplementary Table 1.

- Please see our comment above regarding the selection of female subjects in Figure 1. We provide detailed data about these subjects in Appendix Table S1 of the revised manuscript.

#2-4 When considering bipolar disorder, memory and attention may not be the most prominent deficits that distinguish this psychiatric condition from others. The rationale for focusing on memory/attention, the hippocampus, and the link to this microRNA appears somewhat unclear. For example, in the results section, the transition from observing expression in regions associated with mood disorders and cognitive function, such as the amygdala, frontal cortex, and hippocampus, to solely assessing miR-708-5p expression in rat hippocampal primary neurons introduces a gap. Additionally, while the authors observe effects such as amelioration of despair

(as measured by the tail-suspension test) and trending compulsive behavior, these effects are only present in males, which complicates the interpretation of the results that begins with female subjects.

- Please see our comments above regarding the presence/absence of specific BD-associated behavioural endophenotypes in our mouse model and the focus on the hippocampus. With regard to the sex-specificity, we also provide an intense discussion on p.17 of the revised manuscript: (“Together, these observations suggest a potential sex-specific role of miR-708-5p in BD, which also aligns with our results from mouse behavior. Sex differences in BD manifest across various aspects, ranging from clinical symptoms to the progression of the disorder. Males typically encounter their first manic episode at a younger age compared to females, who are more likely to experience a depressive episode at the onset of BD (Kennedy *et al*, 2005). In contrast, females experience a higher incidence of depressive episodes and hypomania, leading to a more frequent diagnosis of BD type II (Difflorio & Jones, 2010). Females are also more susceptible to mixed episodes (Arnold *et al*, 2000), rapid cycling of mood phases (Tondo & Baldessarini, 1998), and show a greater likelihood of attempting suicide (Clements *et al*, 2013). Thus, it is tempting to speculate that miR-708-5p might play an important role in male-specific aspects of BD, e.g., the development of more common and intense manic episodes. In this regard, miRNAs have been previously implicated in the sexually dimorphic control of circadian, cholinergic, and neurokinin pathways in BD (Lobentanzer *et al*, 2019).”)

#5 Introducing the link between miR-499-5p and miR-708-5p functions in the hippocampus, for example by manipulating the two microRNAs in the animal model concurrently, would have been a more logical precedent to the ROC curve analysis with two candidates. The characterization of miR-499-5p in the other publication is consistent with the use of the rat model and the hippocampus rationale that extends between the animal model and clinical cohort. In the current manuscript, a lot of inconsistencies are introduced with different species, sex, and models which challenge the cohesiveness of the story.

- miR-499-5p is not the main focus of this study, but was extensively characterized in a previous paper (Martins *et al.*, EMBO Reports, 2022).

Therefore, manipulating miR-499-5p and miR-708-5p together is clearly beyond the scope of the present manuscript. However, due to its known role in BD, we included it to further improve the diagnostic discrimination between bipolar and unipolar depression. We have now added additional information about miR-499-5p in the introduction of the revised paper (p.4). In general, we attempted to address all inconsistencies with regard to species, sex, and models in the revised manuscript.

#7 The choice for mononuclear blood cells suggests that there is either a hypothesized involvement of the immune-related response or altered microRNA biogenesis mechanism in those cells specifically. The author's suggestion of a potential route involving the "release of miR-708 encapsulated in extracellular vesicles (EVs) from neurons into the bloodstream, which are subsequently taken up by PBMCs" seems speculative, given the uncertainty around the likelihood of such an occurrence. This raises the question of why microRNA expression isn't measured directly in EVs or in whole blood. Regarding the relaxed statistical cut-off in the RNA-sequencing experiment, the authors commented an acknowledgement of the "high variability of human PBMC expression data," which adds to the question of the origin of microRNAs in PBMC fractions. While the characterization of circulating microRNAs is valuable in psychiatric research, the rationale provided for this specific approach lacks clarity.

- We have provided a rationale for using PBMCs instead of other blood fractions on p.5 of the revised manuscript. ("We chose PBMCs for this analysis since they represent a very reliable source of miRNAs, and the correlation between brain expression and PBMCs is often higher as between brain and plasma or serum (Kos et al., Front. Mol. Neurosci. 2022)."). Moreover, we discuss a potential route of miRNA transfer via brain-derived extracellular vesicles, which is based on previous publications, on p.16 of the revised manuscript. ("How altered miR-708-5p levels in PBMCs are linked to human brain states is currently unknown, but one potential route could involve the release of miR-708 encapsulated in extracellular vesicles (EVs) from neurons into the bloodstream. While the crossing of EVs through the BBB has been reported in the literature (Shi et al, 2019), whether EVs can be taken up by PBMCs is not yet fully understood"). Measuring miR-708-5p in other blood

fractions, e.g., EVs or plasma, would be informative, but is beyond the scope of the present study.

In reference to Figure 2E, I had previously noted that the combined genetic and environmental risk factors seemed to cancel each other out. The authors responded by stating that "GxE risk factors do not cancel each other in the rat model, but lead to miR-708-5p levels comparable to the G and E conditions alone (Fig. 2E, fourth bar)." In their reply the authors explain that such a non-additive effect would be expected if both G and E risk factors impact the same pathway. However, in the Discussion section the authors note "these results suggest that genetic and environmental risk might impinge on common pathways triggering miR-708-5p expression," and it is not fully explained why the fourth bar is not differentially expressed compared to the first bar. The results section also seems inconsistent with the statement that miR-708-5p upregulation is independent of social housing conditions, which appears contradictory based on the "ns" indication between the first and fourth bars. Additionally, a point to consider is the use of relative expression scales between Figures 2A and 2E. U6 may not always be an ideal control for normalizing microRNA expression, and the negative values across all gene expression scales suggest this could be a potential issue.

- Please see our comments above ("Figure 2") regarding the interpretation of Fig. 2E. We have used U6 snRNA as a housekeeping control for microRNA qPCR experiments for many years and have always obtained consistent results. The negative values mean that U6 is more highly expressed than miR-708, which is expected.

The expanded view figures are raw data in excel format. The data should be presented in an easily interpretable figure to aid the reviewers and readers interpreting the results.

- We now present the raw data according to the requirements of the journal.

As a final note, the authors make several strong claims throughout the manuscript, such as "robust," "comprehensive," "unbiased," and "strong evidence," which are not

always fully supported by the results. While it is understandable that adding more samples and experiments to strengthen certain aspects may not be feasible, the authors are encouraged to consider moderating these interpretations.

- We have toned down our statements in the revised manuscript when appropriate.

Prof. Gerhard Schratt
ETH Zurich - D-HEST
Systems Neuroscience, Bldg Y17 L48
Institute for Neuroscience
Winterthurerstr. 190
Zurich, Zurich 8057
Switzerland

Dear Gerhard,

I am very pleased to accept your manuscript for publication in the next available issue of EMBO reports. Thank you for your contribution to our journal.

Best,
Esther
